# The regulatory relationship between NAMPT and PD-L1 in cancer and identification of a dual-targeting inhibitor

Yuan Yang[1,2,6], Zefei Li[3,6], Yidong Wang[1,2], Jiwei Gao[1,2], Yangyang Meng[3], Simeng Wang [1,2], Xiaoyao Zhao[1,2], Chengfang Tang[1,2], Weiming Yang[1,2], Yingjia Li[1,2], Jie Bao[4], Xinyu Fan[5], Jing Tang [4], Jingyu Yang[1,2], Chunfu Wu[1,2], Mingze Qin [3✉] & Lihui Wang [1,2✉]

## Abstract

**Cancer is a heterogeneous disease. Although both tumor metabolism and tumor immune microenvironment are recognized as driving factors in tumorigenesis, the relationship between them is still not well-known, and potential combined targeting approaches remain to be identified. Here, we demonstrated a negative correlation between the expression of NAMPT, an NAD⁺ metabolism enzyme, and PD-L1 expression in various cancer cell lines. A clinical study showed that a NAMPT^High PD-L1^Low expression pattern predicts poor prognosis in patients with various cancers. In addition, pharmacological inhibition of NAMPT results in the transcription upregulation of PD-L1 by SIRT-mediated acetylation change of NF-κB p65, and blocking PD-L1 would induce NAMPT expression through a HIF-1-dependent glycolysis pathway. Based on these findings, we designed and synthesized a dual NAMPT/PD-L1 targeting compound, LZFPN-90, which inhibits cell growth in a NAMPT-dependent manner and blocks the cell cycle, subsequently inducing apoptosis. Under co-culture conditions, LZFPN-90 treatment contributes to the proliferation and activation of T cells and blocks the growth of cancer cells. Using mice bearing genetically manipulated tumors, we confirmed that LZFPN-90 exerted target-dependent antitumor activities, affecting metabolic processes and the immune system. In conclusion, our results demonstrate the relevance of NAD⁺-related metabolic processes in antitumor immunity and suggest that co-targeting NAD⁺ metabolism and PD-L1 represents a promising therapeutic approach.**

**Keywords** NAMPT/PD-L1; Co-targeting Therapy; Cancer; Tumor Immune Microenvironment
**Subject Categories** Cancer; Metabolism

## Introduction

Cancer is a complex and heterogeneous disease caused by multiple genetic changes and poses a major threat to human life (Siegel et al, 2023). The cellular and structural heterogeneity of cancer determines the complexity of its metabolic patterns (Vitale et al, 2021). Interestingly, these complex and diverse metabolic patterns also affect immune cells (Ghesquière et al, 2014; Mao et al, 2021). Increasing evidence suggests that cancer metabolism not only plays a crucial role in sustaining the growth and survival of tumor cells but also contributes to the regulation of antitumor immune responses by regulating the release of metabolites and the expression of immune molecules (Leone et al, 2019; Xia et al, 2021). However, the actual process underlying the interaction between cancer cell metabolism and immune response is not well understood.

Nicotinamide adenine dinucleotide (NAD⁺) is a coenzyme involved in redox reactions (Covarrubias et al, 2021). Nicotinamide phosphoribosyltransferase (NAMPT), the rate-limiting enzyme in NAD⁺ biosynthesis, is a key enzyme in cell metabolism that influences the activity of NAD⁺-dependent enzymes to regulate tumor metabolism (Gardell et al, 2019). It has been reported that tumor cells consume NAD⁺ at a faster rate than normal cells, displaying very active metabolism, and expressing high levels of NAMPT to satisfy the high demand for NAD⁺ (Nakajima et al, 2009). Several studies have shown that NAMPT and NAD⁺ are frequently upregulated in several human malignancies and play key roles in the development, progression, and recurrence of cancer (Lucena-Cacace et al, 2018; Zhang et al, 2019). Furthermore, unlike tumor cells, which mostly depend on NAMPT, normal cells can synthesize NAD⁺ through other biosynthetic pathways, such as nicotinic acid phosphoribosyltransferase (NAPRT) (Duarte-Pereira et al, 2021). NAMPT and NAD⁺ not only affect tumor metabolism but also affect the tumor immune microenvironment. It has been reported that NAD⁺ can be secreted into the extracellular space where it acts as a proinflammatory cytokine that stimulates

¹Department of Pharmacology, Shenyang Pharmaceutical University, Shenyang, PR China. ²Benxi Institute of Pharmaceutical Research, Shenyang Pharmaceutical University, Benxi, PR China. ³Key Laboratory of Structure-Based Drug Design and Discovery, Ministry of Education, Shenyang Pharmaceutical University, Shenyang, PR China. ⁴Research Program in Systems Oncology, Faculty of Medicine, University of Helsinki, Helsinki 00290, Finland. ⁵Department of Pharmacy, Shengjing Hospital of China Medical University, 110004 Shenyang, PR China. ⁶These authors contributed equally: Yuan Yang, Zefei Li. ✉E-mail: qinmingze@syphu.edu.cn; lhwang@syphu.edu.cn

granulocytes, augmenting chemotaxis (Gardell et al, 2019; Nakajima et al, 2009), which in turn fine-tunes the immune response (Adriouch et al, 2007). Furthermore, NAMPT plays an important regulatory role in the antitumor T-cell response (Navas and Carnero, 2021) and is essential for T-cell activation (Wang et al, 2021). Thus, NAMPT plays an important role in tumor metabolism and immune modulation and is therefore a potential therapeutic target for treating tumors.

In the past decades, cancer treatment has been revolutionized by the rise of checkpoint inhibitor immunotherapy. The binding of programmed death-1 (PD-1) to the surface of T cells and PD-1 ligand 1 (PD-L1) to tumor cells can prevent the immune-killing effect of T cells on tumor cells and promote the immune escape of tumor cells (Pang et al, 2023; Yi et al, 2022). Checkpoint inhibitor therapies aim to block PD-1/PD-L1 interaction to boost the immune recognition of tumor cells. However, the main challenges of this therapy are its low response rate and acquired resistance (Jiang et al, 2020). Many factors contribute to immunotherapy resistance, including the expression and presentation of tumor antigens, the immunosuppressive microenvironment, and metabolism or nutrient availability in the tumor microenvironment (TME) (Muir and Vander Heiden, 2018). Hence, a more comprehensive understanding of the regulatory mechanisms of the PD-1/PD-L1 axis is vital for developing combined therapeutic strategies to overcome resistance to the PD-1/PD-L1 blockade (Vesely et al, 2022).

In this study, by investigating the link between NAMPT and PD-L1, we identified a relationship between tumor NAD$^+$ metabolism and tumor immune escape. We also explored the antitumor effects of an inhibitor that co-targets NAMPT and PD-L1. Our results revealed that the expression of PD-L1 and NAMPT was negatively correlated by epigenetic and glycolysis regulatory mechanism, and a NAMPT$^{High}$ PD-L1$^{Low}$ expression pattern predicted a lower survival rate. We also designed, synthesized, and screened a dual-targeting inhibitor, which effectively inhibited the PD-1/PD-L1 interaction and NAMPT activity and affected multiple malignant phenotypes. Our findings raise the possibility that the pharmacological blockade of NAMPT and simultaneous immune checkpoint blockade represent a promising strategy for cancer therapy.

## Results

### NAMPT inhibition promoting PD-L1 expression mediated by SIRT epigenetic regulation

To determine the regulatory relationship between NAMPT and PD-L1, we analyzed the expression of NAMPT and PD-L1 in the TCGA database. Our results showed that NAMPT and PD-L1 (*CD274*) are highly expressed in most cancers (Fig. 1A). Next, we analyzed the relationship between co-expression levels and prognosis of NAMPT and PD-L1, finding that patients with high NAMPT expression and low PD-L1 expression had the worst prognosis (Fig. 1B). This observation suggests that the NAMPT$^{High}$ PD-L1$^{Low}$ expression pattern is significant in tumor progression. Consistent with TCGA data, we experimentally examined the expression of NAMPT and PD-L1 in 11 cell lines from different cancers and found that the expression levels of the two proteins

were negatively correlated (Fig. 1C). Next, we investigated whether the pharmacological blockade of NAMPT upregulates PD-L1 protein levels. Our results indicated that FK866, an inhibitor of NAMPT, upregulated global and membrane PD-L1 expression in A2780 cells, whereas the upregulation of PD-L1 induced by FK866 was reversed by treatment with the NAD$^+$ precursor NMN (Fig. 1D,E). In addition, we performed similar experiments by specific inhibitor FK866 or gene manipulation on murine-derived LLC cells and obtained consistent results (Fig. 1D,E; Appendix Fig. S1A). Importantly, the in vivo study also showed that the administration of FK866 could lead to the upregulation of PD-L1 (Appendix Fig. S1B). The above results indicate that PD-L1 expression is upregulated when NAMPT is inhibited.

SIRT1/2 is a well-known NAD$^+$-dependent deacetylase, and NAD$^+$ concentration can affect the activation of SIRT1 (Zhao et al, 2021). Our results indicated that the pharmacological blockade of NAMPT upregulates PD-L1 levels. Therefore, we tested whether NAD$^+$-deprivation-induced impairment of SIRT activity is responsible for the upregulation of PD-L1 in cancer cells. We treated A2780 and LLC cells with Sirtinol, a specific SIRT1/2 inhibitor, and found that PD-L1 was upregulated at both the mRNA and protein levels (Fig. 1F,G). Notably, compared to the monotherapy groups, combined treatment with FK866 and Sirtinol resulted in an obvious increase in the protein expression of PD-L1 (Fig. 1H; Appendix Fig. S1A). To assess whether FK866 is involved in the regulation of SIRT activity, we also detected the expression of Ac-Tubulin, a substrate of SIRT, in LLC cells. Our results indicated that treatment of FK866 could increase Ac-Tubulin, while NMN reversed the increase of the Ac-Tubulin (Appendix Fig. S1C). The above results showed that the inhibition of NAMPT-induced PD-L1 expression through SIRT. In order to further explore how NAMPT affects the expression of PD-L1 through SIRT, we examined the SIRT-regulated acetylation K310 of NF-κB p65 (Yeung et al, 2004; Chen et al, 2002), which is considered as the transcription factor of PD-L1 (Zhou et al, 2022). After treatment with FK866 and NMN to A2780 and LLC cells, we detected the acetylation level of p65-K310. The data revealed that FK866 increased the acetylation of p65-K310, while NMN reversed the event (Fig. 1I; Appendix Fig. S1D). Next, we want to know whether NAMPT inhibition could affect the binding of p65-K310 on the PD-L1 promoter. According to the JASPAR database, the p65 binding site on the PD-L1 promoter was predicted (Fig. 1J). The results of ChIP-qPCR showed that, after treatment with FK866 or Sirtinol, the enrichment of p65-K310 on the P3 region, but not P1 and P2 region, was increased in A2780 cells, while NMN blocked this binding (Fig. 1J), which suggests that NAMPT inhibition affects PD-L1 expression through regulating acetylation of NF-κB p65 and its activity. Taken together, we found a negative relationship between NAMPT and PD-L1, and the promotion of PD-L1 expression after the treatment of NAMPT inhibition, which was mediated by the epigenetic regulation of SIRT and subsequently resulted in transcription activation of PD-L1 by NF-κB p65.

### PD-L1 inhibition inducing expression of NAMPT by activating the glycolytic pathway at HIF-1α-dependent manner

In order to further explore the relationship between NAMPT and PD-L1, we also detected NAMPT expression after treatment with

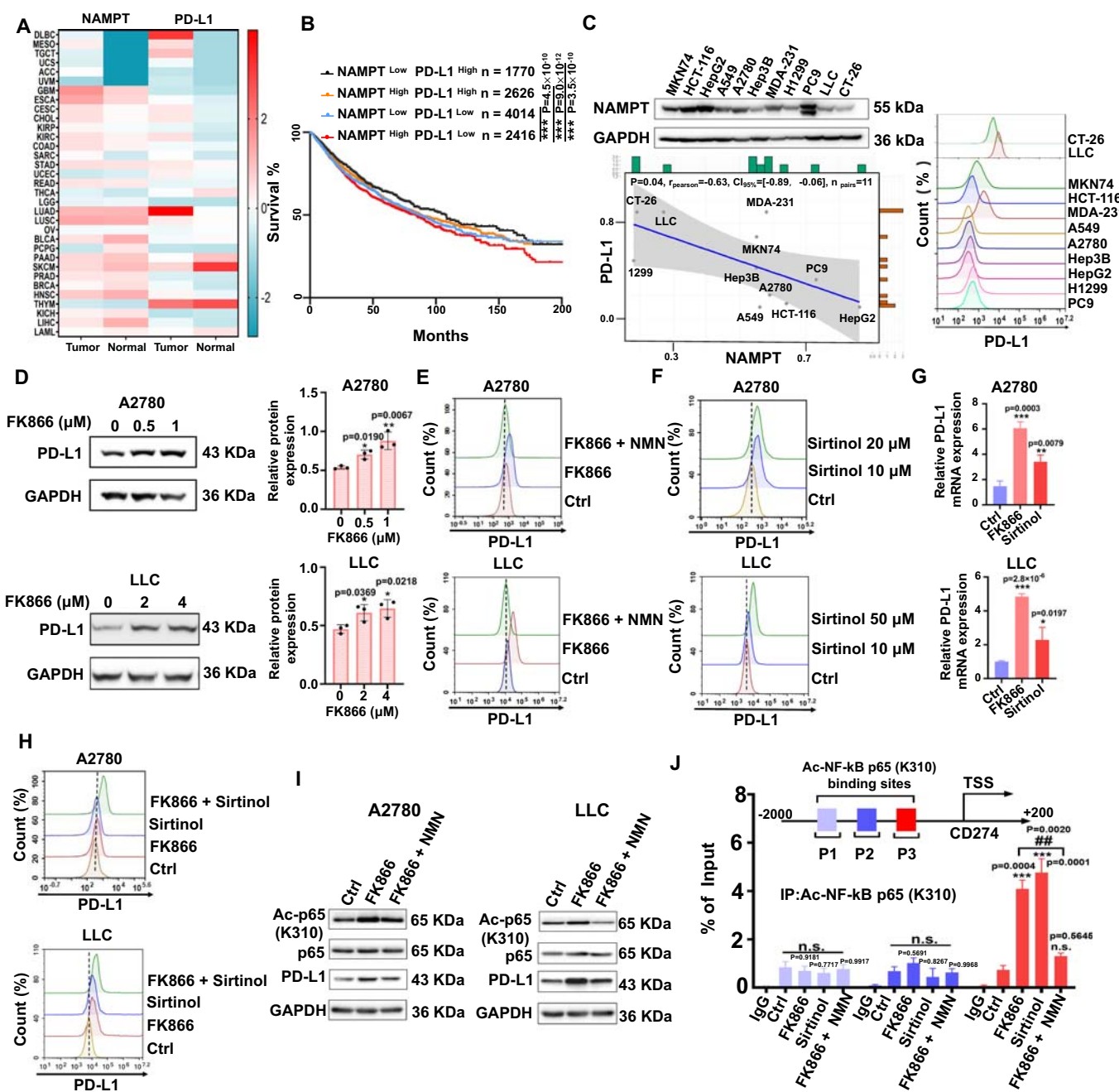

**Figure 1. NAMPT inhibitor upregulates PD-L1 in mouse and human cells.**

(A) Analysis of NAMPT and PD-L1 mRNA levels in various cancer and para-cancerous tissues from TCGA data. (B) Kaplan–Meier curve analysis of overall patient survival based on different NAMPT and PD-L1 expression levels in the TCGA database (NAMPT [Low] PD-L1 [High] $n = 1770$; NAMPT [High] PD-L1 [High] $n = 2626$; NAMPT [Low] PD-L1 [Low] $n = 4014$; NAMPT [High] PD-L1 [Low] $n = 2416$). (C) Correlation of the level of PD-L1 membrane protein with the level of NAMPT protein in different cancer cells. $r =$ Pearson correlation coefficients; $P = P$ values. (D) Western blot analysis of PD-L1 protein expression after adding FK866 to A2780 and LLC cells for 72 h (A2780: 0, 0.5, 1 μM; LLC: 0, 2, 4 μM), $n = 3$ per group. (E) Effect of FK866 and NMN for 72 h on PD-L1 levels in A2780 cells (FK866: 0.4 μM; NMN: 1 mM) and LLC cells (FK866: 0.8 μM; NMN: 1 mM), which were detected by flow cytometry. (F) Effect of Sirtinol for 72 h on the expression of PD-L1 in A2780 cells (10, 20 μM) and LLC cells (10, 50 μM), which were detected by flow cytometry. (G) Expression of PD-L1 mRNA after adding FK866 and Sirtinol to A2780 cells for 72 h (FK866: 0.4 μM; Sirtinol: 10 μM) and LLC cells (FK866: 0.8 μM; Sirtinol: 10 μM). (H) Effect of FK866 and/or Sirtinol for 72 h on PD-L1 levels in A2780 cells (FK866: 0.4 μM; Sirtinol: 10 μM) and LLC cells (FK866: 0.4 μM; Sirtinol: 10 μM), which were detected by flow cytometry ($n = 3$ per group). (I) Analysis of Ac-p65 (K310), PD-L1 protein expression after adding FK866 and NMN to A2780 and LLC cells for 72 h by western blot. (A2780: FK866 0.5 μM, NMN 1 mM; LLC: FK866 2 μM, NMN 1 mM). (J) After treatment with FK866, Sirtinol and NMN for 72 h in A2780 cells, the enrichment of Ac-p65 (K310) in PD-L1 promoter region (P1, P2, P3) was detected by ChIp-qPCR (A2780: FK866 0.5 μM, Sirtinol 10 μM; NMN 1 mM), $n = 3$ per group. Data information: Data represent different numbers ($n$) of biological replicates. Data are shown as mean ± SEM. One-way ANOVA followed by Bonferroni's test is used in (B, D, G, J). Two-tailed Student's $t$ test is used in (B, J). *$P < 0.05$, **$P < 0.01$, ***$P < 0.001$ compared with the control group. ##$P < 0.01$ compared FK866 group with (FK866 + NMN) group. n.s. not significant. Source data are available online for this figure.

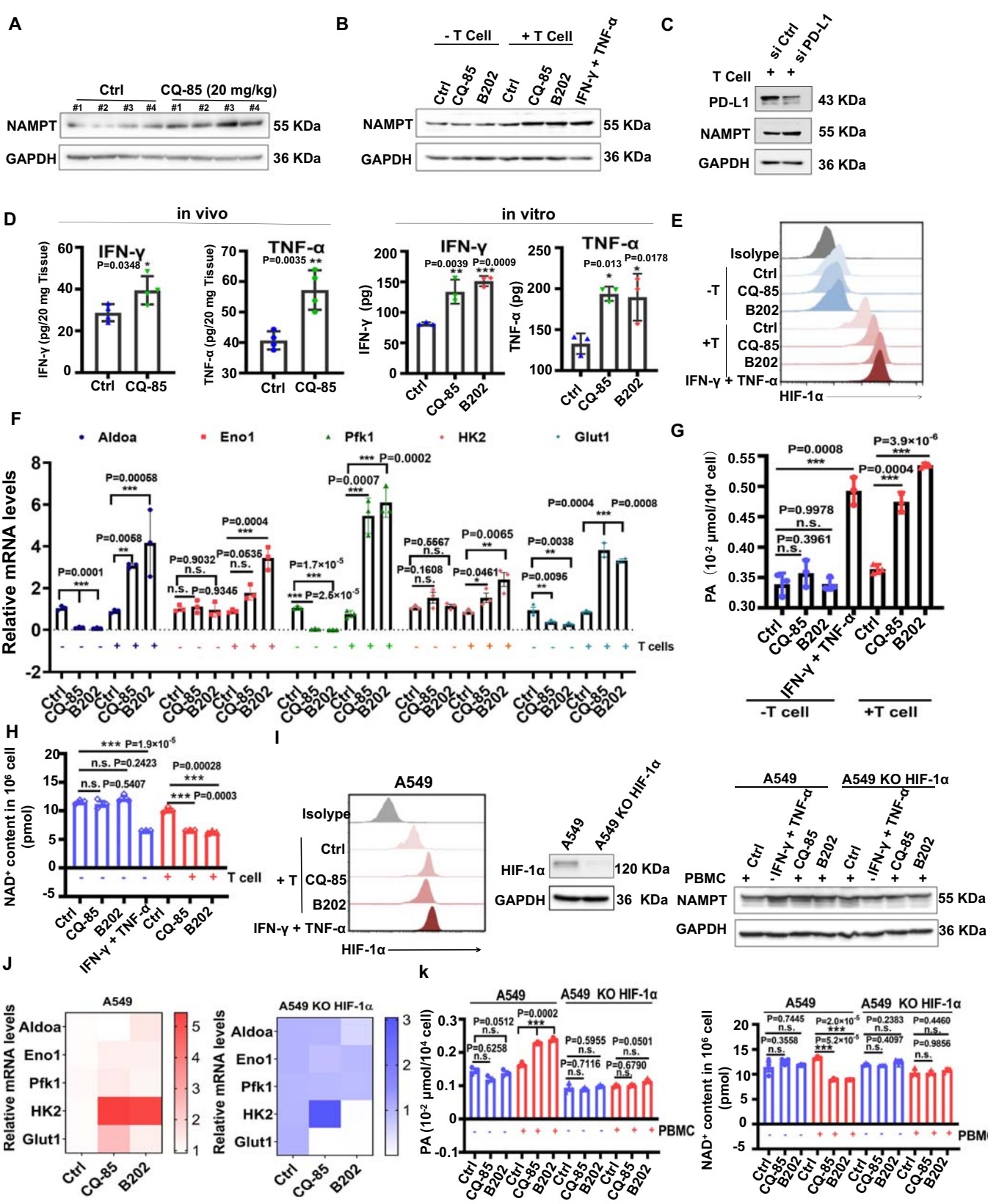

◄

**Figure 2. The inhibition of PD-L1 induces NAMPT depending on T cells.**

(A) Relative expression of NAMPT in tumors from Ctrl and CQ-85 (20 mg/kg) groups in which mice were inoculated with LLC cells with double overexpression of NAMPT and PD-L1 (n = 4 mice/group). (B) With or without the T cells system, western blot analyzed the NAMPT protein expression after adding CQ-85 (0.1 μM), BMS202 (0.1 μM), IFN-γ and TNF-α (50 ng/ml each) for 72 h in LLC cells. (C) After co-culture, the western blot analyzed the NAMPT protein expression after using si Ctrl and si PD-L1 in LLC cells. (D) The levels of IFN-γ and TNF-α were detected in tumor tissues of LLC OE NAMPT/PD-L1 mouse models by ELISA (left), n = 4 mice/group. The contents of IFN-γ and TNF-α in LLC cells were detected by ELISA under the co-culture condition (right), n = 3 per group. (E) With or without the T-cells system, the expression of HIF-1α was detected by flow cytometry after adding CQ-85 (0.1 μM), BMS202 (0.1 μM), IFN-γ and TNF-α (50 ng/ml each) for 72 h to LLC cells. (F) With or without the T cells system, the mRNA levels of several glycolytic genes were tested in LLC cells by RT-PCR (n = 3 per group). (G, H) With or without the T-cells system, the content of pyruvate (PA) (G) and NAD+ (H) was detected in LLC cells (n = 3 per group). (I) With or without PBMC cells system, the expression of HIF-1α was detected by flow cytometry after adding CQ-85 (0.1 μM), BMS202 (0.1 μM), IFN-γ and TNF-α (50 ng/ml each) for 72 h to A549 cells (left). Western blot verified the A549 knocks out HIF-1α cells (middle). Co-cultured with PBMC cells, western blot analyzed the NAMPT protein expression after adding CQ-85 (0.1 μM), BMS202 (0.1 μM), IFN-γ and TNF-α (50 ng/ml each) for 72 h in A549 and A549 KO HIF-1α cells (right). (J) Heatmap represented the expression of several glycolytic genes in A549 and A549 KO HIF-1α cells co-cultured with PBMC cells by RT-PCR. (K) With or without PBMC cells system, the content of pyruvate (PA) and NAD+ was detected in A549 and A549 KO HIF-1α cells (n = 3 per group). Data information: Data represent different numbers (n) of biological replicates. Data are shown as mean ± SEM. One-way ANOVA followed by Bonferroni's test is used in (D, F, G, H, K). Two-tailed Student's t test is used in (F, D). *$P < 0.05$, **$P < 0.01$, ***$P < 0.001$ compared with the control group. n.s. not significant. Source data are available online for this figure.

CQ-85, a small-molecule inhibitor of PD-L1, in LLC constructed xenograft model. Interestingly, our results revealed that inhibition of PD-L1 can upregulate the protein expression of NAMPT compared with the control group (Fig. 2A). The upregulation of NAMPT induced by PD-L1 inhibition was further verified by various PD-L1 inhibitors, including CQ-85 and BMS202, and gene manipulation in vitro (Fig. 2B,C). Notably, the induction of NAMPT by PD-L1 inhibition was dependent on the presence of T cells (Fig. 2B), suggesting the regulation of PD-L1 and NAMPT had immune microenvironment-dependent characteristics.

It has been reported that the presence of inflammatory factors IFN-γ and TNF-α can activate the NAD+-dependent glycolytic pathway by upregulating the expression of HIF-1α (Fang et al, 2023). We hypothesized that PD-L1 inhibition will increase the expression of NAMPT by promoting inflammatory factors release of T cells. Thus, we first assess whether inflammatory factors IFN-γ and TNF-α can regulate NAMPT expression. Our results indicated that the exogenous addition of IFN-γ and TNF-α, without T cells, could induce expression of NAMPT in LLC cells (Fig. 2B). Then, we explored whether PD-L1 inhibitors could regulate the secretion of IFN-γ and TNF-α. The ELISA results showed that treatment with PD-L1 inhibitors would promote the secretion of IFN-γ and TNF-α in tumor tissue and in vitro co-culture system (Fig. 2D). Next, the relationship between PD-L1 inhibition and HIF-1α and glycolysis was further explored. Our results showed that PD-L1 inhibitors could upregulate the expression of HIF-1α under the co-culture with T cells, as well as exogenous addition of IFN-γ and TNF-α (Fig. 2E). At the same time, the results indicated that treatment with PD-L1 inhibitors also could increase the mRNA levels of glycolytic-related HIF-1α target genes (Fig. 2F), raise the contents of pyruvate and consumption of NAD+ under the co-culture condition (Fig. 2G,H). Furthermore, the crucial role of HIF-1α in this process was confirmed by gene manipulation. Compared with wild-type A549 cells, PD-L1 inhibitors induced changes, including the expression of NAMPT and glycolytic-related genes, the contents of pyruvate and NAD+ (Fig. 2I–K) disappeared in HIF-1α knock-out A549 cells. Taken together, these results indicated that PD-L1 inhibition can upregulate the expression of HIF-1α by increasing the level of inflammatory factors, activating the glycolytic pathway promoting the consumption of NAD+, and subsequently inducing NAMPT expression.

## Synthesis and screening of the dual-targeting compound LZFPN-90

The results above suggested that synthesizing a dual NAMPT/PD-L1 inhibitor may be a successful strategy for cancer therapy. To date, no dual inhibitors that simultaneously target the PD-1/PD-L1 interaction and NAMPT have been reported. To explore the possibility of generating such an inhibitor, we first dissected the structural characteristics of the inhibitors of each target. In recent years, several biphenyl-based PD-1/PD-L1 interaction inhibitors with potent immunoregulatory activities have been identified, as exemplified by CQ-34 (Qin et al, 2020) and CQ-85 (Qin et al, 2021) reported by us, and several inhibitors developed by other scientific research groups (Lai et al, 2022; Pan et al, 2021) (Appendix Fig. S2A). Structurally, these inhibitors contain four moieties: a core group featuring a biphenyl or analogous fragment, a linker, an aryl group, and a hydrophilic tail group. Hydrophobic interactions, such as π-π interactions formed between the core group and the aryl group with PD-L1 residues, significantly contribute to inhibitor binding. The tail group, oriented toward the solvent, can form hydrogen bonds with the PD-L1 protein, improving the inhibitor activity. Thus, rational modification of the tail group is a feasible way to discover novel inhibitors with favorable potency. Several NAMPT inhibitors have been reported, including FK866 (Hasmann and Schemainda, 2003) and CHS828 (Olesen et al, 2008) (Appendix Figs. S2B and S8), but none have been approved by the U.S. Food and Drug Administration. Classic NAMPT inhibitors are generally linear in shape and consist of a cap group that mimics nicotinamide (the natural substrate of NAMPT), a linker, and a hydrophobic tail group. The pyridine-containing core-binding moiety plays a critical role in the on-target activity of NAMPT inhibitors.

By analyzing the structural characteristics of the above-mentioned inhibitors, we suspected that dual inhibitors of PD-1/PD-L1 interaction and NAMPT could be constructed using a pharmacophore fusion strategy. Thus, taking CQ-34 and FK866 as the lead compounds, after proof-of-concept and rational structure optimization, more than 40 compounds were preliminarily designed and synthesized as potential dual inhibitors. Upon screening the compounds, LZFPN-90 (hereafter referred to as LZ90) was identified with IC$_{50}$ values of 0.013 μM and 0.028 μM for

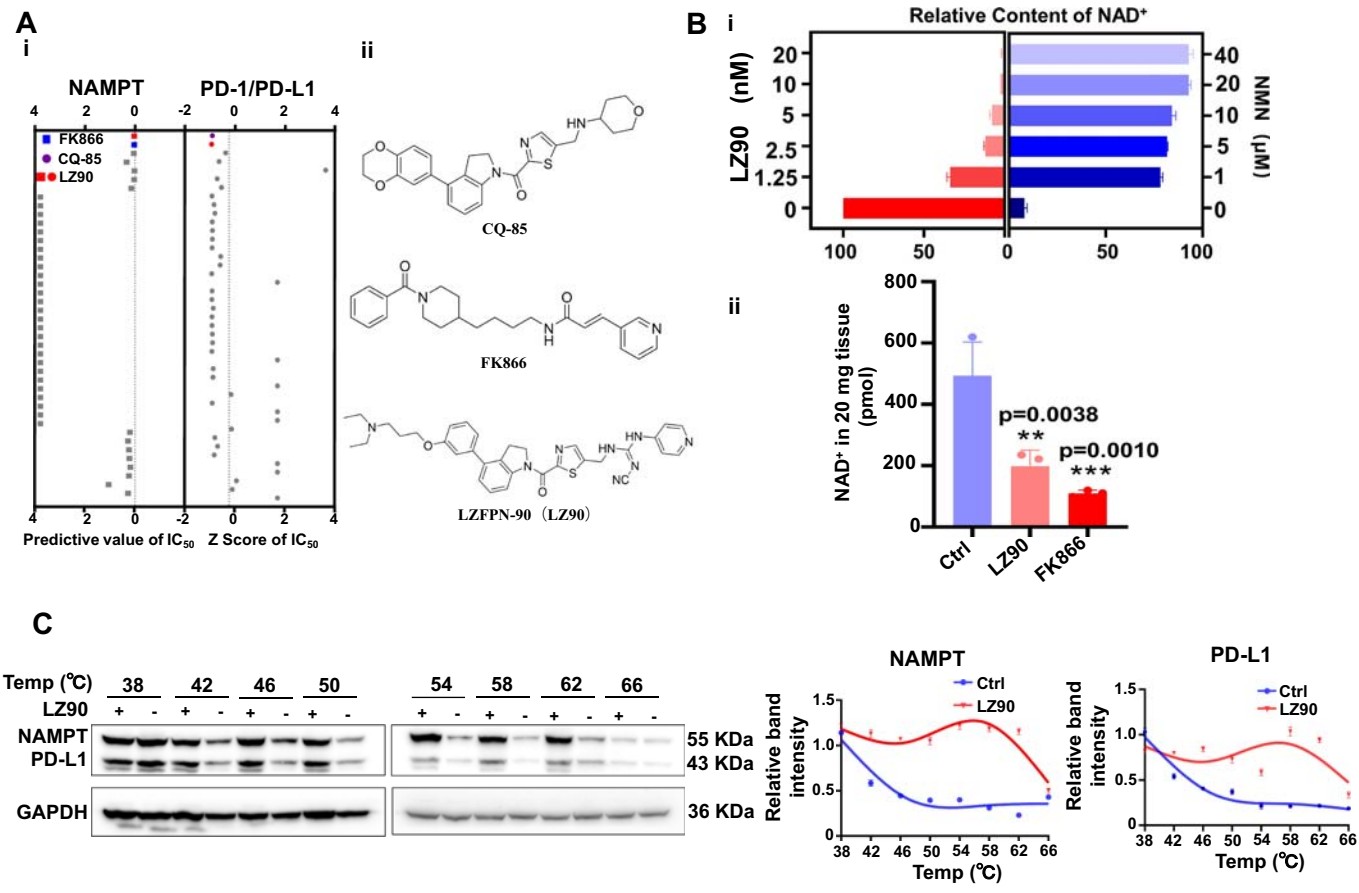

**Figure 3. Screening of compounds for dual targeting of NAMPT/PD-L1.**

(A) Activity screening plots for 40 designed compounds on NAMPT and PD-L1 (left), and structural diagrams of CQ-85, FK866, and LZFPN-90 (hereafter referred to as LZ90) (right). (B) The NAD$^+$ content of A2780 cells was treated with different concentrations of LZ90 and NMN ($n = 3$ per group) (i). Relative NAD$^+$ levels of tissues in FK866 (20 mg/kg) and LZ90 (20 mg/kg) from tumor model in which mice were inoculated with LLC cells with double overexpression of NAMPT and PD-L1 ($n = 3$ mice/group) (ii). (C) Thermal shift assay results. A2780 cells were treated with or without LZ90 (20 nM). Lysates were prepared and heated to the indicated temperature for 3 min and left at room temperature for 3 min, then subjected to immunoblot analysis ($n = 2$ per group). Data information: Data represent different numbers ($n$) of biological replicates. Data are shown as mean ± SEM. One-way ANOVA followed by Bonferroni's test is used in (B). **$P < 0.01$, ***$P < 0.001$ compared with control. n.s. not significant. Source data are available online for this figure.

PD-1/PD-L1 and NAMPT, respectively (Fig. 3A; Appendix Fig. S2C,D). Since the most direct effect of NAMPT inhibition is reduced NAD$^+$ content, we treated A2780 cells with LZFPN-90 and measured NAD$^+$ content. We found that LZFPN-90 treatment resulted in a concentration-dependent reduction in the NAD$^+$ content (Fig. 3B). Moreover, the addition of nicotinamide mononucleotide (NMN), an NAD$^+$ precursor generated by NAMPT, reversed the inhibitory effect of LZFPN-90 on NAD$^+$ content (Fig. 3B). Importantly, our in vivo results indicated that LZFPN-90 reduced NAD$^+$ content in tumor tissues, suggesting that this inhibitor is feasible in vivo (Fig. 3B). This suggests the specificity of LZFPN-90 for the NAMPT target. Next, we performed a thermal migration assay to investigate the binding of LZFPN-90 to both targets. We found that after LZFPN-90 treatment, the stability of NAMPT and PD-L1 increased compared to that in the blank group (Fig. 3C). These results indicate that we successfully identified a dual-targeting small-molecule inhibitor of NAMPT and PD-L1, and we provide preliminary evidence to demonstrate the binding of the inhibitor to its targets.

## The inhibition of LZFPN-90 in cell growth by targeting NAMPT

Inhibition of NAMPT affects the tumor cell cycle and apoptosis (Subedi et al, 2021). To further investigate the effect of LZFPN-90 on tumor cells, we selected three cell lines (A2780, HCT-116, LLC) with different expressions of NAMPT (Fig. 1C) to investigate the inhibitory effect of LZFPN-90 on their proliferation. LZFPN-90 effectively inhibited cell proliferation, and when the higher NAMPT expression, the inhibition in cell growth was stronger (Fig. 4A,B; Appendix Fig. S3A). A similar phenomenon was observed in cells when treated with the NAMPT inhibitor FK866 (Appendix Fig. S3B). Next, we constructed NAMPT knockdown LLC cells and detected the sensitivity to LZFPN-90. Our results demonstrated that the knockdown of NAMPT would result in the decreased sensitivity of LZFPN-90 to LLC cells (Appendix Fig. S3C), suggesting that the anti-proliferation effect of LZFPN-90 was dependent on the NAMPT level. To further verify the finding, we generated overexpressing NAMPT (LLC OE NAMPT) LLC cell

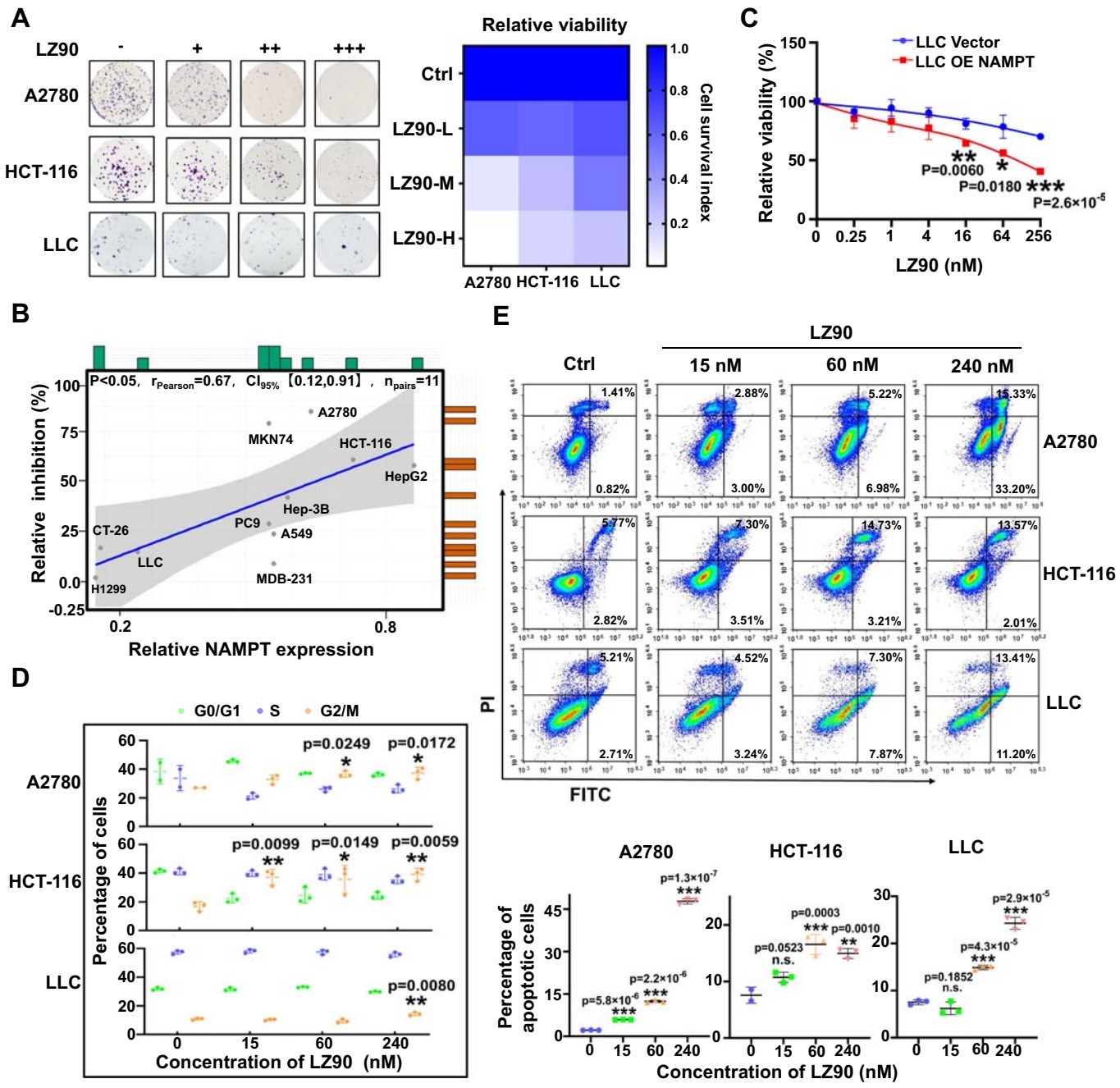

**Figure 4. LZFPN-90 inhibits the proliferation of cancer cells and promotes their apoptosis.**

(A) LZFPN-90 (hereafter referred to as LZ90) inhibits the formation of colonies by A2780 cells (0.5, 1, 2 nM), HCT-116 cells (0.5, 1, 2 nM) and LLC cells (20, 40, 80 nM) for 9 days. (B) Correlation analysis between the inhibitory activity of LZ90 for 48 h and the protein expression of NAMPT in 11 cell lines. (C) The inhibitory activity of compound LZ90 for 72 h on the catalytic activity of NAMPT in control LLC cells (LLC Vector) and LLC cells overexpressing NAMPT (LLC OE NAMPT), $n = 3$ per group. (D) Effect of LZ90 on the cell cycle in A2780, HCT-116, and LLC cells ($n = 3$ per group). (E) Apoptotic effect of LZ90 on A2780, HCT-116, and LLC cells ($n = 3$ per group). Data information: Data represent different numbers ($n$) of biological replicates. Data are shown as mean ± SEM. One-way ANOVA followed by Bonferroni's test is used in (D, E). Two-tailed Student's $t$ test is used in (C). *$P < 0.05$, **$P < 0.01$, ***$P < 0.001$ compared with control. Source data are available online for this figure.

lines. Compared with control cells (LLC Vector), LLC OE NAMPT cells were more sensitive to LZFPN-90 and FK866 (Fig. 4C; Appendix Fig. S3D), further indicating that LZFPN-90 affects tumor cell survival by targeting NAMPT. It has been reported that decreasing $NAD^+$ content causes cell cycle arrest by

inhibiting SIRT activity (Gerner et al, 2018). Thus, we examined the changes in the cell cycle after the administration of different concentrations of LZFPN-90. Our results showed that LZFPN-90 arrested the cell cycle in the G2/M phase (Fig. 4D; Appendix Fig. S3E) and subsequently induced apoptosis in a concentration-

dependent manner (Fig. 4E). Moreover, we found that compared with LLC parental cells, LZFPN-90-induced apoptosis was partially reduced in NAMPT knockdown LLC cells (Appendix Fig. S3F), which further confirming the crucial role of NAMPT in LZFPN-90-induced apoptosis. Taken together, our results suggested that the dual-targeting molecule LZFPN-90 could inhibit cancer cell proliferation, arrest the cell cycle in the G2/M phase, and promote apoptosis by inhibiting NAMPT activity.

## LZFPN-90 enhancing the function of T cells by suppressing PD-L1

We next examined the effect of LZFPN-90 on the immune system, specifically the ability of LZFPN-90 to suppress tumor immune escape through PD-L1 inhibition. We used a co-culture system to test the cytotoxic effect of T cells on tumor cells after treatment with LZFPN-90 and the PD-L1 inhibitor, BMS202. The results showed that in four cell lines (A2780, HCT-116, LLC, CT-26), both BMS202 and LZFPN-90 promoted T-cell killing capacity by tumor cells. At the same concentration, LZFPN-90 was more effective than BMS202 (Fig. 5A). As the killing effect of T cells depends on their proliferation and activation, we examined the effects of LZFPN-90 on the activation of $CD3^+$, $CD4^+$ and $CD8^+$ T cells in a co-culture system with LLC cells. The results showed that LZFPN-90 effectively promoted T-cell proliferation (Fig. 5B) and activated $CD8^+$ T cells (Fig. 5C). Simultaneously, we examined the changes in functional factors secreted by $CD8^+$ T cells and found that LZFPN-90 increased the levels of IFN-γ and GzmB secreted by $CD8^+$ T cells (Fig. 5D; Appendix Fig. S4A). These results indicated that LZFPN-90 can activate $CD8^+$ T cells and increase the secretion of functional factors, thereby contributing to the killing of tumor cells. Next, we verified the dependence of LZFPN-90 on the PD-L1 target. In the co-culture system, BMS202 and LZFPN-90 had a more significant killing effect on LLC cells overexpressing PD-L1 (LLC OE PD-L1) than on control cells (LLC Vector) (Fig. 5E). On the contrary, knockdown of PD-L1 in LLC cells would contribute to killing effect weaken of LZFPN-90 and PD-L1 inhibitors (Appendix Fig. S4B,C). This suggested that LZFPN-90 enables tumor cells to be killed by T cells by targeted inhibition of PD-L1. We next used the CFSE assay to investigate the effect of LZFPN-90 on T-cell proliferation when T cells were co-cultured with LLC and LLC OE PD-L1 cells. CFSE data indicated that LZFPN-90 induced a stronger increase in T-cell proliferation in the presence of LLC OE PD-L1 cells than in the presence of LLC cells (Fig. 5F). This suggested that the effect of LZFPN-90 on promoting T-cell proliferation is dependent on PD-L1 levels. In summary, our results showed that LZFPN-90 could promote T-cell proliferation, activate $CD8^+$ toxic T cells, and enhance the secretion of IFN-γ and GzmB, thereby killing tumor cells.

## LZFPN-90 retarding tumor growth in xenograft mouse and organoid models

To evaluate the pharmacodynamics of LZFPN-90 in vivo, we selected the A20 cell line to construct a tumor-bearing mouse model, which was because the A20 cell line has high PD-L1 expression and a relatively high $NAD^+$ content (Appendix Fig. S5A). Both FK866 and CQ-85, an in vivo-feasible PD-L1 inhibitor, effectively inhibited tumor growth in mice at the same

dose (20 mg/kg), whereas combined treatment with both inhibitors showed a stronger antitumor effect (Fig. 6A,B; Appendix Fig. S5B,C). As expected, dual-target inhibitor LZFPN-90 administration could obviously retard tumor growth in a dose-dependent manner, with the tumor inhibition rates (TIR) reaching 73.76% (20 mg/kg) and 83.11% (40 mg/kg) (Fig. 6B). In addition, we also found that LZFPN-90 had no obvious effect on mouse organ or body weight (Appendix Fig. S5D,E). In contrast, FK866 treatment caused a reduction in the body weight of the mice (Appendix Fig. S5D,E), which is related to the reported toxicity of FK866 (Nahimana et al, 2009). To further verify the effect of the LZFPN-90-dependent immune environment, we examined the effect of different treatments on $CD4^+$ and $CD8^+$ T cells in tumor tissue. The results showed that $CD8^+$ and $CD4^+$ T cells were activated more effectively by LZFPN-90 and combined treatment groups than CQ-85 and FK866 single treatment groups (Fig. 6C; Appendix Fig. S6A). Simultaneously, we found that LZFPN-90 promoted the secretion of CD69 and IFN-γ more effectively than single and combined inhibitor treatment groups (Fig. 6D; Appendix Fig. S6A). In addition, our results indicated that myeloid-derived suppressor cell (MDSC) content in tumors was suppressed after LZFPN-90 treatment (Appendix Fig. S6B). Importantly, our results revealed that the TIR of LZFPN-90 in Balb/c mice was 73.8%, which of LZFPN-90 in Balb/c nude mice was 46.2% (Fig. 6E). This suggests that the antitumor efficacy of LZFPN-90 is partially dependent on the immune system. In addition, TUNEL and immunohistochemistry results showed that LZFPN-90 (20 mg/kg) promoted cell apoptosis and inhibited cell proliferation compared to the control group (Fig. 6F). Next, to evaluate the application potential in the clinic, we evaluated the antitumor efficacy of LZFPN-90 using lung cancer organoids derived from lung cancer patients. After the treatment of LZFPN-90 for 9 days, compared with the control group, the survival of the organoids was decreased significantly (Fig. 6G; Appendix Fig. S6C). Then, we co-cultured the organoids with PBMC and observed a beginning PBMC infiltration and later killed the process in the LZFPN-90-treated group (Appendix Fig. S6D). In addition, CFSE staining data indicated the number of PBMC cells penetrating was significantly increased than the control group (Fig. 6H). In conclusion, our results showed that LZFPN-90 effectively inhibited tumor growth, increased lymphocyte infiltration and secretion of functional factors, and induced tumor cell apoptosis.

## LZFPN-90 reducing the tumor burden in mice bearing tumors derived from genetically altered LLC cells with NAMPT/PD-L1 overexpression

Based on clinical data, we demonstrated a negative correlation between the expression of NAMPT and PD-L1 expression in various cancer cell lines, and NAMPT and PD-L1 are co-expressed in various human cancers. To mimic this expression pattern, we constructed a tumor model in which mice were inoculated with LLC cells overexpressing both NAMPT and PD-L1 (Fig. 7A; Appendix Fig. S7A). Tumor-bearing mice were then treated with FK866, CQ-85, and LZFPN-90. Our data indicated that LZFPN-90 treatment resulted in an obvious reduction in tumor volume in the LLC model, with a TIR of 76.01%, which was higher than that of FK866 (43.62%) and CQ-85 (45.36%) (Fig. 7B). In addition, LZFPN-90 had a minor effect on the organs of mice (Appendix Fig. S7B). In

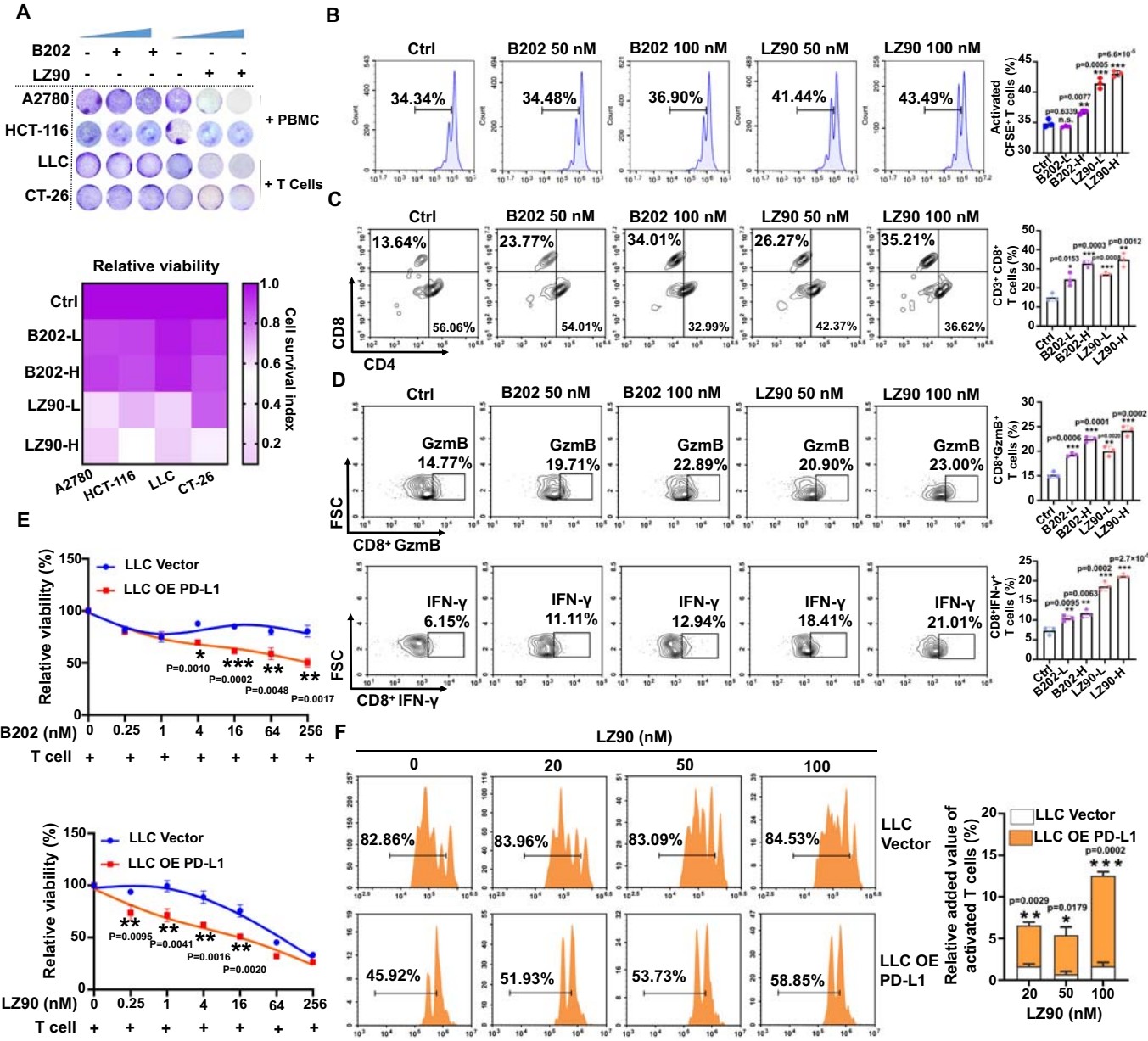

**Figure 5. Antitumor immunity of LZFPN-90 in vitro.**

(A) T-cell-mediated tumor cell killing assay. T cells were co-cultured with A2780, HCT-116, LLC, and CT-26 cells in the presence of increasing concentrations of LZ90 or BMS202 for 72 h. Surviving tumor cells were stained by crystal violet. (B) LLC cells were co-cultured with T cells. CFSE staining was used to detect the effect of LZFPN-90 (LZ90) and BMS202 (B202) on T-cell proliferation ($n = 3$ per group). (C, D) The proportion and representative staining of CD4$^+$ and CD8$^+$ cells (C) ($n = 3$ per group), and IFNγ$^+$ and Gzmb$^+$ CD8$^+$ T cells (D) ($n = 3$ per group) after co-culture with LLC cells. (E) LLC Vector cells or LLC OE PD-L1 cells were co-cultured with T cells in the presence of B202 or LZ90, and then CCK-8 was used to determine the viability of the LLC cells ($n = 3$ per group). (F) CFSE staining was used to detect the effect of LZ90 in LLC Vector and LLC OE PD-L1 cells (left). The relative added value of activated T cells is shown at the right ($n = 3$ per group). Data information: Data represent different numbers ($n$) of biological replicates. Data are shown as mean ± SEM. One-way ANOVA followed by Bonferroni's test is used in (B–D). Two-tailed Student's $t$ test is used in (E, F). *$P < 0.05$, **$P < 0.01$, ***$P < 0.001$ compared with control. n.s. not significant. Source data are available online for this figure.

parallel, analysis of tumor-infiltrating lymphocytes (TILs) showed that LZFPN-90 effectively increased lymphocyte infiltration, while inhibiting the infiltration of MDSCs (Fig. 7C; Appendix Fig. S7C). LZFPN-90 also increased the population of CD45$^+$, CD4$^+$, CD8$^+$, and CD69$^+$ cells, and the levels of GzmB, IFN-γ, and IL-2 in CD8$^+$ T cells (Fig. 7D; Appendix Fig. S7D,E). In addition, LZFPN-90 enhanced the immunohistochemical staining intensities of CD45$^+$,

CD8$^+$, and TUNEL, while reducing the proportion of Ki67-positive cells (Fig. 7E,F). These data confirmed that LZFPN-90 can expand and revive antitumor immunity, effectively promote cell apoptosis, and inhibit cell proliferation. We also measured the relative tumor inhibition rate of LZFPN-90 in the LLC OE NAMPT/PD-L1 and LLC Vector tumor models. We found that the TIR in the LLC OE NAMPT/PD-L1 group was 76.0% compared to 36.0% in the LLC

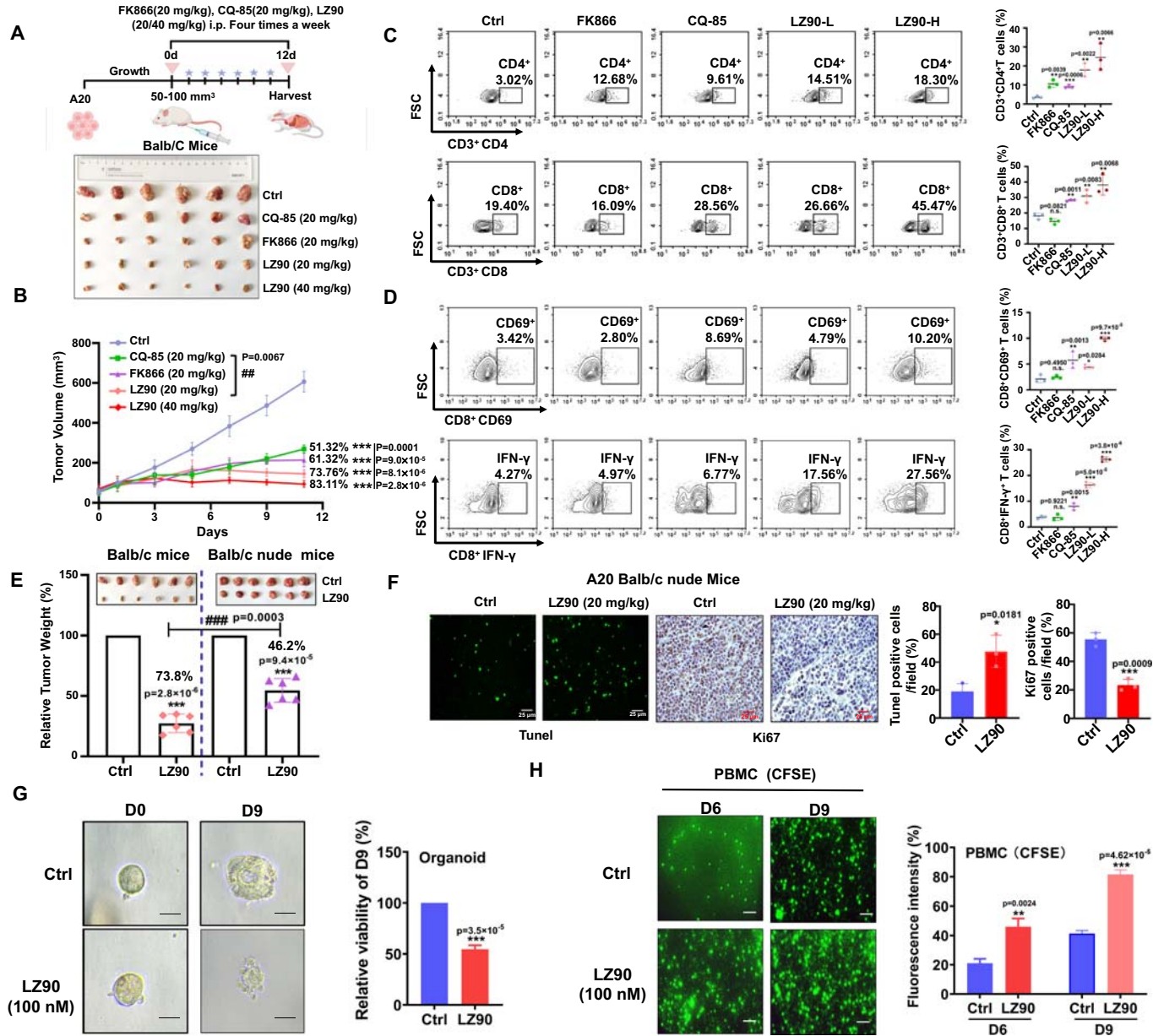

**Figure 6. LZFPN-90 inhibits tumor growth in double-target overexpressed xenograft models and organoids.**

(A, B) Tumor images at endpoint (A) and tumor growth curves (B). (B) The percent represents the tumor growth inhibition (TGI) rate at the endpoint of (A). A20 tumor-bearing Balb/c mice were treated with CQ-85 (20 mg/kg, i.p.), FK866 (20 mg/kg, i.p.), LZ90 (20 mg/kg, 40 mg/kg, i.p.), $n = 6$ mice/group. (C, D) The proportion and representative staining of tumor-infiltrating CD4$^+$ and CD8$^+$ T cells (C) ($n = 3$ mice/group) and the proportion and the representative staining of CD69$^+$ and IFN-γ among tumor-infiltrating CD8$^+$ T cells (D) ($n = 3$ mice/group). (E) Relative tumor weight of Balb/c mice and Balb/c nude mice treated with LZ90 (20 mg/kg, i.p.), $n = 6$ mice/group, as compared to the A20 Balb/c mice. For a more intuitive data comparison, the tumor images of Balb/c mice in the Ctrl group and the LZ90 group were derived from (A). (F) The percentage of tumor cells are positive for Tunel staining and Ki67 in Balb/c null mice treated with LZ90 (20 mg/kg, i.p.) by immunohistochemistry ($n = 3$ mice/group). (G) Representative bright-field images of the lung cancer organoid diameter treatment with 100 nM LZ90 for 9 days (scale bar 100 μm) ($n = 3$ per group). The representative picture is reused in Appendix S6C, which shows the complete process of organoid growth. (H) Representative fluorescent images of PBMC infiltration during co-culture treatment with 100 nM LZ90 for 6 and 9 days (scale bar 25 μm) (right), $n = 3$ per group. Data information: Data represent different numbers ($n$) of biological replicates. Data are shown as mean ± SEM. One-way ANOVA followed by Bonferroni's test is used in (B–D). Two-tailed Student's $t$ test is used in (E–H). *$P < 0.05$, **$P < 0.01$, ***$P < 0.001$ compared with control. ##$P < 0.01$ compared CQ-85 group (20 mg/kg) with LZ90 group (20 mg/kg), ###$P < 0.001$ compared LZ90 group (Balb/c mice) with LZ90 group (Balb/c nude mice), n.s. not significant. Source data are available online for this figure.

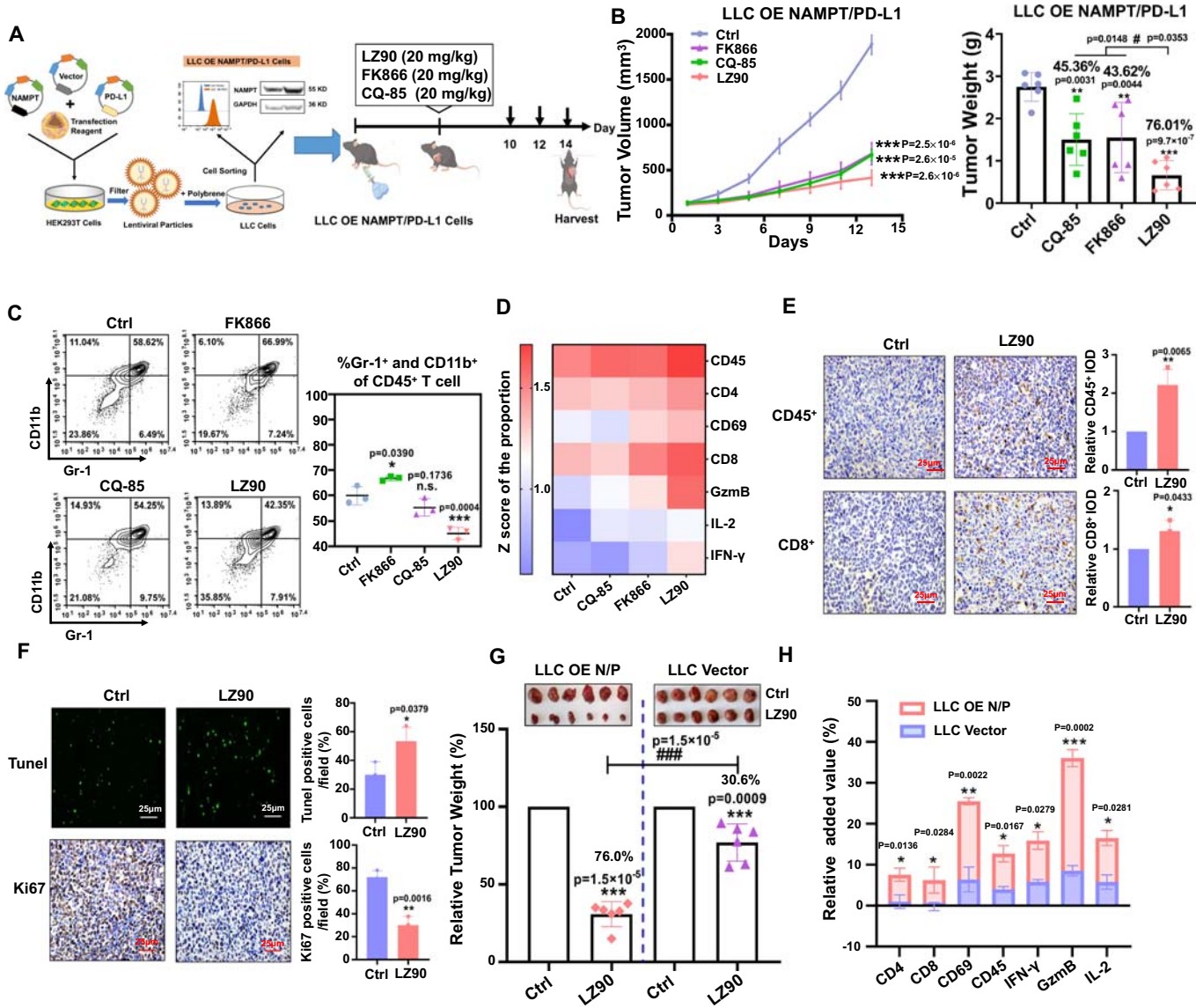

**Figure 7. LZFPN-90 exerts antitumor effects dependent on NAMPT and PD-L1 in the LLC OE NAMPT/PD-L1 mouse model.**

(A) Schematic showing the construction of the LLC OE NAMPT/PD-L1 (LLC OE N/P) mouse model and subsequent drug administration. (B–D) Tumor growth curve and weight (B) (*n* = 6 mice/group) and the infiltration of MDSCs (C) (*n* = 3 mice/group) and heatmap of tumor infiltration of CD45⁺, CD4⁺, CD69⁺, CD8⁺, GzmB⁺, IL-2⁺, IFN-γ⁺ (D). (B) The percent represents the tumor growth inhibition (TGI) rate at the endpoint of the experiment. LLC OE NAMPT/PD-L1 tumor-bearing C57BL/6 mice were treated with Ctrl, CQ-85 (20 mg/kg, i.p.), FK866 (20 mg/kg, i.p.), and LZ90 (20 mg/kg, i.p.). (E) Immunohistochemical detection of CD45⁺ and CD8⁺ T cells in the LLC OE N/P model (*n* = 3 mice/group). (F) The percentage of tumor cells positive for Tunel staining and Ki67 in the LLC OE N/P model treated with or without LZ90 (20 mg/kg, i.p.), which were detected by immunohistochemistry (*n* = 3 mice/group). Scale bar: 25 μm. (G) Relative tumor weight in the LLC Vector and LLC OE N/P models treated with or without LZ90 (20 mg/kg, i.p.), *n* = 6 mice/group. (H) The tumor infiltration in the LLC Vector and LLC OE N/P models treated with LZ90 (20 mg/kg, i.p.) were detected by flow cytometry, including CD45⁺, CD4⁺, CD69⁺, CD8⁺, GzmB⁺, IL-2⁺, IFN-γ⁺, *n* = 3 mice/group. Data information: Data represent different numbers (*n*) of biological replicates. Data are shown as mean ± SEM. One-way ANOVA followed by Bonferroni's test is used in (B, C). Two-tailed Student's *t* test is used in (E–H). \**P* < 0.05, \*\**P* < 0.01, \*\*\**P* < 0.001 compared with control; #*P* < 0.05 compared with LZ90 group, ###*P* < 0.001 compared with LLC Vector group. n.s. not significant. Source data are available online for this figure.

Vector group (Fig. 7G). This suggested that the activity of LZFPN-90 is dependent on the presence of its targets. Next, we analyzed the effect of LZFPN-90 on lymphocyte infiltration in tumors from the LLC OE NAMPT/PD-L1 and Vector groups. Compared with tumors from the LLC Vector group, LZFPN-90 induced a stronger immune response in tumors from the LLC OE NAMPT/PD-L1

group, by detecting the populations of CD4⁺, CD8⁺, CD69⁺, and CD45⁺ cells and the contents of IFN-γ, GzmB, and IL-2 in CD8⁺ T cells (Fig. 7H). These observations indicated that LZFPN-90 revives antitumor immunity by targeting PD-L1. Altogether, these results suggested that LZFPN-90 can suppress tumor growth and increase T-cell infiltration, which depends on NAMPT and PD-L1.

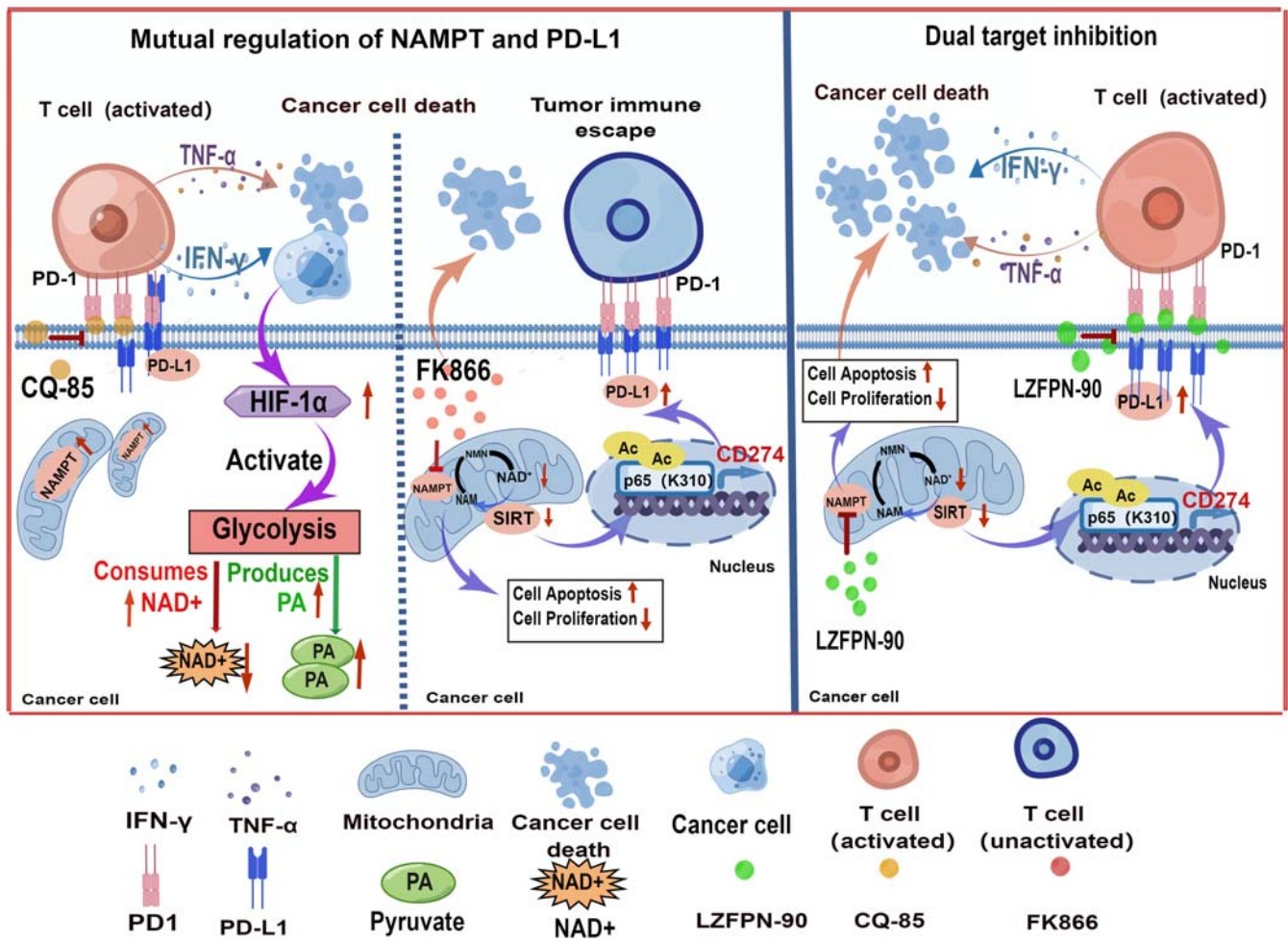

**Figure 8.  Schematic showing the proposed interaction of NAMPT and PD-L1 and their dual targeting by LZFPN-90.**

(Left) Inhibition of PD-1/PD-L1 interactions induces NAMPT expression through the HIF-1α/glycolysis pathway. (Middle) Inhibition of NAMPT promotes PD-L1 expression by regulating SIRT epigenetics. (Right) LZFPN-90 hinders PD-1/PD-L1 interactions and NAMPT activity and affects multiple malignant phenotypes.

## Discussion

In the last two decades, "immunometabolism" has emerged as a continuously increasing area of research with a significant impact on tumorigenesis (Deretic, 2021). Our study and other recent studies have demonstrated a reciprocal effect between tumor metabolism and immunity (Elia et al, 2022). In this study, we demonstrated that NAD+ metabolism plays a critical role in T-cell-mediated antitumor immunity. Our results reveal that NAMPT inhibition upregulates PD-L1 expression via SIRT epigenetic regulation; on the other hand, PD-L1 inhibition induces NAMPT expression depending on the co-culture conditions by cytokines-regulated glycolysis pathway. Our research showed the antitumor potential of the simultaneous targeting of NAMPT and PD-L1. Moreover, we designed and synthesized a dual-targeting compound, LZFPN-90, which effectively inhibited PD-1/PD-L1 interactions and NAMPT activity and affected multiple malignant phenotypes (Fig. 8). Therefore, our findings provided insight into the significance of tumor therapy by targeting tumor metabolism and immunity.

The regulation of PD-L1 expression is a complex and highly coordinated series of molecular signaling events. In tumor cells, PD-L1 expression may be related to metabolism, which in turn affects the immune status of the TME and even antitumor immunity. Studies have shown that tumors stabilized PD-L1 expression to control glucose metabolism by enhancing N-glycosylation and the EGFR/ERK/c-Jun pathways (Li et al, 2016; Yu et al, 2021). In addition, several studies have revealed that PKM2, which converted phosphoenolpyruvate into pyruvate, was highly expressed in tumors (Christofk et al, 2008; Elf and Chen, 2014) and enhanced PD-L1 expression in both tumor and immune cells via epigenetic mechanisms (Palsson-McDermott et al, 2017). Furthermore, Wang et al reported that NAD+ depletion and autophagy induced by NAMPT inhibitors mediate the upregulation of PD-L1 transcripts and cell surface protein levels in glioblastoma cells (Wang et al, 2019). A recent study demonstrated study revealed that PD-L1 expression also could be regulated by IFN-γ driven by NAMPT in a CD8+ T-cell-dependent manner, suggesting the possible role of TME in this process (Lv et al, 2021). Our study found the same phenomenon: NAMPT inhibition would result in

the upregulation of PD-L1 in vitro and in vivo. Mechanistically, our results reveal that NAMPT inhibition decreases SIRT activity, and raises acetylation level of the NF-κB p65-K310 site which is critical for NF-κB transcriptional activity (Yeung et al, 2004; Chen et al, 2002), and next contributes to the binding of NF-κB p65 on *CD274* (PD-L1) promoter region, subsequently results in the upregulation of PD-L1. Our results suggested that NAMPT regulated PD-L1 via SIRT, providing an epigenetic perspective to explain the possible regulatory mechanism of PD-L1. Moreover, our study revealed another phenomenon, namely that in a microenvironment with limited NAD$^+$ metabolism, which can decrease antitumor immunity through upregulation of PD-L1 in cancer cells. Our findings explained why targeting NAD$^+$ metabolism alone has limited efficacy in preclinical and clinical studies (Sampath et al, 2015) and suggested a new feasible strategy.

It has been recognized that metabolism and the immune is interact with each other during cancer development and progression (Liu et al, 2022). Here, we found that inhibition of PD-L1 also could induce the expression of NAMPT in the presence of T cells. To be specific, inhibition of the PD-1/PD-L1 axis could activate the T cells immunity, promoting the release of IFN-γ, TNF-α, and other inflammatory factors, and upregulating the expression of HIF-1α under the normal oxygen state. The upregulation of HIF-1α will increase the expression of downstream glycolytic-related genes and the content of pyruvate, the product of glycolysis, thus promoting the consumption of NAD$^+$ and inducing the expression of NAMPT to provide the high demand for NAD$^+$. The preliminary findings of our study indicate that the immune response exerts a certain influence on tumor metabolism, and HIF-1α plays a crucial role as a regulatory factor in this process.

Cancer is a disease caused by multiple factors, there are various treatment methods that are widely applied in the treatment of various cancers (Shen et al, 2020). Drugs with multiple targets, such as sorafenib, exert therapeutic effects on several signaling pathways (Sunay et al, 2017). Some drug combinations, such as EGFR inhibitors combined with anti-angiogenesis inhibitors can act on different malignant phenotypes (Matar et al, 2004). Therapeutic effects can also be achieved by combining interdependent molecules via synthetic lethality (Zatreanu et al, 2021). However, there are two main problems with drug therapies based on the targeting of specific molecules or signaling pathways. Cancers are heterogeneous; therefore, only some cancer cells may be dependent on specific oncogenes (Pe'er et al, 2021). Targeting one protein or signaling pathway often results in feedback activation of other oncogenic pathways. This leads to tumor cell adaptation, drug resistance, and disease recurrence (Jiang et al, 2021; Wu et al, 2020). Recently, a pioneering study in a melanoma mouse model showed that BRAF-mutated tumors are resistant to BRAF inhibitors, mainly due to the aberrant activation of cancer-associated fibroblasts (CAFs). In turn, CAF activation elicits BRAF downstream signaling in resistant cancer cells and promotes cell survival (Liu et al, 2022). These observations suggest that feedback regulation in the tumor microenvironment may be another crucial mechanism mediating drug resistance. In our study, we found reciprocal feedback regulation between NAMPT and PD-L1, which may lead to resistance to PD-L1 or NAMPT therapy. Based on these findings, we designed and identified a dual-targeting compound, LZFPN-90. On the one hand, by targeting PD-L1, it may improve the tumor immune microenvironment. On the other hand, by targeting NAMPT, LZFPN-90 can also inhibit cell growth, block the cell cycle, and subsequently induce apoptosis. Therefore, our study demonstrated the feasibility of co-targeting tumor metabolism and the tumor immune environment, which paved the way for the treatment of multiple cancers.

With the in-depth research on NAMPT, it has been found that many strategy target NAMPT achieve potent efficacy against tumor. Bi K et al reported targeting NAMPT by PROTAC technology gained potent tumor growth inhibition and demonstrated good biosafety (Bi et al, 2023). Furthermore, several other studies showed that co-targeting strategy, including NAMPT-HDAC (Dong et al, 2017; Yue et al, 2024) and NAMPT-IDO1 (Wang et al, 2023), also displayed encouraging effects in inhibiting the growth of tumors or overcoming drug resistance. Here, based on the interacted relationship between NAMPT and PD-L1, we developed a co-targeting strategy from NAMPT and PD-L1. Compared with other NAMPT-based target inhibitors, dual-targeting NAMPT and PD-L1 inhibitor has the priority effect of activating the immune system. In addition, the reciprocal regulatory relationship between NAMPT and PD-L1 makes the inhibitor have an enhanced effect on both targets, which is not reported by other dual-targeted strategy. Taken together, NAMPT-based antitumor strategy might own a promising future, especially in tumor immune environment regulation.

In conclusion, our research uncovered a negative feedback signaling between NAMPT and PD-L1. NAMPT inhibition upregulated PD-L1 expression, whereas blocking PD-L1 induced NAMPT expression, both in vitro and in vivo. Our dual-targeting chemical, which effectively inhibited PD-1/PD-L1 interactions, reduced NAMPT activity and affected multiple malignant phenotypes, providing a new strategy for multi-target therapy.

## Methods

### Mice

Male nude mice aged 6–8 weeks were purchased from Beijing Huafukang Biotechnology, and male Balb/c and C57BL/6 mice aged 6–8 weeks were purchased from Changsheng Biotechnology. All mice were SPF-grade mice and were housed in an isolated space without pathogens. All animal experiments were performed in strict accordance with the guidelines for the management and use of laboratory animals. All animal protocols were reviewed and approved by the Institutional Animal Care and Use Committee of the Laboratory Animal Center of Shenyang Pharmaceutical University (SYPU-IACUC-S2023-0629-102).

### Cell lines and culture

Mouse colon cancer cell line CT-26 (ATCC CRL-2638), mouse Lewis lung cancer cell line (LLC, ATCC CRL-1642), mouse lymphatic cancer cell line A20 (TIB-208), human ovarian cancer cell line A2780(ECACC; 93112519), and human colon cancer cell line HCT-116 (ATCC CCL-247), human non-small cell lung cancer cell line A549 (ATCC CCL-185), H1299 (ATCC CRL-5803), and PC9 (RRID: CVCL-B206), human gastric cancer cell line MKN74 (RRID: CVCL-2791), human liver cancer cell line HepG2(ATCC HB-8065) and Hep3B(ATCC HB-8064), human breast cancer cell

line MDA-MB-231(ATCC HTB-26), were purchased from the ATCC (Manassas, VA, USA). CT-26, A20, HCT-116, A549, H1299, PC9, MKN74, and MDA-MB-231 cell lines were cultured in RPMI-1640 medium (Gibco, USA) containing 10% fetal bovine serum (Gibco, USA), supplemented with 10% fetal bovine serum (Gibco, USA) and 1% penicillin–streptomycin (Gibco, Waltham, MA, USA) at 37 °C in a 5% $CO_2$ incubator. Every 3 months, the experimental cells were stained with a DNA staining reagent (Hoechst 33258) to observe if there was mycoplasma contamination. The cell lines were recently authenticated. Mycoplasma contamination was not detected in all experimental cells.

## In vitro PD-1/PD-L1-binding assay

The biochemical activity of the final compounds was evaluated using a PD-1/PD-L1 binding assay kit (Cisbio, cat. No. 64ICP01PEG). Experiments were performed according to the manufacturer's instructions.

## In vitro NAMPT activity assay

NAMPT enzymatic activity was determined by a NAMPT Activity Assay Kit (catalog no: # ab221819, Abcam) following the manufacturer's protocols. The assay is based on a multistep reaction that converts WST-1 to WST-1 formazan, which can be easily detected at an absorbance of 450 nm using a Tecan Infinite M1000 microplate reader (Jin et al, 2022). The fluorescent intensity data were analyzed using the computer software Graphpad Prism.

## Creation of LLC overexpression lines

The target plasmids with NAMPT overexpression, PD-L1 overexpression, and Vector were packaged into lentivirus and added to 293T cells to generate the virus for 48 h. The virus solution was collected and added to LLC cells. Cells successfully overexpressing NAMPT and Vector were selected using puromycin (MCE, USA), and LLC cells successfully overexpressing PD-L1 were selected by the addition of G418 (MCE, USA). LLC OE NAMPT/PD-L1 cells were constructed by transfection of LLC OE PD-L1 cells with lentivirus overexpressing NAMPT.

## Cell viability assay

In vitro cell viability was determined using a Cell Counting Kit-8 (MCE) assay. Cells ($5 \times 10^3$/well) were seeded in 96-well culture plates. After 24 h, they were incubated with various concentrations of the test compounds for 48–72 h at 37 °C in a 5% $CO_2$ incubator. Next, 10% of the CCK-8 solution was added to each well, and the plates were incubated for an additional 4 h at 37 °C. The optical density of each well was measured at 450 nm using a multimode plate reader (Molecular Devices, San Jose, CA, USA).

## MTT assay

Cells ($5 \times 10^3$/well) were seeded into a 96-well plate and cultured at 37 °C in a 5% $CO_2$ incubator. The concentrations of FK866, and LZFPN-90 ranged from 0.25 nM to 256 nM for 48 h, with a minimum of three technical replicates. At the final time point, 10 µl MTT solution (5 mg/ml, Sigma, Burlington, MA, USA) was added to each well and incubated for an additional 4 h at 37 °C in a 5% $CO_2$ incubator. Then,100 µl of DMSO was added to each well and mixed thoroughly. Optical density was measured at 492 nm using a microplate reader (Molecular Devices, San Jose, CA, USA).

## Clonogenic assay

Cells ($3 \times 10^2$/well) were plated in a six-well plate and the culture medium was replaced every 2 days. After culturing for 7–10 d, the cells were fixed with 4% paraformaldehyde for 15 min, stained with 0.1% crystal violet for 30 min, and imaged and counted.

## T-cell-mediated tumor cell killing assay

To acquire activated T cells, human peripheral blood mononuclear cells (PBMC; LTS1077, Yanjin Biological) were cultured in CTSTM AIIM VTM SFM (A3021002; Gibco) with ImmunoCult Human CD3/CD28/CD2 T-cell activator (25 µL/mL; STEMCELL Technologies, Cat#10970) for 48 h according to the manufacturer's protocol. The experiments were performed using mouse anti-CD3 antibody (1 µg/mL; Thermo Scientific, 16-0032-82) and anti-CD28 antibody(5 µg/mL; Thermo Scientific, 16-0281-82). Cancer cells were allowed to adhere to the plates overnight and then incubated for 48 h with activated T cells in the presence or absence of DX (50 mM). Different proportions of cancer cells and activated T cells were utilized according to the purpose of each experiment. The T-cell proliferation assay was 1:3; the T-cell killing assay was 1:10; T cells and cell debris were removed by PBS washing, and the living cancer cells were then quantified by a spectrometer at OD (570 nm) followed by crystal violet staining.

## Flow cytometry

For surface marker detection, tumor single-cell suspensions and spleen lymphocytes were incubated at 4 °C for 30 min in PBS with 0.1% Fixable Viability Dye (eBioscience 65-0863-14). Cells were washed and incubated in staining buffer (PBS with 1% FBS) with surface marker antibodies, the dilution ratio of all flow antibodies was 1:200 (BioLegend, CD3-APC 100235, CD3-PE 100205, CD4-FITC 100405, CD8-APC 100711, CD8-PE 100707, CD45-APC 103112, PD-L1-PE 124307, adsorbed secondary antibody-APC, and CD69-PE 104511). The cells were then washed and resuspended in staining buffer for flow cytometry. For cytokine detection, tumor single-cell suspensions and spleen lymphocytes were incubated in RPMI-1640 with 10% FBS and 0.2% Cell Stimulation Cocktail (plus protein transport inhibitors) (eBioscience 00-4975-93) at 37 °C for 12 h. Cells were stained with Fixable Viability Dye and surface marker antibody as described above and then were fixed and permeabilized using a BD Cytofix/Cytoperm Fixation/Permeabilization Kit (BioLegend, BD 554714). Cells were incubated at 4 °C for 30 min in 1× BD Perm/Wash buffer with cytokine antibody (eBioscience IFN-γ-APC 17-7311-81, BioLegend IL-2-APC 562041) and then washed in 1× BD Perm/Wash buffer two times. The cells were resuspended in staining buffer for flow cytometry measurements. Flow cytometry was performed on ACEA Novo-Cyte, and the data were analyzed using NovoExpress.

## T cells suppression study

Spleens were removed from 8-week female C57BL/6 wild-type mice and placed in a 70-µm cell sieve and gently ground until no obvious tissue

mass was observed. Then, 4–5 ml of lymphocyte separation fluid (Dakewe, Cat. No. 7211011, China) was added to resuspend the tissues. The cell suspension was placed in a 15-ml centrifuge tube for gradient centrifugation at 800 g for 30 min. The lymphocytes were purified from the liquid and stained with CFSE (Thermo, Cat. No. C34554, USA). CFSE-labeled lymphocytes were placed in 24-well plates supplemented with complete RPMI-1640 medium with 1 μg/ml anti-CD3 (Thermo Scientific, 16-0032-82) and 5 μg/ml anti-CD28 (Thermo Scientific, 16-0281-82) antibodies. Isolated tumor cells (1:10) were added to a 24-well plate. After incubation for 72 h, the T cells were collected and stained with anti-CD3, anti-CD4, and anti-CD8 antibodies (BioLegend, CD3-APC 100235, CD3-PE 100205, CD4-FITC 100405, CD8-APC 100711, CD8-PE 100707) for flow cytometric analysis, the antibody dilution ratio is 1:200.

## Apoptosis assay

The cells are treated with the compounds. After 48–72 h, the cells were harvested and washed twice with cold PBS. The cells were then stained with Annexin V/propidium iodide (PI) (BD Biosciences) and examined using a NovoCyte Flow Cytometer (ACEA Biosciences). Data were analyzed using NovoExpress software (ACEA Biosciences).

## Immunohistochemistry and TUNEL assay

Follow the instructions of the immunohistochemical kit (ZSGB-BIO, PV-9000). Formalin-fixed 5-μm thick serial tumor sections were deparaffinized in xylene and rehydrated in ethanol. Endogenous peroxidase was blocked with 3% $H_2O_2$ for 15 min. The samples were incubated with primary antibodies (1:200) and secondary antibodies (1:2000) and then reacted with DAB detection reagents (ZSGB-BIO, ZLI-9018, China). Antibodies against Ki67 (Thermo, MA5-14520,1:200), CD45 (Thermo, 14-0451-82,1:200), and Goat anti-human IgM (heavy chain) cross-adsorbed secondary antibodies (HRP) (Thermo, A18835,1:1000) were purchased from Thermo Fisher Scientific (USA). The TUNEL system (Roche, 11684795910) was used to detect apoptosis in tumor sections on slides, according to the manufacturer's protocol. The TUNEL reaction solution was replaced with a TdT-free solution to create a negative control. The sections were incubated with DNase for 10 min and visualized using DAB staining. Positive nuclei were identified based on the presence of a brown color. The percentage of positive cells among the total number of cells was calculated.

## Cellular thermal shift assay (CETSA)

The cells were resuspended in 1× PBS supplemented with protease inhibitors (Roche) and then divided into strip tubes for 30 min. The tubes were centrifuged at 15,000× g at 4 °C for 15 min. After centrifugation, the supernatant was divided into DMSO and LZFPN-90 (20 nM) groups, and incubated at room temperature for 30 min. According to standard procedures (38 °C, 42 °C, 46 °C, 50 °C, 54 °C, 58 °C, 62 °C, 66 °C), each temperature group was heated for 3 min, incubated at room temperature for 3 min. Then centrifuged at 20,000× g for 25 min, the supernatant was collected, 1× loading buffer was added, the mixture was heated in a metal bath at 95 °C for 10 min, and the amount of non-denatured protein per sample was quantified by western blotting.

## Western blotting

Protein lysates were isolated from cells or tumor sections using RIPA buffer (CST, Danvers, MA, USA). Protein separation was performed using electrophoresis in 10–15% arc-bis gels, and the proteins were transferred onto PVDF using a transfer system (Bio-Rad, Hercules, CA, USA). The membranes were incubated with the appropriate primary antibodies (1:1000) and secondary antibodies (Thermo, A11034, A32723 1:10000), reacted with ECL detection reagents (Thermo Fisher Scientific, Waltham, MA, USA), and incubated for several minutes in a dark room. Antibodies against PD-L1 (Cell Signaling Technology, 60475, 1:1000), NAMPT (Proteintech, MA5-24108, 1:1000), β-actin (Abcam, ab8226, 1:1000), GAPDH (Abcam, ab8245, 1:1000), Ac-tubulin (Abcam, ab24610, 1:1000), α-Tubulin (Abcam, ab7291, 1:1000), Ac-NF-κB p65 (K310) (Thermo, PA5-17264, 1:1000), NF-κB p65 (Thermo, PA5-23170, 1:1000), HIF-1α (Thermo, MA1-516, 1:1000) were purchased from commercial company.

## RNA isolation and quantitative real-time PCR (qRT-PCR)

Total RNA was isolated using TRIzol reagent (Invitrogen, 15596026) according to the manufacturer's protocol. First-strand cDNA synthesis was performed using a RevertAid First Strand cDNA Synthesis Kit (Thermo Fisher, K1622). Quantitative real-time PCR analysis was performed by mixing 10 μl SYBR, 0.4 μM forward primer, 0.4 μM reverse primer, 1 μl cDNA, and 8.2 μl distilled water per sample with Top Green qPCR SuperMix (Transgene, AQ601-04) by the manufacturer's protocols. GAPDH was used as an endogenous control. All qRT-PCRs were set up in triplicate using three biological replicates for each sample. Primer information is shown in Appendix Table S1.

## TCGA data analysis

The mRNA expression data of the 33 TCGA cancer samples were downloaded from the Gene Expression Database (https://portal.gdc.cancer.gov/). The mRNA expression levels of the PD-L1 and NAMPT genes in cancer tissues and adjacent tissues of the 33 cancer samples were analyzed, and their average values were obtained and homogenized according to the Z method. Kaplan–Meier analysis of the overall survival of patients with different mRNA levels of PD-L1 and NAMPT in pan-cancer. They were divided into four groups (low expression of PD-L1 and NAMPT, low expression of PD-L1 and high expression of NAMPT, high expression of PD-L1 and low expression of NAMPT, and high expression of PD-L1 and NAMPT) for statistical analyses.

## NAD$^+$ assay

NAD$^+$ assays were performed to quantify the total NAD$^+$ content. NAD$^+$ assay was performed using the NAD/NADH Assay Kit (ab65348; Abcam) according to the manufacturer's recommended protocol.

## Establishment and analysis of tumor-bearing mice xenograft models

### In mice bearing A20 tumors

The 4–6 weeks-old Balb/c mice and Balb/c nude mice were maintained in a specific pathogen-free (SPF) facility. A20 cells

$(5 \times 10^6$ cells in 0.2 ml phosphate-buffered saline) were subcutaneously injected into the right armpit of Balb/c mice and Balb/c nude mice. Tumor volume was measured using calipers every 2 days and calculated using the following formula: (long diameter) $\times$ (short diameter)$^2$/2. Balb/c mice were divided into groups when the tumor grew to 50–100 cm$^3$, in order to avoid the influence of human factors, the tumor volume was randomly grouped. then they were treated with the solvent group, CQ-85 group (20 mg/kg), FK866 group (20 mg/kg), and LZFPN-90 (20 mg/kg, 40 mg/kg). Balb/c nude mice were treated with LZFPN-90 (20 mg/kg). After 12 days of administration, the animals were anesthetized, and tumors were excised. Some fresh tumor tissues were subjected to immune cell isolation. Tissues were either fixed in 4% formalin or stored at −80 °C until further analysis.

### In mice bearing LLC tumors

The 4–6 weeks-old C57BL/6 mice were maintained in a specific pathogen-free (SPF) facility. LLC Vector and LLC OE NAMPT/PD-L1 cells $(2 \times 10^6$ cells in 0.2 ml phosphate-buffered saline) were subcutaneously injected into the right armpit of C57BL/6 mice. Tumor volume was measured using calipers every 2 days and calculated using the following formula: (long diameter)$\times$(short diameter)$^2$/2. C57BL/6 mice were divided into groups when the tumor grew to 50–100 cm$^3$ by random grouping. In mice bearing LLC OE NAMPT/PD-L1 tumors, were treated with solvent group, CQ-85 group (20 mg/kg), FK866 group (20 mg/kg), and LZFPN-90 (20 mg/kg). In mice bearing LLC Vector, tumors were treated with LZFPN-90 (20 mg/kg). After 14 days of administration, the animals were anesthetized, and tumors were excised. Some fresh tumor tissues were subjected to immune cell isolation. Tissues were either fixed in 4% formalin or stored at −80 °C until further analysis.

## Immunohistochemistry

For the A20 and LLC tumor xenografts, tumor tissues were isolated from mice, soaked in formalin, embedded, and sliced. The samples were dipped into 1× citrate buffer (Solarbio, China) for microwave antigen repair and then into peroxidase blockers to block endogenous peroxidase activity. The samples were incubated with goat serum, then with primary antibodies Ki67 (Thermo, MA5-14520, 1:200), CD8 (Thermo, 36-0081-85, 1:200), CD45 (Thermo,14-0451-82, 1:200) at 4 °C overnight. Normal mouse/rabbit/sheep IgG was used as the negative control to ensure specificity. Next, a biotinylated secondary antibody working solution and HRP-labeled streptavidin (Thermo, A18835, 1:1000) were added. Finally, DAB (ZSGB-bio company, ZLI-9018, China) was used for color rendering, and hematoxylin was used for re-staining to conduct image acquisition.

## Chromatin immunoprecipitation (ChIP) and ChIP-qPCR

Chromatin Extraction Kit (Abcam, ab117152, USA) and ChIP Kit Magnetic-One Step (Abcam, ab156907, USA) were used to perform ChIP. Experimental methods followed the instructions provided by the manufacturer. We used Chromatin Extraction Kit to extract the chromatin in A2780 cells, then enriched chromatin sequences containing Acetyl-NF-κB p65 (Lys310) (Thermo, PA5-17264) by ChIP Kit Magnetic-One Step, the ratio of antibody was 1:50. The purified DNA was analyzed by the same methods as the cDNA in

RT-qPCR to detect the enrichment of Acetyl-NF-κB p65 (Lys310) on the *CD274* promoter. The primers for ChIP-qPCR are provided in Appendix Table S1.

## Human specimens statement

According to National Comprehensive Cancer Network (NCCN) guidelines, after consulting the patients, those who consented to the use of their remaining biological samples for scientific research provided the signed informed consent form for biological sample analysis. We used such patients' remaining biological samples in our study. Before being conducted, this study was approved by the ethics committee of the Affiliated Cancer Hospital of Shengjing Hospital of China Medical University (IRB No. 2023PS1105K). All protocols adhered to the principles outlined in the Declaration of Helsinki and the Department of Health and Human Services Belmont Report.

## Patient-derived organoids culture and viability assay

Human lung cancer samples were obtained from Shengjing Hospital of China Medical University (IRB No. 2023PS1105K). The establishment of lung cancer patient-derived organoids was performed as described (Ganesh et al, 2019). Organoids were

---

**The paper explained**

**Problem**

Both NAMPT and PD-L1 are targets for tumor therapy. Unfortunately, the clinical activity of NAMPT inhibitors is limited, suggesting that the approaches for targeting NAD$^+$ production in tumors need to be refined. PD-L1 plays an important role in various malignancies. Nevertheless, the low response rate of α-PD-1/PD-L1 therapy remains to be resolved. In addition, the actual process underlying the interaction between cancer cell NAD$^+$ metabolism and immune response is not well understood.

**Results**

In our study, we found reciprocal feedback regulation between NAMPT and PD-L1: NAMPT inhibition upregulates PD-L1 expression via SIRT epigenetic regulation; on the other hand, PD-L1 inhibition induces NAMPT expression depending on the co-culture conditions by cytokines-regulated glycolysis pathway, which may lead to a reduction in the efficacy of PD-L1 or NAMPT single inhibitor therapy. Based on these findings, we designed and identified a dual-targeting compound, LZFPN-90, targeting PD-L1 may improve the tumor's immune microenvironment, and targeting NAMPT can inhibit cell growth, block the cell cycle, and subsequently induce apoptosis. Using mice bearing genetically manipulated tumors, we confirmed that LZFPN-90 exerted target-dependent antitumor activities, affecting metabolic processes and the immune system.

**Impact**

Increasing evidence suggests that cancer metabolism not only plays a crucial role in sustaining the growth and survival of tumor cells but also contributes to the regulation of antitumor immune responses by regulating the release of metabolites and the expression of immune molecules. Our findings provided insight into the significance of tumor therapy by targeting tumor metabolism and immunity, suggesting that co-targeting NAD$^+$ metabolism and PD-L1 represents a promising therapeutic approach.

seeded into 96-well plates and treated with the corresponding compounds for 9 days, and cell viability was determined by CellTiter Glo 3D cell viability assay (Promega, G9683). The trial did not involve the use of the blinding method.

### CFSE (carboxyfluorescein diacetate succinimidyl ester) staining

After extraction of PBMC, CFSE (NO.M5117, AbMole, USA) was dyed at room temperature for 5–10 min, adding PBS (Thermo, USA) to wash excess dye, 300 g was centrifuged for 5 min, the supernatant was discarded, and the wash steps were repeated twice. CFSE-stained PBMC was added to organoids, and cultured for 3, 6, and 9 days, and the infiltration of PBMC was observed by fluorescence microscopy.

### Statistical analysis

Statistical analyses were performed using GraphPad Prism 8.0. Data in all graphs are presented as the mean ± standard error of mean (SEM) of biological triplicates. Statistical significance was determined using the Student's $t$ test, Ordinary one-way ANOVA, or two-way ANOVA. For all statistical tests, a 0.05 level of confidence was accepted for statistical significance. The significance of the $P$ value was marked $*P < 0.05$, $**P < 0.01$, $***P < 0.001$ or 0.0001.

### For more information

TCGA data are publicly available at the Gene Expression Database (https://portal.gdc.cancer.gov/).

## Data availability

This study includes no data deposited in external repositories.

## Peer review information

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

ROS production in chronic cerebral hypoperfusion models through Sirt1/PGC-1α pathway. J Neuroinflammation 18:207

## Acknowledgements

This work was supported by the "Xingliao Talents" Program of Liaoning Province (XLYC1902008 and XLYC2007159), Natural Science Foundation of Shenyang (22-315-6-11), National Natural Science Foundation of China (22177080 and 22377082), 345 Talent Project of Shengjing Hospital of China Medical University (M1312), and Academy of Finland (317680 and 320131).Thanks to FigDraw for drawing support (Figdraw ID: RYPAI1a8b1; Figdraw ID: TPITYece00).

## Author contributions

**Yuan Yang**: Conceptualization; Data curation; Formal analysis; Supervision; Validation; Investigation; Visualization; Methodology; Writing—original draft; Project administration. **Zefei Li**: Data curation; Investigation; Writing—original draft. **Yidong Wang**: Data curation; Software; Formal analysis; Visualization; Methodology. **Jiwei Gao**: Data curation; Software; Investigation; Visualization; Methodology. **Yangyang Meng**: Data curation; Software; Formal analysis; Investigation; Methodology. **Simeng Wang**: Data curation; Formal analysis; Supervision; Visualization. **Xiaoyao Zhao**: Data curation; Software; Methodology. **Chengfang Tang**: Data curation; Validation; Methodology. **Weiming Yang**: Data curation; Formal analysis; Investigation. **Yingjia Li**: Data curation; Investigation. **Jie Bao**: Data curation; Supervision. **Xinyu Fan**: Formal analysis; Methodology. **Jing Tang**: Data curation; Supervision. **Jingyu Yang**: Supervision; Project administration. **Chunfu Wu**: Supervision; Validation. **Mingze Qin**: Resources; Data curation; Supervision; Funding acquisition; Methodology; Writing—review and editing. **Lihui Wang**: Conceptualization; Resources; Supervision; Funding acquisition; Investigation; Visualization; Project administration; Writing—review and editing.

## Disclosure and competing interests statement

The authors declare no competing interests.

