## [Peer Review File · EMBO Molecular Medicine]

The relationship between NAMPT and PD-L1 in cancer, and identification of a dual-targeting inhibitor

Yuan Yang, Zefei Li, Yidong Wang, Jiwei Gao, Yangyang Meng, Simeng Wang, Xiaoyao Zhao, Chengfang Tang, Weiming Yang, Yingjia Li, Jie Bao, Jing Tang, Xinyu Fan, Jingyu Yang, Chunfu Wu, Mingze Qin, and Lihui Wang

Corresponding authors: Lihui Wang (lhwang@syphu.edu.cn) , Mingze Qin (qinmingze@syphu.edu.cn)

Review Timeline:

Submission Date:	23rd Aug 23
Editorial Decision:	12th Sep 23
Revision Received:	3rd Jan 24
Editorial Decision:	31st Jan 24
Revision Received:	20th Feb 24
Accepted:	22nd Feb 24

Editor: Poonam Bheda

Transaction Report:

12th Sep 2023

Dear Prof. Wang,

Thank you for the submission of your manuscript to EMBO Molecular Medicine. We have now received feedback from the three reviewers who agreed to evaluate your manuscript. As you will see from the reports below, the referees acknowledge the interest of the study and are overall supporting publication of your work pending appropriate revisions.

Addressing the reviewers' concerns in full will be necessary for further considering the manuscript in our journal, and acceptance of the manuscript will entail a second round of review. EMBO Molecular Medicine encourages a single round of revision only and therefore, acceptance or rejection of the manuscript will depend on the completeness of your responses included in the next, final version of the manuscript. For this reason, and to save you from any frustrations in the end, I would strongly advise against returning an incomplete revision.

We are expecting your revised manuscript within three months, if you anticipate any delay, please contact us.

We require:

4) A .docx formatted letter INCLUDING the reviewers' reports and your detailed point-by-point responses to their comments. As part of the EMBO Press transparent editorial process, the point-by-point response is part of the Review Process File (RPF), which will be published alongside your paper.

5) A complete author checklist, which you can download from our author guidelines (<https://www.embopress.org/page/journal/17574684/authorguide#submissionofrevisions>). Please insert information in the checklist that is also reflected in the manuscript. The completed author checklist will also be part of the RPF.

6) Please note that all corresponding authors are required to supply an ORCID ID for their name upon submission of a revised manuscript.

7) It is mandatory to include a 'Data Availability' section after the Materials and Methods. Before submitting your revision, primary datasets produced in this study need to be deposited in an appropriate public database, and the accession numbers and database listed under 'Data Availability'. Please remember to provide a reviewer password if the datasets are not yet public (see <https://www.embopress.org/page/journal/17574684/authorguide#dataavailability>).

In case you have no data that requires deposition in a public database, please state so in this section. Note that the Data Availability Section is restricted to new primary data that are part of this study. This study includes no data deposited in external repositories.

8) For data quantification: please specify the name of the statistical test used to generate error bars and P values, the number (n) of independent experiments (specify technical or biological replicates) underlying each data point and the test used to calculate p-values in each figure legend. The figure legends should contain a basic description of n, P and the test applied. Graphs must include a description of the bars and the error bars (s.d., s.e.m.). Please provide exact p values.

9) Our journal encourages inclusion of *data citations in the reference list* to directly cite datasets that were re-used and obtained from public databases. Data citations in the article text are distinct from normal bibliographical citations and should directly link to the database records from which the data can be accessed. In the main text, data citations are formatted as

follows: "Data ref: Smith et al, 2001" or "Data ref: NCBI Sequence Read Archive PRJNA342805, 2017". In the Reference list, data citations must be labeled with "[DATASET]". A data reference must provide the database name, accession number/identifiers and a resolvable link to the landing page from which the data can be accessed at the end of the reference. Further instructions are available at .

13) Author contributions: CRediT has replaced the traditional author contributions section because it offers a systematic machine readable author contributions format that allows for more effective research assessment. Please remove the Authors Contributions from the manuscript and use the free text boxes beneath each contributing author's name in our system to add specific details on the author's contribution. More information is available in our guide to authors.

Please also suggest a striking image or visual abstract to illustrate your article as a PNG file 550 px wide x 300-600 px high. Share synopsis text and image, as well as eTOC:

Please note that these would be the final versions and changes during proofing are usually not allowed

16) As part of the EMBO Publications transparent editorial process initiative (see our Editorial at <http://embomolmed.embopress.org/content/2/9/329>), EMBO Molecular Medicine will publish online a Review Process File (RPF) to accompany accepted manuscripts.

In the event of acceptance, this file will be published in conjunction with your paper and will include the anonymous referee reports, your point-by-point response and all pertinent correspondence relating to the manuscript. Let us know whether you agree with the publication of the RPF and as here, if you want to remove or not any figures from it prior to publication.

I look forward to receiving your revised manuscript.

Yours sincerely,

Poonam Bheda

Poonam Bheda, PhD
Scientific Editor
EMBO Molecular Medicine

**** Reviewer's comments ****

Referee #1 (Remarks for Author):

Cancer's complexity arises from its genetic diversity and metabolic patterns, which also influence immune cells. The coenzyme NAD⁺ and its key enzyme, NAMPT, play pivotal roles in tumor metabolism and immune modulation. With the advent of modern oncology, there has been a pronounced shift towards checkpoint inhibitor immunotherapy, which specifically targets the PD-1/PD-L1 interaction, aiming to amplify the immune system's ability to detect and combat tumor cells. This work explored the link between NAMPT and PD-L1, revealing their potential as therapeutic targets. However, upon meticulous review and assessment, we think that this work, while valuable, does not meet the rigorous standards for publication in EMBO Molecular Medicine. Specifically, the mechanistic investigations presented seem to be premature.

1. NAMPT and PD-L1 exhibit a negative correlation. Inhibiting NAMPT leads to an elevated expression of PD-L1, which enhances the immune evasion of tumor cells and subsequently weakens the effectiveness of immunotherapy. Given this interplay, the author's decision to design a dual-target inhibitor for both NAMPT and PD-L1 raises the question: does this strategy possess significant potential for medical application?
2. In section 3.1 on Analysis of the relationship between NAMPT and PD-L1 in cancer cells, the author believes that inhibiting NAMPT promotes the expression of PD-L1 through the epigenetic regulation of SIRT. However, the intricate interplay between NAMPT and PD-L1 remains somewhat elusive. In 2021, Wang et al. highlighted that NAMPT drives interferon γ (IFN γ)-induced PD-L1 expression in multiple types of tumors and governs tumor immune evasion in a CD8⁺ T cell-dependent manner (Cell Metabolism 2021 33, 110-127). The author might need to continue in-depth investigation into the mechanism.
3. The authors found that inhibition of PD-L1 by QC-85 in the presence of T cells can lead to the upregulation of NAMPT by affecting the tumor immune microenvironment. However, the experiment only considered the effect of CQ-85 on PD-L1, not excluding its influence on T cells. Similarly, the authors indicated that LZFPN-90 can enhance the functionality of T cells by inhibiting PD-L1, however, does not rule out the effect of LZFPN-90 on T cells. In addition, in this assay, the author used BMS202 as the control group. To exclude uncertainties and for better comparison, CQ-85 should be included in the assay.
4. The studies in the current paper are mainly carried out through inhibiting the biological targets (NAMPT and PD-L1). The knockout or knockdown studies are required for further comparison and verification.
5. Besides, in the animal study, dual inhibitor LZFPN-90 should also be compared with the co-treatment (FK866 + CQ-85) group.

Referee #2 (Remarks for Author):

In the present manuscript, Yang et al. present a compelling therapeutic approach with broad applicability to various types of cancer. Using both in vitro and in vivo models, the authors demonstrate that simultaneous targeting of NAMPT and PD-L1 yields a more potent antitumor response compared to targeting each protein independently. A significant highlight of this study is the synthesis of the first dual NAMPT/PD-L1 targeting small molecule "LZFPN-90". This molecule is shown to effectively inhibit cancer cell proliferation, arrest the cell cycle, induce apoptosis, and substantially reduce tumor growth in xenograft models across multiple cancer types. The study is well-executed and elegantly presented. The authors employ a range of methodologies to substantiate their findings, which adds significant credibility to their work. Nevertheless, there are minor points that could be addressed to further consolidate the study's conclusions.

Minor comments

1. In the current study, the authors have used various mouse strains (nude, BALB/c, C57BL/6) to create xenograft models. However, details concerning model generation are lacking, making it difficult to ascertain specifics like the mouse strain used (in each reported experiment), the number of injected cells, pre-transplantation irradiation, drug administration timing and protocols, as well as dosage schedules. Providing this information is crucial for the experiment's reproducibility. What is the rationale for

housing BALB/c and C57BL/6 mice in SPF conditions?

2. The study presents intriguing results involving a dual-targeting inhibitor. Is the antitumor activity equally attributable to both NAMPT and PD-L1 inhibition? Additionally, can LZFPN-90 induce cell death when NMN, nicotinic acid, or alternative NAD precursors (NR, NRH, NAR) are present in tumor microenvironment?
3. In the graphical abstract, there's a typographical error: the dual-targeting inhibitor "LZFPN-90" is incorrectly spelled as "LZ-90."
4. Existing literature already discusses the modulation of PD-L1 by NAD metabolism. I recommend that the authors incorporate these works into their discussion for a more comprehensive analysis.
5. The "Cell lines and culture" section mentions only five cell lines, yet Figure 1C utilizes eleven. It would be beneficial to include information on the missing cell lines.
6. For LZFPN-90, as a novel compound, crucial details are absent, including synthesis methods, pharmacokinetics, pharmacodynamics, and toxicity data.
7. Concerning in-vitro studies, I would suggest that the authors offer more specific information on the experimental methodology, such as the duration of drug incubation.
8. To conclusively show that LZFPN-90 functions as a NAMPT inhibitor, NAMPT Activity Assays could provide direct evidence. I recommend conducting such an experiment for robust validation.
9. Figures 4B and 4C indicate that LZFPN-90 promotes T cell proliferation and CD8+ T cell activation. Could the authors explain these observations? Prior studies, such as Bruzzone et al. 2009 (PMID: 19936064), suggest that NAMPT inhibition has a detrimental effect on activated T cells. This observation is in line with one of the side effect "lymphopenia" of NAMPT inhibitors. lymphocytopenia is a consequence of NAMPT inhibition.
10. As tumor cell lines differ from primary cancer cells, I would recommend the authors to perform same in vitro studies, using primary tumor cells from patients, if possible

Referee #3 (Remarks for Author):

In this manuscript, the author investigated the reciprocal feedback regulation between NAMPT and PD-L1, and designed a dual-targeting inhibitor LZFPN-90 to disrupt PD-1/PD-L1 interactions and inhibit NAMPT activity, thereby enhancing T-cell mediated antitumor immunity. The observation of negative correlation between the expression of NAMPT and PD-L1 in various cancer cell lines is novel, and the dual inhibitor exhibited strong inhibition to both targets. However, one major issue, how SIRT epigenetic regulation can affect the expression of PD-L1, and Vice versa? Possible mechanisms are not investigated and discussed. In addition, there are a few other major issues need to be clarified or verified.

Major:

1. The author used FK866 and CQ-85 to inhibit NAMPT and PD-L1, respectively, and observed negative correlation between the expression of NAMPT and PD-L1. To exclude off target possibilities of these small molecular inhibitors, genetic knock down/out, or overexpression of the two genes individually, are needed to further confirm the correlation between the expression of NAMPT and PD-L1.
2. Fig. 2C, the relative band intensity plots of LZ90 do not fit into the curve at all!
3. In Figure 3C, compound LZ90 basically had no effect on the relative viability of LLC cells at 64 nM, but in Figure 3E, LZ90 could significantly induce apoptosis of LLC cells at 60 nM. The authors need to explain the inconsistency of the LZ90 compound to LLC cells.
4. In results 3.1, the authors stated they constructed a tumor model in which mice were inoculated with LLC cells. But in fig 1I, it seems they used LLC cells with double over-expression of NAMPT and PD-L1. Which cell did you use exactly? In addition, I cannot tell from fig 1I that whether the two proteins are successfully overexpressed.
5. In T cell-mediated tumor cell killing assay, PBMCs were cultured at the beginning, and T cells were used in the later experiment. Is there a sorting step in this process?
6. In the discussion section, the authors explain the necessity of double-target NAMPT and PD-L1 in a redundant manner. The author should reframe the way of writing. how SIRT epigenetic regulation can affect the expression of PD-L1, and Vice versa should be discussed in detail.

Minor:

There are formatting errors of graph annotation in many figures. For example, there should be one space between the numerical value and the unit symbol, please check and revise it.

Revision Decision (Highlight the revised manuscript in yellow)

Response to Reviewer 1

Point-by-point reply to reviewers' comments

Reviewer 1:

Cancer's complexity arises from its genetic diversity and metabolic patterns, which also influence immune cells. The coenzyme NAD⁺ and its key enzyme, NAMPT, play pivotal roles in tumor metabolism and immune modulation. With the advent of modern oncology, there has been a pronounced shift towards checkpoint inhibitor immunotherapy, which specifically targets the PD-1/PD-L1 interaction, aiming to amplify the immune system's ability to detect and combat tumor cells. This work explored the link between NAMPT and PD-L1, revealing their potential as therapeutic targets. However, upon meticulous review and assessment, we think that this work, while valuable, does not meet the rigorous standards for publication in EMBO Molecular Medicine. Specifically, the mechanistic investigations presented seem to be premature.

<Response>

: Thanks to the reviewer for his/her valuable suggestions. According to the reviewer's comment, we further explored the underlying regulation mechanism between NAMPT and PD-L1. The results demonstrate:

1) NAMPT epigenetic regulates PD-L1 by inhibiting the deacetylation NF-κB p65 K310, a non-histone substrate of SIRT.

Our results reveal that NAMPT inhibition would result in the upregulation of PD-L1 *in vitro* and *in vivo*. Mechanistically, our data demonstrates that NAMPT inhibition decreases SIRT activity, and raises the acetylation level of NF-κB p65 K310 site which is critical for NF-κB transcriptional activity, and next contributes to binding of NF-κB p65 on CD274 (PD-L1) promoter region, subsequently results in the upregulation of PD-L1 (Figure 1). In addition, NMN, a NAD⁺ precursor, could reverse the above epigenetic regulation, which further confirmed the crucial role of NAMPT in PD-L1 regulation.

Figure 1: NAMPT inhibition promoting PD-L1 expression mediated by SIRT1 epigenetic regulation (Graphics are drawn by Figdraw)

2) PD-L1 inhibition inducing NAMPT expression through HIF-1 α /glycolysis pathway.

We found that inhibition of PD-L1 also could induce the expression of NAMPT in the presence of T cells. To be specific, inhibition of the PD-1/PD-L1 axis could activate the T cells immunity, promoting the release of IFN- γ , TNF- α , and other inflammatory factors, and upregulating the expression of HIF-1 α under the normal oxygen state. The upregulation of HIF-1 α will increase the expression of downstream glycolytic-related genes and the content of pyruvate, the product of glycolysis, thus promoting the consumption of NAD⁺ and inducing the expression of NAMPT to provide the high demand for NAD⁺ (Figure 2). Notably, when we knock out HIF-1 α , the above regulatory relationship disappears. The preliminary findings of our study indicate that the immune response exerts a certain influence on tumor metabolism, and HIF-1 α plays a crucial role as a regulatory factor in this process.

Figure 2: PD-L1 inhibition inducing NAMPT expression through HIF-1 α /glycolysis pathway.

(Graphics are drawn by Figdraw)

1. <Comments>

: NAMPT and PD-L1 exhibit a negative correlation. Inhibiting NAMPT leads to an elevated expression of PD-L1, which enhances the immune evasion of tumor cells and subsequently weakens the effectiveness of immunotherapy. Given this interplay, the author's decision to design a dual-target inhibitor for both NAMPT and PD-L1 raises the question: does this strategy possess significant potential for medical application?

<Response>

Thanks for the reviewer raising this meaningful comment. Evaluating the value of a therapeutic strategy must be based on its clinical potential. The issue can be addressed as the following facts:

1) Both NAMPT and PD-L1 play important roles in tumor development and progression.

Nicotinamide adenine dinucleotide (NAD⁺) is a coenzyme involved in redox reactions (1). Nicotinamide phosphoribosyltransferase (NAMPT), the rate-limiting enzyme in NAD⁺ biosynthesis, is a key enzyme in cell metabolism that influences the activity of NAD⁺-dependent enzymes to regulate tumor metabolism (2). It has been reported that tumor cells displayed very active metabolism and consume NAD⁺ at a faster rate than normal cells and expressed high levels of NAMPT to satisfy the high demand for

NAD⁺ (3). Several studies have shown that NAMPT and NAD⁺ are frequently upregulated in several human malignancies and play key roles in the development, progression, and recurrence of cancer (4,5). Thus, NAMPT is considered as a potential anti-tumor target. Research has mainly focused on inhibiting the key enzyme of the latter NAD⁺ production route, NAMPT, leading to the identification of numerous inhibitors, including FK866 and CHS-828. Unfortunately, the clinical activity of these agents is limited, suggesting that the approaches for targeting NAD⁺ production in tumors need to be refined. Notably, two compounds of the second generation of NAMPT inhibitors (OT-82 and KPT-9274) are being currently investigated in phase I trials. So far, no results have been disclosed (6). Thus, NAMPT is a potential target in anti-cancer drugs.

Programmed Cell Death Ligand 1 (PD-L1) is a trans-membrane protein that is considered to be a co-inhibitory factor of the immune response. PD-L1 can combine with PD-1 to reduce the proliferation of PD-1-positive cells, inhibit their cytokine secretion, and induce apoptosis. PD-L1 also plays an important role in various malignancies where it can attenuate the host immune response to tumor cells. PD-L1 is usually highly expressed in tumor cells to evade immune surveillance, and it has been reported that high expression of PD-L1 can make tumor cells more sensitive to PD-1/PD-L1 inhibitors, having a huge effect on cancer therapy. PD-1/PD-L1 axis can be modulated by various signals in cancer cells, which exert a critical role in tumorigenesis. Studies have found that nuclear PD-L1 expression is associated with poor prognosis in human colorectal and prostate cancers. Based on the immense success in clinical trials, ten α -PD-1 (nivolumab, pembrolizumab, cemiplimab, sintilimab, camrelizumab, toripalimab, tislelizumab, zimberelimab, prolgolimab, and dostarlimab) and three α -PD-L1 antibodies (atezolizumab, durvalumab, and avelumab) have been approved for various types of cancers. Nevertheless, the low response rate of α -PD-1/PD-L1 therapy remains to be resolved. The combined strategy of PD-1/PD-L1 with other targets would be effective approach to anti-tumor therapy (7).

Therefore, the accumulated studies have found that NAMPT and PD-L1 are involved in tumor development and progression, which are considered promising anti-cancer drug targets.

2) NAMPT and PD-L1 are highly expressed in tumor tissues from the TCGA database.

We analyzed the prognosis between NAMPT and PD-L1 expression levels and found that patients with high expression of NAMPT and low expression of PD-L1 had the worst prognosis (see *Fig.1A* in the current manuscript). This observation suggests that the NAMPT^{High} PD-L1^{Low} expression pattern is significant in tumor progression. According to the analysis of TCGA data, the proportion of patients with high expression of NAMPT was 22.32%, and the proportion of patients with high expression of PD-L1 was 16.35%. Considering the negative regulatory relationship between NAMPT and PD-L1, our targeted drugs should have a good effect on clinical cases with high expression of single target and double targets. In total, the proportion of that suitable cases could reach 62.93% (Fig. 3A), which means that most of patients could benefit from double targets compound LZFPN-90.

3) Experimental verification of the potential of double target strategy.

For further verification, we established a series of LCC cell lines with overexpression of NAMPT or/and PD-L1 and co-cultured with T cells *in vitro*. Next, we treated with PD-L1 inhibitor CQ-85 and BMS202, NAMPT inhibitor FK866, and dual-target inhibitor LZFPN-90. **(1)** Compared with the single target inhibitors, we found that the double target inhibitor LZFPN90 had a better effect and could significantly inhibit the growth of cancer cells, indicating that the effect of the double target small molecule inhibitor is dependent on the expression levels of the two targets (Fig. 3B). **(2)** In addition, we compared the different effects of LZFPN-90 in LCC cell lines with overexpression of NAMPT or/and PD-L1. Our results indicated that LZFPN-90 displayed the strongest killing efficacy in LLC OE N/P cells with high expression of both targets with the inhibition rate of about 80%, showed moderate anti-tumor action in LLC OE PD-L1 and LLC OE

NAMPT cells with inhibition rate about 60%, and exhibited a weak killing effect on LLC Vector cells with an inhibition rate about 40%. The above results suggest that LZFPN-90 is effective in cell lines with overexpression in single target or double targets. (see Fig. 3B) **(3)** In addition, considering that the safety of dual-targeted small molecule inhibitor is generally better than the combination of treatment with two inhibitors separately, dual-targeted compound LZFPN-90 might own potential clinical value.

4) In summary, NAMPT and PD-L1 are crucial targets in anti-tumor strategy. Considering the higher proportion of single expression or co-expression of both targets and superior anti-tumor efficacy of dual-target compound LZFPN-90, it would be a promising strategy to co-target NAMPT and PD-L1 in clinical application.

A

Expression Patten	NAMPT LowPD-L1 ^{High}	NAMPT ^{High} PD-L1 ^{High}	NAMPT ^{low} PD-L1 ^{low}	NAMPT ^{High} PD-L1 ^{low}
N	1770	2626	4014	2416
Rate (%)	16.35	24.26	37.08	22.31

B
Fig 3: The killing ability of LZFPN-90 on LLC cells with different NAMPT and PD-L1 expression patterns.

(A) The different expression patterns of NAMPT and PD-L1 in cancers. Data was obtained from the TCGA database. (B) T cell-mediated tumor cell killing assay. T cells were co-cultured with LLC Vector, LLC OE PD-L1, LLC OE NAMPT, and LLC OE N/P cells in the presence of LZFPN-90, FK866, CQ-85, or BMS202. The concentration of these inhibitors was 100 nM, and surviving tumor cells were stained by crystal violet. The relative added value of activated T cells is shown below. Data are presented as mean \pm SEM; * P < 0.05, ** P < 0.01, *** P < 0.001, compared with control. n.s., not significant.

References

1. Covarrubias AJ, Perrone R, Grozio A, Verdin E. NAD⁺ metabolism and its roles in cellular processes during aging. *Nat Rev Mol Cell Biol* **2021**;22(2):119-41.
2. Gardell SJ, Hopf M, Khan A, Dispagna M, Hampton Sessions E, Falter R, *et al.* Boosting NAD⁺ with a small molecule that activates NAMPT. *Nat Commun* **2019**;10(1):3241.
3. Nakajima TE, Yamada Y, Hamano T, Furuta K, Gotoda T, Katai H, *et al.* Adipocytokine levels in gastric cancer patients: resistin and visfatin as biomarkers of gastric cancer. *J Gastroenterol* **2009**;44(7):685-90.
4. Lucena-Cacace A, Otero-Albiol D, Jiménez-García MP, Muñoz-Galvan S, Carnero A. NAMPT Is a Potent Oncogene in Colon Cancer Progression that Modulates Cancer Stem Cell Properties and Resistance to Therapy through SIRT1 and PARP. *Clin Cancer Res* **2018**;24(5):1202-15.
5. Zhang H, Zhang N, Liu Y, Su P, Liang Y, Li Y, *et al.* Epigenetic Regulation of NAMPT by NAMPT-AS Drives Metastatic Progression in Triple-Negative Breast Cancer. *Cancer Res* **2019**;79(13):3347-59.
6. Ghanem MS, Monacelli F, Nencioni A. Advances in NAD-Lowering Agents for Cancer Treatment. *Nutrients*. **2021**;13(5):1665.
7. Yi M, Zheng X, Niu M, Zhu S, Ge H, Wu K. Combination strategies with PD-1/PD-L1 blockade: current advances and future directions. *Mol Cancer*. **2022**;21(1):28.

2. <Comments>

: In section 3.1 on Analysis of the relationship between NAMPT and PD-L1 in cancer cells, the author believes that inhibiting NAMPT promotes the expression of PD-L1 through the epigenetic regulation of SIRT. However, the intricate interplay between NAMPT and PD-L1 remains somewhat elusive. In 2021, Wang et al. highlighted that NAMPT drives interferon γ (IFN γ)-induced PD-L1 expression in multiple types of tumors and governs tumor immune evasion in a CD8⁺ T cell-dependent manner (Cell Metabolism 2021 33, 110-127). The author might need to continue in-depth investigation into the mechanism.

<Response>

According to the reviewer's comment, we explore the underlying mechanism of PD-L1 expression regulated by NAMPT. **First**, we want to know whether NAMPT inhibitor FK866 could affect the SIRT activity. Thus, we detected the substrate of acetylation Tubulin. As shown in Figure 4A, FK866 treatment could increase acetylation Tubulin, whereas adding NMN would reverse the increase of acetylation Tubulin induced by FK866, indicating that FK866, depending on its NAMPT inhibitory action, would affect the de-acetylation ability of SIRT (Fig. 4A). **Second**, as shown in Figure 4B, we further detected the expression of PD-L1 after treatment with NAMPT inhibitor or/and SIRT inhibitor. Our results showed that whether

NAMPT inhibitor FK866 or SIRT inhibitor Sirtinol could upregulate PD-L1, and combined treatment could enhance the upregulation of PD-L1, suggesting upregulation of PD-L1 might be involved in SIRT activity. In addition, NAMPT knockdown by siRNA also got a similar result, suggesting the specific action of NAMPT on PD-L1 regulation (Fig. 4B). **Third**, in order to clarify the upregulation of PD-L1 whether is related to epigenetic regulation by SIRT, we detected the transcription of PD-L1 (CD274) using Real-time PCR. As shown in Figure 4C, a single treatment of FK866 or Sirtinol could upregulate the transcription of PD-L1. Meanwhile, the combination treatment could enhance this upregulation, suggesting the increase of PD-L1 by FK866 or Sirtinol happens at the transcription level (Fig. 4C). **Fourth**, it has been reported that SIRT1 can directly deacetylate the K310 site of NF- κ B p65, and enhance its activity as a transcription factor, thus affecting the expression of PD-L1 (1-4). Therefore, we speculated that inhibition of NAMPT might promote the acetylation of the p65 K310 site, promote the transcription of PD-L1, and upregulate the expression of PD-L1 by affecting the activity of SIRT1. We next conducted experimental verification, and the results showed that inhibition of NAMPT could increase the acetylation of the p65 K310 site (Fig. 4D), and upregulate the PD-L1 expression (Fig. 4D). The addition of NMN reversed the increased acetylation of the p65 K310 site and expression of PD-L1 caused by FK866 inhibiting SIRT activity (Fig. 4D). **Last**, to detect whether the PD-L1 transcription activation induced by FK866 was caused by NF- κ B p65, we performed Chip-qPCR experiments. As shown in Figure 4E, the results showed that acetylation of p65 K310 could be enriched at the CD274 gene locus P3, but not P1 and P2, after the use of FK866 and Sirtinol, while the accumulation of acetylation of p65 K310 could be reduced at the CD274 gene locus after the use of NMN. Therefore, our study revealed a new regulation mechanism of PD-L1 by NAMPT (see Figure 1 in the response letter).

As the reviewer comments, different underlying mechanisms might be involved in PD-L1 regulation. It has been reported that NAD⁺ maintains PD-L1 expression at tumor immune checkpoint through epigenetic modification in response to IFN- γ secreted by T cells and drives tumor immune escape (Cell Metabolism 2021 33,

110-127). In addition, several studies have revealed that PKM2, which converts phosphoenolpyruvate into pyruvate, is highly expressed in tumors (5, 6) and enhances PD-L1 expression in both tumor and immune cells via epigenetic mechanisms (7). A recent study demonstrated that NAD⁺ depletion and autophagy induced by NAMPT inhibitors mediate the upregulation of PD-L1 transcripts and cell surface protein levels in glioblastoma cells (8). Here, we found that PD-L1 might be also regulated by NAMPT which is dependent on the epigenetic regulation of SIRT. In detail, inhibition of NAMPT would contribute to the inactivation of SIRT, next, promote acetylation at the NF- κ B p65 K310 site, and then activate p65 to bind its target gene, subsequently upregulating PD-L1 expression at the transcription level. Taken together, our study illustrates a new mechanism to regulate PD-L1 in tumor cells. **The related content has been updated in the current manuscript (Results and Discussion section).**

Fig 4: Effects of NAMPT inhibition on PD-L1 expression

(A) The Ac-tubulin protein expression was analyzed after adding FK866 and NMN for 72 h to LLC cells by western blot. (LLC: FK866 2 μ M, NMN 1 mM). (B) The PD-L1 protein expression was analyzed after adding FK866, Sirtinol, and a combination of both drugs for 72 h to A2780 and LLC cells by western blot, using the NAMPT knockdown cells by siRNA to analyze the PD-L1 protein expression. (A2780: FK866 0.5 μ M, Sirtinol 10 μ M; LLC: FK866 2 μ M, Sirtinol 10 μ M). (C). Expression of PD-L1 mRNA after adding FK866, Sirtinol and combination of both drugs for 72 h to A2780 cells (FK866: 0.5 μ M; Sirtinol: 10 μ M) and LLC cells (FK866: 2 μ M; Sirtinol: 10 μ M). (D) In the Ac-p65 (K310), the protein levels of PD-L1 were detected after adding FK866 and NMN for 72 h to A2780 and LLC cells by western blot. (A2780: FK866 0.5 μ M, NMN 1 mM; LLC: FK866 2 μ M, NMN 1 mM). (E) After treatment with FK866, Sirtinol and NMN for 72 h to A2780 cells, the enrichment of Ac-p65 (K310) in PD-L1 promoter region (P1, P2, P3) was detected by ChIP-qPCR (A2780: FK866 0.5 μ M, Sirtinol 10 μ M; NMN 1 mM). Data are presented as mean \pm SEM; * P < 0.05, ** P < 0.01, *** P < 0.001 compared with the control group. n.s., not significant.

References

1. Ghisays F, Brace CS, Yackly SM, Kwon HJ, Mills KF, Kashentseva E, Dmitriev IP, Curiel DT, Imai SI, Ellenberger T. The N-Terminal Domain of SIRT1 Is a Positive Regulator of Endogenous SIRT1-Dependent Deacetylation and Transcriptional Outputs. *Cell Rep*. **2015**;10(10):1665-1673.
2. Yeung F, Hoberg JE, Ramsey CS, Keller MD, Jones DR, Frye RA, Mayo MW. Modulation of NF-kappaB-dependent transcription and cell survival by the SIRT1 deacetylase. *EMBO J*. **2004**;23(12):2369-80.
3. Chen LF, Mu Y, Greene WC. Acetylation of RelA at discrete sites regulates distinct nuclear functions of NF-kappaB. *EMBO J*. **2002**;21(23):6539-48.
4. Zhou Y, Jin X, Yu H, Qin G, Pan P, Zhao J, Chen T, Liang X, Sun Y, Wang B, Ren D, Zhu S, Wu H. HDAC5 modulates PD-L1 expression and cancer immunity via p65 deacetylation in pancreatic cancer. *Theranostics*. **2022**;12(5):2080-2094.
5. Christofk HR, Vander Heiden MG, Harris MH, Ramanathan A, Gerszten RE, Wei R, *et al*. The M2 splice isoform of pyruvate kinase is important for cancer metabolism and tumor growth. *Nature* **2008**;452(7184):230-3.
6. Elf SE, Chen J. Targeting glucose metabolism in patients with cancer. *Cancer* **2014**;120(6):774-80.
7. Palsson-McDermott EM, Dyck L, Zasłona Z, Menon D, McGettrick AF, Mills KHG, *et al*. Pyruvate Kinase M2 Is Required for the Expression of the Immune Checkpoint PD-L1 in Immune Cells and Tumors. *Front Immunol* **2017**;8:1300.
8. Wang S, Yao F, Lu X, Li Q, Su Z, Lee J-H, *et al*. Temozolomide promotes immune escape of GBM cells via upregulating PD-L1. *Am J Cancer Res* **2019**;9(6):1161-71.

3. <Comments>

: The authors found that inhibition of PD-L1 by QC-85 in the presence of T cells can lead to the upregulation of NAMPT by affecting the tumor immune microenvironment. 1) However, the experiment only considered the effect of CQ-85 on PD-L1, not excluding its influence on T cells. Similarly, the authors indicated that LZFPN-90 can enhance the functionality of T cells by inhibiting PD-L1, however, does not rule out the effect of LZFPN-90 on T cells. 2) In addition, in this assay, the author used BMS202 as the control group. To exclude uncertainties and for better comparison, CQ-85 should be included in the assay.

<Response>

Response for 1) *However, the experiment only considered the effect of CQ-85 on PD-L1, not excluding its influence on T cells. Similarly, the authors indicated that LZFPN-90 can enhance the functionality of T cells by inhibiting PD-L1, however, does not rule out the effect of LZFPN-90 on T cells.*

As the reviewer said, the experiment only considered the effect of CQ-85 and LZFPN-90 on PD-L1 of tumor cells, not excluding its influence on T cells. To address this issue, we performed the following experiments.

- 1) We incubated T cells with CQ-85 and LZFPN-90 (LZ90) for 48 h and 72 h before co-culture with LLC cells and detected the expression of NAMPT and the proportion of CD8⁺ T cells after drug treatment in T cells. As shown in Figure 5A, T cells expressed less NAMPT than LLC cells. Compared with co-culture cells without drug pre-treatment, there was no significant change in the expression of NAMPT after treating T cells with CQ-85 and LZFPN-90 (Fig. 5A), and the proportion of CD8⁺ T cells also did not increase significantly (Fig. 5B).
- 2) It should be noted that the treatment of CQ-85 and LZFPN-90 in the co-culture system with LLC cells and T cells can significantly upregulate the expression of NAMPT in tumor cells (see *Fig. 2B* in current revised manuscript) as well as increase the proportion of CD8⁺ T cells (Fig 5B), but not in T cells(see Fig. 5C), indicating that NAMPT regulation and immune function change mediated by PD-L1 inhibitor of CQ-85 first targets cancer cells, but not T cells, and subsequently affects tumor immune microenvironment.

Response for 2): *In addition, in this assay, the author used BMS202 as the control group. To exclude uncertainties and for better comparison, CQ-85 should be included in the assay.*

As the reviewer said, to exclude uncertainties and for better comparison, CQ-85 should be included in the assay. In the co-culture system, we added the comparison between CQ-85 and BMS202 under the same conditions as before. Next, we compared the effects of CQ-85 and BMS202 on the activation of CD8⁺ T cells and the levels of IFN- γ in a co-culture system with LLC cells and found that both of them could increase the proportion of CD8⁺ T cells and the content of IFN- γ , indicating that both PD-L1 inhibitors owned same effect (Fig. 5C). **The results were also included in the current revised manuscript (see Fig S4A).**

Fig 5: Effects of CQ-85 and LZFPN-90 on tumor immune microenvironment.

(A) After T cells were incubated with CQ-85 (100 nM) and LZFPN-90 (LZ90, 100 nM) for 48 h and 72 h, the expression of NAMPT in LLC cells and T cells was detected by western blot. (B) LLC cells were co-cultured with T cells. T + CQ-85 and T + LZFPN-90 were known as pre-treatment groups, which indicated that T cells were treated with CQ-85 (100 nM) and LZFPN-90 (100 nM) for 48 h and then co-cultured with LLC cells. CQ-85 (100 nM) and LZFPN-90 (100 nM) groups were remixed groups, which indicated that drug incubation was performed during co-culture with LLC cells and T cells. (C) The proportion and representative staining of CD8⁺ T cells and

IFN γ ⁺ of CD8⁺ T cells after co-culture with LLC cells, CQ-85 (100 nM), BMS202 (100 nM). Data are presented as mean \pm SEM; * P < 0.05, ** P < 0.01, *** P < 0.001 compared with control. n.s., not significant.

4. <Comments>

: The studies in the current paper are mainly carried out through inhibiting the biological targets (NAMPT and PD-L1). The knockout or knockdown studies are required for further comparison and verification.

<Response>

: According to the reviewer's suggestion, we constructed LLC NAMPT knockdown (sh-NAMPT) and PD-L1 knockdown (sh-PD-L1) LLC cells, respectively (Fig. 6A), and performed the apoptosis and colony formation analysis to verify whether both targets and what's the extent of them, are involved into anti-tumor action of LZFPN-90. The data was listed as follows:

- 1) **Apoptosis assessment:** We detected the apoptosis of sh-NAMPT and sh-PD-L1 cells after treatment with LZFPN-90 and FK866 for three days. Our results showed that the rate of apoptosis in the sh-NAMPT group was significantly lower than that in the Vector group and sh-PD-L1 group, indicating that NAMPT is mainly related to apoptosis induction function (Fig. 6B).
- 2) **Colony formation assessment after knockdown NAMPT:** Colony formation assay usually is used to detect the proliferation and self-renewal ability of cancer cells. As shown in Figure 6C, treatment with LZFPN-90 and FK866 could significantly reduce the colony formation ability in control cells, whereas the effects of both inhibitors were weakened in NAMPT knockdown cells, suggesting both inhibitors exhibit their anti-proliferation and anti-self-renewal ability partially dependent on NAMPT.
- 3) **Colony formation assessment after knockdown PD-L1:** To further access the role of PD-L1, we established co-cultured conditions using PD-L1 knockdown cells. The results showed that knocking down PD-L1 significantly enhanced the killing ability of T cells to tumor cells, which was consistent with the conclusion of previous studies (Fig. 6D). After inhibitor treatment, the relative killing ability

of T cells to sh Vector group was significantly stronger than the sh-PD-L1 group, indicating that LZFPN-90 promoted the killing ability of T cells to tumor cells by inhibiting PD-L1 (Fig. 6D).

Taken together, our results indicate that, knockdown either NAMPT or PD-L1, would affect the malignant phenotype, including proliferation and anti-apoptosis, and weaken the apoptosis induction and anti-proliferation action of LZFPN-90.

The results of this part are included in the article (see Fig S3 and S6).

Fig 6: Effects of LZFPN-90 on the apoptosis and proliferation of LLC cells with knockdown NAMPT or PD-L1.

(A) NAMPT and PD-L1 were knocked down by sh RNA in the LLC cells and verified by western blot. (B) The apoptosis of the LLC sh Vector and LLC sh NAMPT cells was detected after three days of treatment with LZFPN-90 (100 nM) and FK866 (100 nM). (C) In LLC sh Vector and LLC sh NAMPT cells, after three days of treatment with LZFPN-90 (100 nM) and FK866 (100 nM), the cell proliferation was detected by cloning assay. (D) T cells were co-cultured with LLC sh Vector and LLC sh PD-L1 cells, and treated with CQ-85 (100 nM), BMS202 (100 nM), and LZFPN-90 (100 nM) to detect the killing ability of tumor cells. Data is presented as mean \pm SEM; * $P < 0.05$, ** $P < 0.01$, *** $P < 0.001$ compared with control. n.s., not significant.

5. <Comments>

: Besides, in the animal study, dual inhibitor LZFPN-90 should also be compared with the co-treatment (FK866 + CQ-85) group.

<Response>

: According to the reviewer's suggestion, we added the co-treatment (FK866 + CQ-85) group *in vivo* animal experiments. In addition, we also repeated the LZFPN-90 administration experiments *in vivo* to compare with the co-treatment (FK866 + CQ-85) group. The results are shown in the following:

- 1) Effect of co-treatment FK866 and CQ-85 on tumor inhibition rate.** The pharmacodynamics of LZFPN-90 and co-treatment (FK866 + CQ-85) group were evaluated *in vivo* in mice bearing tumors derived from A20 cells. Co-treatment FK866(20 mg/kg) and CQ-85(20 mg/kg) effectively inhibited tumor growth in A20 xenograft mice, with the tumor inhibition rate (TIR) reached 75.11%. Similarly, LZFPN-90 also showed a stronger effect at the dose of 40 mg/kg with the tumor inhibition rate (TIR) reaching 82.70% (Fig. 7A, B). Therefore, the anti-tumor effect of dual-target compound LZFPN-90 is considered as similar, or a little bit enhanced tendency, with the combined treatment of single inhibitors at the same dose.
- 2) Effect of co-treatment FK866 and CQ-85 on gross toxicity.** We also compared the toxicity characteristics of both treatments. Our results indicated that LZFPN-90 treatment had no obvious effect on mouse organ index or body weight (Fig. 7C, D). In contrast, co-treatment FK866 and CQ-85 resulted in a reduction in the body weight of the mice (Fig. 7C). These results indicated that the dual-target inhibitor LZFPN-90 is safer than the co-treatment single inhibitors at the same dose.
- 3) Effect of co-treatment FK866 and CQ-85 on immune-infiltration.** In addition, we examined the effect of LZFPN-90 and co-treatment (FK866 + CQ-85) on CD4⁺ and CD8⁺ T cells in A20 xenograft mice models. The results showed that CD4⁺ and CD8⁺ T cells were effectively activated by LZFPN-90 and co-treatment (FK866 + CQ-85) (Fig. 7E). Simultaneously, we also found that LZFPN-90 and co-treatment (FK866 + CQ-85) effectively promoted the secretion of CD69 and

IFN- γ (Fig. 7E). As for the CD69 and IFN- γ , LZFPN-90 displayed more significant efficacy than co-treatment group. As for the CD8, LZFPN-90 exhibited an enhanced tendency as compared with co-treatment.

4) **In summary**, it can be demonstrated that compared with combined administration at the same dose, the dual-target compound LZFPN-90 displayed enhanced efficacy and decreased toxicity.

The results of this part are included in the article (see Fig. S5 and S6).

Figure 7. Effect of LZFPN-90 and co-treatment (FK866 + CQ-85) on tumor growth, toxicity, and immune microenvironment.

(A-B) Tumor images and tumor weight at endpoint (A) and tumor growth curves (B). In (A), the percent represents the tumor growth inhibition (TGI) rate at the endpoint of the experiment. (C-D) Mouse body weight (C) and organ index (D) of A20 model mice. A20 tumor-bearing Balb/c mice were treated with CQ-85 (20 mg/kg, i.p.) and FK866 (20 mg/kg, i.p.) combination and LZFPN-90 (40 mg/kg, i.p.), n=6. (E) Flow cytometry analyzed the proportion of CD4⁺, CD8⁺, and CD69⁺ T cells and the content of IFN- γ in CD8⁺ T cells in the tumors from the A20 mouse model. Data is presented as mean \pm SEM; * P < 0.05, ** P < 0.01 compared with control. n.s., not significant.

Response to Reviewer 2

Point-by-point reply to reviewers' comments

Reviewer 2:

In the present manuscript, Yang et al. present a compelling therapeutic approach with broad applicability to various types of cancer. Using both in vitro and in vivo models, the authors demonstrate that simultaneous targeting of NAMPT and PD-L1 yields a more potent antitumor response compared to targeting each protein independently. A significant highlight of this study is the synthesis of the first dual NAMPT/PD-L1 targeting small molecule "LZFPN-90". This molecule is shown to effectively inhibit cancer cell proliferation, arrest the cell cycle, induce apoptosis, and substantially reduce tumor growth in xenograft models across multiple cancer types. The study is well-executed and elegantly presented. The authors employ a range of methodologies to substantiate their findings, which adds significant credibility to their work. Nevertheless, there are minor points that could be addressed to further consolidate the study's conclusions.

Minor comments

1. <Comments>

: In the current study, the authors have used various mouse strains (nude, Balb/c, C57BL/6) to create xenograft models. However, details concerning model generation are lacking, making it difficult to ascertain specifics like the mouse strain used (in each reported experiment), the number of injected cells,

pre-transplantation irradiation, drug administration timing, and protocols, as well as dosage schedules. Providing this information is crucial for the experiment's reproducibility. What is the rationale for housing Balb/c and C57BL/6 mice in SPF conditions?

<Response>

1) **In mice bearing A20 tumors.** The 4-6 weeks-old Balb/c mice and Balb/c nude mice were used in this experiment. It should be noted that, to assess whether the LZFPN-90 exhibited the anti-tumor effects depending on the microenvironment, both immune-deficient and immune-system-intact mice were used in this study. In detail, A20 cells (5×10^6 cells in 0.2 ml phosphate-buffered saline) were subcutaneously injected into the right armpit of Balb/c mice and Balb/c nude mice. Tumor volume was measured using calipers every 2 days and calculated using the following formula: $(\text{long diameter}) \times (\text{short diameter})^2/2$. Balb/c mice were divided into groups when the tumor grew to 50-100cm³, including solvent group, CQ-85 group (20 mg/kg), FK866 group (20 mg/kg), and LZFPN-90 (20 mg/kg, 40 mg/kg). For comparison, Balb/c nude mice were treated with the solvent group and LZFPN-90 (20 mg/kg). After 12 days of administration, the animals were anesthetized, and tumors were excised. Partial fresh tumor tissues were subjected to immune cell isolation. Tissues were either fixed in 4% formalin or stored at -80 °c until further analysis. **The model construction process has been rewritten in this revised manuscript.** (*see Materials and Methods*)

2) **In mice bearing LLC tumors.** The 4-6 weeks-old C57BL/6 mice were used in this study. LLC Vector control cells and LLC OE NAMPT/PD-L1 cells (2×10^6 cells in 0.2 ml phosphate-buffered saline) were subcutaneously injected into the right armpit of C57BL/6 mice. Tumor volume was measured using calipers every 2 days and calculated using the following formula: $(\text{long diameter}) \times (\text{short diameter})^2/2$. C57BL/6 mice were divided into groups when the tumor grew to 50-100 cm³, including the solvent group, CQ-85 group (20 mg/kg), FK866 group (20 mg/kg), and LZFPN-90 (20 mg/kg). For comparison, mice bearing LLC Vector tumors were

treated with solvent or LZFPN-90 (20 mg/kg). After 14 days of administration, the animals were anesthetized, and tumors were excised. Partial fresh tumor tissues were subjected to immune cell isolation. Tissues were either fixed in 4% formalin or stored at $-80\text{ }^{\circ}\text{C}$ until further analysis. **The model construction process has been rewritten in this revised manuscript.** (*see Materials and Methods*).

3) Housing Balb/c and C57BL/6 mice in SPF conditions. All of our animal experiments are conducted in an SPF environment to ensure that they do not carry specific pathogens, thus providing a more controlled and accurate laboratory environment. SPF animals are essential in research such as immunology, genetics, and disease modeling. For example, animal models studying immune system dysfunction can not accurately assess the effects of immune deficiency if the experimental animals themselves carry unknown pathogens. Similarly, in oncology research, SPF animals provide an ideal environment to evaluate the effects of anti-tumor drugs. Therefore, to ensure the accuracy and repeatability of the experiment, all our animals are raised in an SPF environment.

2. <Comments>

*: The study presents intriguing results involving a dual-targeting inhibitor.1) Is the antitumor activity equally attributable to both NAMPT and PD-L1 inhibition?
2) Additionally, can LZFPN-90 induce cell death when NMN, nicotinic acid, or alternative NAD precursors (NR, NRH, NAR) are present in the tumor microenvironment?*

<Response>

1) The antitumor activity is equally attributable to both NAMPT and PD-L1 inhibition.

To address this issue, we constructed LLC sh NAMPT cells and LLC sh PD-L1 cells and established the co-culture condition with T cells. As shown in Figure 8A, the addition of LZFPN-90 (100 nM) could significantly inhibit colony formation number, with an inhibition rate reaching 48.25% in LLC cells. However, the inhibitory ability of LZFPN-90 (100 nM) was reduced in LLC sh-NAMPT or sh-PD-L1 co-cultured system, with inhibition rates of 26.8% and 21.94%, respectively, which suggests that

both NAMPT and PD-L1 play equally contribution to the antitumor activity of LZFPN-90 (Fig 8B).

2) Additionally, can LZFPN-90 induce cell death when NMN, nicotinic acid, or alternative NAD precursors (NR, NRH, NAR) are present in the tumor microenvironment?

NMN is the precursor substance that most directly affects the content of NAD⁺, so we conducted experiments using NMN. Without T cells, the combination of LZFPN-90 and NMN completely reversed the inhibitory effect of LZFPN-90 on cell survival. It is worth noting that in the co-culture condition, we found that the LZFPN-90 significantly inhibited cell survival, while the NMN did not cause significant changes in cell survival. The effect of LZFPN-90 on cell survival was partially reversed after the administration of NMN, but the overall effect was inhibitory (Fig 8C). These results preliminarily indicated that in the tumor immune microenvironment, the change of NAD⁺ content can affect the inhibitory effect of LZFPN-90 on tumor cells, but in general, LZFPN-90 can still exert its inhibitory effect on tumor cells.

Fig 8: Effects of LZFPN-90 and NMN on tumor immune microenvironment.

(A-B) In LLC sh Vector (A), LLC sh NAMPT, and LLC sh PD-L1 cells (B), the cell proliferation was detected by T cell killing assay after treatment with LZFPN-90 (100 nM) for three days. (C) T cells were co-cultured with LLC cells, and treated with LZFPN-90 (100 nM), and NMN (1 mM) to detect the cell proliferation by T cell killing assay. Data are presented as mean \pm SEM; * $P < 0.05$, ** $P < 0.01$, *** $P < 0.001$ compared with control. n.s., not significant.

3. <Comments>

: In the graphical abstract, there's a typographical error: the dual-targeting inhibitor "LZFPN-90" is incorrectly spelled as "LZ-90."

<Response>

Thanks to the reviewer for his/her valuable suggestion. In fact, LZ-90 is the abbreviation of LZFPN-90. We think that this abbreviation might cause misunderstandings for the readers. Thus, we have changed LZ-90 to LZFPN-90 in the current manuscript. (see graphical abstract)

4. <Comments>

: Existing literature already discusses the modulation of PD-L1 by NAD metabolism. I recommend that the authors incorporate these works into their discussion for a more comprehensive analysis.

<Response>

According to the reviewer's comment, we have added the mentioned points and our new finding in regulation mechanism in the current manuscript (*Discussion section*).

→

The regulation of PD-L1 expression is a complex and highly coordinated series of molecular signaling events. In tumor cells, PD-L1 expression may be related to metabolism, which in turn affects the immune status of the TME and even antitumor immunity. Studies have shown that tumors stabilized PD-L1 expression to control glucose metabolism by enhancing N-glycosylation and the EGFR/ERK/c-Jun pathways (1,2). In addition, several studies have revealed that PKM2, which converted phosphoenolpyruvate into pyruvate, was highly expressed in tumors (3,4) and enhanced PD-L1 expression in both tumor and immune cells via epigenetic mechanisms (5). Furthermore, Wang S *et al* reported that NAD⁺ depletion and autophagy induced by NAMPT inhibitors mediate the upregulation of PD-L1 transcripts and cell surface protein levels in glioblastoma cells (6). A recent study demonstrated study revealed that PD-L1 expression also could be regulated by IFN- γ driven by NAMPT in a CD8⁺ T cell-dependent manner, suggesting the possible role of TME in this process (7). Our study found the same phenomenon: NAMPT inhibition would result in the upregulation of PD-L1 *in vitro* and *in vivo*. Mechanistically, our results reveal that NAMPT inhibition decreases Sirt activity, and raises the acetylation level of the NF- κ B p65 K310 site which is critical for NF- κ B transcriptional activity (8, 9), and next contributes to the binding of NF- κ B p65 on CD274(PD-L1) promoter region, subsequently results in the upregulation of PD-L1. Our results suggested that NAMPT regulated PD-L1 via Sirt, providing an epigenetic perspective to explain the possible regulatory mechanism of PD-L1. Moreover, our study revealed another phenomenon, namely that in a microenvironment with limited NAD⁺ metabolism, which can decrease antitumor immunity through upregulation of PD-L1 in cancer cells. Our findings explained why targeting NAD⁺ metabolism alone has limited efficacy in preclinical and clinical studies (10) and suggested a new feasible strategy.

References

1. Li C-W, Lim S-O, Xia W, Lee H-H, Chan L-C, Kuo C-W, *et al.* Glycosylation and stabilization of programmed death ligand-1 suppresses T-cell activity. *Nat Commun* **2016**;7:12632.
2. Yu Y, Liang Y, Li D, Wang L, Liang Z, Chen Y, *et al.* Glucose metabolism involved in PD-L1-mediated immune escape in the malignant kidney tumour microenvironment. *Cell Death Discov* **2021**;7(1):15.
3. Christofk HR, Vander Heiden MG, Harris MH, Ramanathan A, Gerszten RE, Wei R, *et al.* The M2 splice isoform of pyruvate kinase is important for cancer metabolism and tumour growth. *Nature* **2008**;452(7184):230-3.
4. Elf SE, Chen J. Targeting glucose metabolism in patients with cancer. *Cancer* **2014**;120(6):774-80.
5. Palsson-McDermott EM, Dyck L, Zaslona Z, Menon D, McGettrick AF, Mills KHG, *et al.* Pyruvate Kinase M2 Is Required for the Expression of the Immune Checkpoint PD-L1 in Immune Cells and Tumors. *Front Immunol* **2017**;8:1300.
6. Wang S, Yao F, Lu X, Li Q, Su Z, Lee J-H, *et al.* Temozolomide promotes immune escape of GBM cells via upregulating PD-L1. *Am J Cancer Res* **2019**;9(6):1161-71.
7. Lv H, Lv G, Chen C, Zong Q, Jiang G, Ye D, Cui X, He Y, Xiang W, Han Q, Tang L, Yang W, Wang H. NAD⁺ Metabolism Maintains Inducible PD-L1 Expression to Drive Tumor Immune Evasion. *Cell Metab.* **2021** Jan 5;33(1):110-127.
8. Yeung F, Hoberg JE, Ramsey CS, Keller MD, Jones DR, Frye RA, Mayo MW. Modulation of NF-kappaB-dependent transcription and cell survival by the SIRT1 deacetylase. *EMBO J.* **2004** Jun 16;23(12):2369-80.
9. Chen LF, Mu Y, Greene WC. Acetylation of RelA at discrete sites regulates distinct nuclear functions of NF-kappaB. *EMBO J* **2002** Dec 2;21(23):6539-48.
10. Sampath D, Zabka TS, Misner DL, O'Brien T, Dragovich PS. Inhibition of nicotinamide phosphoribosyltransferase (NAMPT) as a therapeutic strategy in cancer. *Pharmacol Ther* **2015**;151:16-31.

5. <Comments>

: The "Cell lines and culture" section mentions only five cell lines, yet Figure 1C utilizes eleven. It would be beneficial to include information on the missing cell lines.

<Response>

According to the reviewer's suggestion, we added information on the missing cell lines in the current manuscript (*see Materials and Methods*).

→

Mouse colon cancer cell line CT-26, mouse Lewis lung cancer cell line (LLC), mouse lymphatic cancer cell line A20, human ovarian cancer cell line A2780, and human colon cancer cell line HCT-116, human non-small cell lung cancer cell line A549, H1299, and PC9, human gastric cancer cell line MKN74, human liver cancer cell line

HepG2 and Hep3B, human breast cancer cell line MDA-MB-231, were purchased from the ATCC (Manassas, VA, USA). CT-26, A20, HCT-116, A549, H1299, PC9, MKN74, and MDA-MB-231 cell lines were cultured in RPMI 1640 medium (Gibco, USA) containing 10% fetal bovine serum (Gibco, USA), supplemented with 10% fetal bovine serum (Gibco, USA) and 1% penicillin-streptomycin (Gibco, USA). LLC, A2780, HepG2, and Hep3B cells were cultured in DMEM medium containing 10% fetal bovine serum (FBS) (Gibco, USA) with 10% fetal bovine serum (Gibco, USA) and 1% penicillin-streptomycin (Gibco, USA) at 37 °C in a 5% CO₂ incubator.

6. <Comments>

: For LZFPN-90, as a novel compound, crucial details are absent, including synthesis methods, pharmacokinetics, pharmacodynamics, and toxicity data.

<Response>

Thank you for your insightful comments and professional recommendations.

1) Synthesis methods: The synthetic route of LZFPN-90 was presented in Scheme 1, and the detailed synthetic procedure was provided as well. All these contents have been added in the article Supporting Information of the revised manuscript. Commencing with commercially available 4-bromoindole, the Gribble indole reduction reaction was performed to yield intermediate **1**. Subsequent amidation of **1** with 5-(1,3-dioxolane-2-yl)thiazole-2-carboxylic acid generated intermediate **2**, from which the deprotection of the acetal group afforded aldehyde **3** in exceptional yield. Aldehyde **3** was then reduced to alcohol **4** using NaBH₄, which underwent a Suzuki-Miyaura coupling reaction with commercially available (3-hydroxyphenyl) boronic acid to yield intermediate **5**. Further transformations from **5** included successive nucleophilic substitution reactions with 1-bromo-3-chloropropane and diethylamine, generating intermediate **7**, which was converted to **8** by the reaction with thionyl chloride. The conversion of chlorine **8** to primary amine **10** was then achieved through the classical Gabriel reaction involving nucleophilic substitution with phthalimide and hydrazinolysis using hydrazine hydrate. Eventually, the target compound LZFPN-90 was obtained by reacting intermediate **10** with methyl

(Z)-N'-cyano-N-(pyridine-4-yl) carbamimidothioate. The chemical structure of LZFPN-90 has been confirmed by mass spectra, ¹H NMR spectra, and ¹³C NMR spectra.

2) Pharmacokinetics: Human liver microsomal stability assay is widely adopted in the preliminary evaluation for pharmacokinetic properties, reflecting the potential metabolic stability of indicated compounds in the human body. Thus, considering the reviewer's suggestion as well as the potent immunoregulatory activity and antitumor activity of LZFPN-90, it was further evaluated for potential metabolism in human liver microsomes. The intrinsic clearance for LZFPN-90 in human liver microsomes is 0.035 mL/min/mg. According to the classical criteria (Low clearance: $CL_{int} < 0.01$ mL/min/mg; medium clearance: $0.01 \leq CL_{int} \leq 0.1$ mL/min/mg; high clearance: $CL_{int} > 0.1$ mL/min/mg.), LZFPN-90 possessed moderate stability in human liver microsomes, indicating its potential for further development. The related data has been added in the revised manuscript.

3) Pharmacodynamics: Through rational compound design and structural optimization, LZFPN-90 was identified as a dual inhibitor of PD-L1 and NAMPT. The pharmacodynamic properties of LZFPN-90 have been comprehensively evaluated. The inhibitory activities of LZFPN-90 against the PD-1/PD-L1 interaction and NAMPT were tested at the biochemical level. The results demonstrated that LZFPN-90 could simultaneously and potently inhibit the PD-1/PD-L1 interaction and NAMPT, with IC_{50} values of 0.013 and 0.028 μ M, respectively. The discovery process of LZFPN-90 and the key biochemical data have been presented in Figure S1. The experimental methods for biochemical assays were described in the "**Materials and Methods**" section of the manuscript.

4) Toxicity profile: Body weight and organ index are widely used to evaluate gross toxicity of drug administration. Here, we evaluated the gross toxicity of LZFPN-90 in vivo using the body weight and organ index of mice in two models, and the results showed that LZFPN-90 did not affect the body weight and organ index of mice in two animal models (Fig 9 A-D). Thus, based on our preliminary results, LZFPN-90 might have good safety characteristics.

Scheme 1. The synthesis of compound LZFPN-90. Reagents and conditions: (a) NaBH_3CN , AcOH , r.t.; (b) 5-(1,3-dioxolane-2-yl)thiazole-2-carboxylic acid, HATU, DIPEA, DMF, r.t.; (c) p -TsOH· H_2O , acetone/ H_2O (4:1), reflux; (d) NaBH_4 , $\text{CH}_2\text{Cl}_2/\text{EtOH}$ (1:1), r.t.; (e) (3-hydroxyphenyl) boronic acid, PdCl_2 (dppf), K_2CO_3 , 1,4-dioxane/ H_2O (4:1), 70 °C; (f) $\text{Cl}(\text{CH}_2)_3\text{Br}$, Cs_2CO_3 , DMF, 50 °C; (g) diethylamine, NaI , K_2CO_3 , DMF, 80 °C; (h) SOCl_2 , MeCN, r.t.; (i) phthalimide, K_2CO_3 , DMF, 40 °C; (j) $\text{N}_2\text{H}_4\cdot\text{H}_2\text{O}$, EtOH, reflux; (k) methyl (Z)- N' -cyano- N -(pyridin-4-yl) carbamimidothioate, DMAP, Et_3N , pyridine, 70 °C.

Table 1. *In Vitro* Metabolic Stability of LZFPN-90 in Human Liver Microsome

Compound	$T_{1/2}$ (min) ^a	CL_{int} , in vitro (mL/min/mg) ^b	Remaining (T = 60 min)
LZFPN-90	39.6	0.035	36.8%

^a $T_{1/2}$ is the half-life.

^b CL_{int} : intrinsic clearance, $\text{CL}_{\text{int}(\text{mic})} = 0.693/T_{1/2}/\text{mg}$ microsomal protein/mL

Fig 9: Effects of LZFPN-90 on body weight and organ index of tumor-bearing mice.

(A-B). Mouse body weight and organ index of A20 model mice. Balb/c mice bearing A20-derived tumors were treated with CQ-85 (20 mg/kg, i.p.), FK866 (20 mg/kg, i.p.), LZ90 (20 mg/kg, 40 mg/kg, i.p.), n=6. (C-D). Mouse body weight and organ index of LLC OE NAMPT/PD-L1 model mice. C57BL/6 mice bearing LLC OE NAMPT/PD-L1-derived tumors were treated with CQ-85 (20 mg/kg, i.p.), FK866 (20 mg/kg, i.p.), LZ90 (20 mg/kg, i.p.), n=6. Data are presented as mean \pm SEM; * P < 0.05, ** P < 0.01, *** P < 0.001, compared with control. n.s., not significant.

7. <Comments>

: Concerning *in-vitro* studies, I would suggest that the authors offer more specific information on the experimental methodology, such as the duration of drug incubation.

<Response>

According to the suggestions of reviewers, we supplemented and improved the experimental method (*see Materials and Methods*).

Specific information about the drug has been supplemented in the experimental methodology and figure legend, including the duration of drug incubation, concentration, action cells, etc.

→

Cell viability assay

In vitro cell viability was determined using a Cell Counting Kit-8 (MCE) assay. Cells (5×10^3 /well) were seeded in 96-well culture plates. After 24 h, they were incubated with various concentrations of the test compounds **for 48-72 h** at 37 °C in a 5% CO₂ incubator. Following Next, 10% of the CCK-8 solution was added to each well, and the plates were incubated for an additional 4 h at 37 °C. The optical density of each well was measured at 450 nm using a multimode plate reader (Molecular Devices, San Jose, CA, USA).

MTT assay

Cells (5×10^3 /well) were seeded into a 96-well plate and cultured at 37 °C in 5% CO₂ incubator. The various concentrations of FK866 or LZFPN-90, ranging from 0.25 nM to 256 nM, were administrated to cells for **48 h**, with a minimum of three technical replicates. At the final time point, 10 µl MTT solution (5 mg/ml, Sigma, Burlington, MA, USA) was added to each well and incubated for an additional 4 h at 37 °C in a 5% CO₂ incubator. Then, 100 µl of DMSO was added to each well and mixed thoroughly. Optical density was measured at 492 nm using a microplate reader (Molecular Devices, San Jose, CA, USA).

8. <Comments>

: To conclusively show that LZFPN-90 functions as a NAMPT inhibitor, NAMPT Activity Assays could provide direct evidence. I recommend conducting such an experiment for robust validation.

<Response>

We appreciate your insightful comments and professional advice. In line with your comment, the inhibitory activity of LZFPN-90 against NAMPT was examined at the biochemical level. The biological data indicated that LZFPN-90 could potently inhibit NAMPT, with an IC₅₀ value of 0.028 µM. The results of the biochemical assay as well as the discovery of LZFPN-90 are shown in Figure 10 A and B. To further confirm its inhibitor action on NAMPT, we also used the NAD⁺ assay kit to detect the NAD⁺

activity after treatment with LZFPN-90. Our results have shown that LZFPN-90 reduced the activity of NAD⁺ in a dose-dependent manner, and the NAD⁺ activity inhibitory action of LZFPN-90 was reversed after the addition of NMN (Fig. 10C). The above results demonstrate that LZFPN-90 is a NAMPT competitive inhibitor. The brief experimental procedure was also described in the “*Materials and Methods*” section of the manuscript.

→

NAMPT enzymatic activity was determined by a NAMPT Activity Assay Kit (catalog no: #ab221819, Abcam) following the manufacturer’s protocols. The assay is based on a multistep reaction that converts WST-1 to WST-1 formazan, which can be easily detected at an absorbance of 450 nm using a Tecan Infinite M1000 microplate reader (1). The fluorescent intensity data were analyzed using the computer software, Graphpad Prism.

A

Table 1: The influence of LZ90 to NAMPT enzyme activity

LZ90	NAMPTFluorescence		% Activity	
	Repeat1	Repeat2	Repeat1	Repeat2
No compound	2231	2265		
4 nM	2114	2099	94	93
8 nM	1758	1642	77	71
16 nM	1359	1265	58	53
31 nM	1228	1142	51	47
63 nM	749	784	29	30
125 nM	702	632	26	23
250 nM	544	456	19	15
500 nM	244	221	5	4
1000 nM	195	201	2	3
Background	134	157		

Table 2: NAMPT enzyme results table

	Repeat 1 (nM)	Repeat 2 (nM)	Average (nM)	SD
LZ90	30.2	25.3	27.8	3.5
FK-866	22.9	16.8	19.9	4.3

B

C

Fig 10: Effect of LZFPN-90 on NAMPT and NAD⁺ activity.

(A). **Table 1:** The influence of LZFPN-90 on NAMPT enzyme activity. **Table 2:** NAMPT enzyme results table. (B) The curve of LZFPN-90 inhibits NAMPT activity. (C) The NAD⁺ content of A2780 cells was treated with different concentrations of LZFPN-90 and NMN.

Reference

(1) Jin X, Li X, Li L, Zhong B, Hong Y, Niu J, Li B. Glucose-6-phosphate dehydrogenase exerts antistress effects independently of its enzymatic activity. J Biol Chem. 2022 Dec;298(12):102587.

9. <Comments>

:Figures 4B and 4C indicate that LZFPN-90 promotes T cell proliferation and CD8+ T cell activation. Could the authors explain these observations? Prior studies, such as Bruzzone et al. 2009 (PMID: 19936064), suggest that NAMPT inhibition has a detrimental effect on activated T cells. This observation is in line with one of the side effects "lymphopenia" of NAMPT inhibitors. lymphocytopenia is a consequence of NAMPT inhibition.

<Response>

Previous studies, such as Bruzzone et al. 2009 (PMID: 19936064), demonstrate that NAMPT inhibitor FK866 could interfere with the survival of activated T cells and lead to the release of immunogenic cytokines, including IFN- γ , TNF- α , and IL-6. In our co-cultured condition, treatment of dual-target inhibitor (NAMPT and PD-L1) LZFPN-90 could activate T cells, which seems to conflict with the previous finding. In fact, this inconsistency could be explained by the following fact:

1) **Under co-culture conditions, NAMPT inhibition may preferentially affect tumor cells because tumor cells express more NAMPT than T cells.** To address this issue, we first detected the NAMPT level of tumor cells and T cells and found that the expression of NAMPT in tumor cells was higher than that in T cells (Fig. 11A). In addition, we also analyzed NAMPT expression levels in different tumor tissue and normal tissue according to open access database (Fig. 11B), and results showed that tumor tissue showed a higher level of NAMPT than normal tissue. Thus, we hypothesized that, under co-culture conditions, NAMPT inhibition may preferentially affect tumor cells because tumor cells expressed more NAMPT than T

cells.

Fig 11: Protein expression of NAMPT in LLC cells and T cells.

(A) . After T cells and LLC cells were co-cultured for 48 h and 72 h, the expression of NAMPT was detected by western blot. (B). Analysis of NAMPT and PD-L1 mRNA levels in various cancer and para-cancerous tissues from TCGA data.

2) Inhibition of NAMPT would contribute to the up-regulation of PD-L1, which might promote the activation of T cells by LZFPN-90. In the tumor immune microenvironment, the upregulated PD-L1 would result in the inactivation of T cells and the reduction of immunogenic cytokines. Simultaneously, the upregulated PD-L1 also can contribute to the enhanced efficacy of PD-L1 inhibitors. In the current study, our data indicates that inhibition of NAMPT leads to upregulation of PD-L1. Under the condition of co-culture of tumor cells and T cells, tumor cells showed the increased expression of PD-L1, which would result in the enhanced sensitivity to NAMPT/PD-L1 inhibitor LZFPN-90, subsequently leading to activation of T cells. Therefore, we hypothesize that the NAMPT/PD-L1 inhibitor might be superior to the NAMPT inhibitor for side effects, such as lymphocytopenia.

In order to address this comment, the above response points have been added to the current manuscript (discussion section).

10.<Comments>

: As tumor cell lines differ from primary cancer cells, I would recommend the authors to perform the same in vitro studies, using primary tumor cells from patients, if possible.

<Response>

According to the reviewer's suggestion, we perform the same in vitro studies using primary tumor cells from patients. The results were shown in the following:

1) Organoid model construct: Organoids are self-organizing 3D structures grown from stem cells that recapitulate essential aspects of organ structure and function. In our study, we evaluated the pharmacodynamics of LZFPN-90 using lung cancer organoids derived from lung cancer patients themselves *in vitro*. When the length of the organoids was about 50 μ m and obvious 3D cell spheres, it was considered that the organoids had been formed (Fig.12 A).

2) Effect of LZFPN-90 on organoid-self: After the addition of 100 nM LZFPN-90 for 3 days, the diameter of the organoids did not increase significantly compared with the control group. After 6 days of treatment, it was found that the diameter of the

organoids in the LZFPN-90 group decreased, while the control group continued to increase. After the 9 days of treatment, compared with the control group, the diameter of the organoids in the LZFPN-90 group decreased significantly and there were fragments at the edges. These results indicated that LZFPN-90 could significantly inhibit organoid growth (Fig.12 A).

3) Effect of LZFPN-90 on T cell infiltration to organoid by immunofluorescent imaging: Fluorescent dye CFSE can penetrate cell membranes, and is widely used as a cell marker. We first labeled PBMC cells with CFSE before co-culture, and then co-culture CFSE pre-labeled T cells with organoids. On days 6 and 9 of co-culture, we used fluorescence microscopy to detect the number of T cells penetrating the matrix glue covering the organoids. In the LZFPN-90 group, we found that the number of T cells penetrating was significantly increased than the control group. This indicates that LZFPN-90 can promote the invasion of T cells (Fig. 12B).

4) Effect of LZFPN-90 on T cell infiltration and killing activity in co-cultured conditions by bright-field imaging: At the beginning of organoid formation, human peripheral blood mononuclear cells (PBMC) were co-cultured with organoids. LZFPN-90(100 nM) was administrated in this co-cultured condition. After 3 days of co-culture, T-cell infiltration was observed at the edge of organoids in the LZFPN-90 group and the control group, with more infiltration in the LZFPN-90 group. After 6 days of co-culture, the organoids were surrounded by T cells in the LZFPN-90 group. After 9 days of co-culture, the LZFPN-90 group organoid fragmentation was significantly observed compared with the control group, suggesting organoids might be killed by infiltrated T cells. This result indicated that LZFPN-90 significantly promoted T cell infiltration in primary tumor cells, and next enhanced the killing of tumor cells (Fig. 12C). (*see Supplementary Figure 6*)

Fig 12: Effect of LZFPN-90 in organoids derived from lung cancer.

(A) Representative bright-field images of the lung cancer organoid diameter treatment with 100 nM LZFPN-90 for 3, 6, and 9 days (scale bar 100 μ m). (B) Representative fluorescent images of PBMC infiltration during co-culture for 6 and 9 days (scale bar 25 μ m). (C) Representative bright-field images after PBMC co-cultured with organoids treatment with 100 nM LZFPN-90 for 3, 6, and 9 days (scale bar 100 μ m).

Response to Reviewer 3

Point-by-point reply to reviewers' comments

Reviewer 1:

In this manuscript, the author investigated the reciprocal feedback regulation between NAMPT and PD-L1 and designed a dual-targeting inhibitor LZFPN-90 to disrupt PD-1/PD-L1 interactions and inhibit NAMPT activity, thereby enhancing T-cell mediated antitumor immunity. The observation of a negative correlation between the expression of NAMPT and PD-L1 in various cancer cell lines is novel, and the dual inhibitor exhibited strong inhibition to both targets. However, one major issue, is how SIRT epigenetic regulation can affect the expression of PD-L1, and Vice versa. Possible mechanisms are not investigated and discussed. In addition, there are a few other major issues that need to be clarified or verified.

<Response>

However, one major issue, how SIRT epigenetic regulation can affect the expression of PD-L1, and Vice versa? Possible mechanisms are not investigated and discussed.

: Thanks to the reviewer for his/her valuable suggestions. According to the reviewer's comment, we further explored the underlying regulation mechanism between NAMPT and PD-L1. The results demonstrate:

1) NAMPT epigenetic regulates PD-L1 by inhibiting the deacetylation NF- κ B p65 K310, a non-histone substrate of SIRT.

Our results reveal that NAMPT inhibition would result in the upregulation of PD-L1 *in vitro* and *in vivo*. Mechanistically, our data demonstrates that NAMPT inhibition decreases SIRT activity and raises the acetylation level of NF- κ B p65 K310 site which is critical for NF- κ B transcriptional activity, and next contributes to binding of NF- κ B p65 on CD274 (PD-L1) promoter region, subsequently results in the upregulation of PD-L1 (Figure 13). The treatment of NMN, a NAD⁺ precursor, could reverse the above epigenetic regulation, which further confirmed the crucial role of NAMPT in PD-L1 regulation.

Figure 13: NAMPT inhibition promoting PD-L1 expression mediated by the epigenetic regulation of SIRT (Graphics are drawn by Figdraw)

2) PD-L1 inhibition inducing NAMPT expression through HIF-1 α /glycolysis pathway.

We found that inhibition of PD-L1 also could induce the expression of NAMPT in the presence of T cells. To be specific, inhibition of the PD-1/PD-L1 axis could activate the T cells immunity, promoting the release of IFN- γ , TNF- α and other inflammatory factors, and upregulating the expression of HIF-1 α under the normal oxygen state. The upregulation of HIF-1 α will increase the expression of downstream glycolytic related genes and the content of pyruvate, the product of glycolysis, thus promoting the consumption of NAD⁺ and inducing the expression of NAMPT to provide the high demand for NAD⁺ (Figure 14). Notably, when we knockout HIF-1 α , the above regulatory relationship is disappeared. The preliminary findings of our study indicates that the immune response exerts a certain influence on tumor metabolism, and HIF-1 α plays a crucial role as a regulatory factor in this process.

Figure 14: PD-L1 inhibition inducing NAMPT expression through HIF-1 α /glycolysis pathway.

(Graphics are drawn by Figdraw)

The above finding has been added in the results and discussion section in the current manuscript.

1.<Comments>

: The author used FK866 and CQ-85 to inhibit NAMPT and PD-L1, respectively, and observed negative correlation between the expression of NAMPT and PD-L1. To exclude off target possibilities of these small molecular inhibitors, genetic knock down/out, or overexpression of the two genes individually, are needed to further confirm the correlation between the expression of NAMPT and PD-L1.

<Response>

We appreciate your insightful comments and professional advice. To exclude off target possibilities of these small molecular inhibitors, we individually knocked down NAMPT and PD-L1 in LLC cells and detected the expression of NAMPT and PD-L1. As shown in Figure 15A, there was an increased protein expression of PD-L1 in NAMPT knock-down cells. Similarly, in co-culture condition, an increased protein expression of NAMPT was observed in PD-L1 knockdown cells (Fig. 15 B). This is consistent with the results of small molecule inhibitor treatment, indicating that there is a negative correlation between the expression of NAMPT and PD-L1.

Figure 15: Effect of Knocking down NAMPT and PD-L1 on LLC cells.

(A). NAMPT was knocked down by siRNA in the LLC cell and the protein expressions of NAMPT and PD-L1 were detected in LLC si Ctrl and si NAMPT cells by western blot. (B) Under co-culture conditions, PD-L1 was knocked down by siRNA in the LLC cell, and the protein expressions of NAMPT and PD-L1 were detected in LLC si Ctrl and si PD-L1 cells by western blot.

2. <Comments>

: Fig. 2C, the relative band intensity plots of LZFPN-90 do not fit into the curve at all!

<Response>

According to the reviewer's comment, we re-analyzed the band average intensity and selected the new representative blot picture in the current manuscript (see Figure 16).

We are sorry about this carelessness.

Fig 16: Effect of LZFPN-90 on the thermal shift.

A2780 cells were treated with or without LZFPN-90 (20 nM). Lysates were prepared and heated to the indicated temperature for 3 minutes and left at room temperature for 3 minutes, then subjected to immunoblot analysis.

3. <Comments>

: In Figure 3C, compound LZFPN-90 basically had not effect on the relative viability of LLC cells at 64 nM, but in Figure 3E, LZFPN-90 could significantly induce apoptosis of LLC cells at 60 nM. The authors need to explain the inconsistency of the LZFPN-90 compound to LLC cells.

<Response>

Thanks to the reviewer for his/her valuable suggestions. The reason for this “conflict”

could be explained by the following fact.

1) Different treatment times in both experiments: In the MTT assay, compound LZFPN-90, treatment for 48 h at 64 nM, basically had no effect on the relative viability of LLC cells (Fig. 17A). However, in the apoptosis study, LZFPN-90 treatment for 72 h at 60 nM, could significantly induce LLC cell apoptosis (Fig. 17B). As we known, prolonged treatment time of drug could result in the enhancement of drug action. Therefore, we think the “conflict” was caused by different treatment time.

2) Same treatment time got consistent results: To further addressed this issue, we also performed MTT and apoptosis assay at the same treatment time. As shown in Figure 17A-B, the changes in inhibiting cell proliferation and promoting apoptosis induced by about 60nM LZFPN-90 were consistent under the same treatment time. As for 48h, we could see that the survival rate of LLC cells was basically not affected, and apoptosis was also not significantly induced. However, after treatment of LZFPN-90 for 72h, the survival rate of LLC cells was inhibited at 64 nM concentration of LZFPN-90, and the LLC cells apoptosis was significantly induced at similar concentrations.

At last, to avoid this misunderstanding, we added treatment time in both experiments in the current revised manuscript.

Figure 17: Effects of LZFPN-90 on proliferation and apoptosis of LLC cells at different time points.

(A) Effect of treatment with different concentrations of LZFPN-90 for 48 h and 72 h on the proliferation of LLC Vector cells. (B) Apoptotic ratio of treatment with different concentrations of LZFPN-90 for 48 h and 72 h in LLC cells. Data are presented as mean \pm SEM; * P < 0.05, ** P < 0.01, *** P < 0.001, compared with control.

4. <Comments>

: In results 3.1, 1) the authors stated they constructed a tumor model in which mice were inoculated with LLC cells. But in Fig 1I, it seems they used LLC cells with double over-expression of NAMPT and PD-L1. Which cell did you use exactly? 2) In addition, I cannot tell from fig 1I that whether the two proteins are successfully overexpressed.

<Response>

1) the authors stated they constructed a tumor model in which mice were inoculated with LLC cells. But in fig 1I, it seems they used LLC cells with double over-expression of NAMPT and PD-L1. Which cell did you use exactly?

Thanks to the reviewer for his/her valuable suggestions. In fact, we used three mouse models in the present study, including the A20 xenograft model, and the LLC xenograft model, and constructed the LLC (Overexpressed PD-L1/NAMPT) xenograft model. In Fig 1I, we used constructed LLC (Overexpressed PD-L1/NAMPT) tumors to detect the expression changes of PD-L1 and NAMPT after being treated with FK866 and CQ-85, respectively. **To eliminate this misunderstanding, we have revised the description of the legend in Figure 1I in the current manuscript.**

2) In addition, I cannot tell from fig 1I that whether the two proteins are successfully overexpressed.

To increase the clarity of the experimental results, we conducted protein detection in tissues and found that the protein expressions of NAMPT and PD-L1 were increased, indicating that our LLC OE NAMPT/PD-L1 mice model was successfully constructed (Fig. 18A). It should be stated that, in order to clearly show a change tendency in Figure 1I, we used short exposure time in the western blot experiment. Thus, the blot results of PD-L1 and NAMPT in the LLC OE NAMPT/PD-L1 control group look a little bit weak.

Fig 18: The expression of NAMPT and PD-L1 in LLC Vector and LLC OE NAMPT/PD-L1 xenograft model tumor tissues.

(A) The expression of NAMPT and PD-L1 in LLC Vector and LLC double over-expression of NAMPT and PD-L1 xenograft model tumor tissue. (B) Relative expression of PD-L1 and NAMPT in FK866 (20 mg/kg) and CQ-85(20 mg/kg) were detected from LLC OE NAMPT/PD-L1 xenograft model tumor tissue. (n = 4).

5. <Comments>

: In T cell-mediated tumor cell killing assay, PBMCs were cultured at the beginning, and T cells were used in the later experiment. Is there a sorting step in this process?

<Response>

1) Two types of immune cells are used in our system. PBMC is used in human tumor cell co-culture, most of the PBMC is made up of lymphocytes. Lymphocytes are needed from the same species for experiments. In our study, we mainly used human cancer cells A2780 and HCT-116, which need to extract human peripheral blood mononuclear cells (PBMC) for the experiment. To maintain its function, we have carried out the following processing: according to the manufacturer's protocol, the experiments were performed with anti-human CD3 antibody (100 ng/mL, Thermo Scientific) to activate the PBMC. Then, A2780 or HCT-116 cells were co-cultured with activated PBMC cells for 72 h.

Extraction of PBMC:

2) The other cancer cells were LLC cells, which were derived from C57BL/6 mice. We need to extract and activate **T cells** from the spleen of C57BL/6 mice for co-culture. To acquire activated T cells, we used anti-mouse CD3/CD28 antibodies (STEMCELL Technologies) to activate the T cells for 48-72h. Then LLC cells were co-cultured with activated T cells for 72 h.

Extraction of T cells:

At last, to avoid this misunderstanding, we added the information in the current revised manuscript.

6. <Comments>

: In the discussion section, the authors explain the necessity of double-target NAMPT and PD-L1 in a redundant manner. The author should reframe the way of writing. how SIRT epigenetic regulation can affect the expression of PD-L1, and Vice versa should be discussed in detail.

<Response>

According to the reviewer's comment, we have revised the manuscript in two sections.

1) **Results section**, we added the new data and re-wrote this part; 2) **Discussion section**, we added the regulation mechanism between NAMPT and PD-L1.

→

Results

2.1. NAMPT inhibition promoting PD-L1 expression mediated by SIRT epigenetic regulation

To determine the regulatory relationship between NAMPT and PD-L1, we analyzed the expression of NAMPT and PD-L1 in the TCGA database. Our results showed that NAMPT and PD-L1 (CD274) are highly expressed in most cancers (Fig. 19A). We analyzed the relationship between co-expression levels and prognosis of NAMPT and PD-L1, finding that patients with high NAMPT expression and low PD-L1 expression had the worst prognosis (Fig. 19B). This observation suggests that the NAMPT^{High} PD-L1^{Low} expression pattern is significant in tumor progression. Consistent with TCGA data, we experimentally examined the expression of NAMPT and PD-L1 in 11 cell lines from different cancers and found that the expression levels of the two proteins were negatively correlated (Fig. 19C). Next, we investigated whether the pharmacological blockade of NAMPT upregulates PD-L1 protein levels. Our results indicated that FK866, an inhibitor of NAMPT, upregulated global and membrane PD-L1 expression in A2780 cells, whereas the upregulation of PD-L1 induced by FK866 was reversed by treatment with the NAD⁺ precursor NMN (Fig. 19D, E). In addition, we performed similar experiments by specific inhibitor FK866 or gene manipulation on murine-derived LLC cells and obtained consistent results (Fig. 19D, E, S1A). Importantly, the *in vivo* study also showed that the administration of FK866 could lead to the upregulation of PD-L1. The above results indicate that PD-L1 expression is upregulated when NAMPT was inhibited.

Sirt1/2 is a well-known NAD⁺-dependent deacetylase, and NAD⁺ concentration can affect the activation of Sirt1 (1). Our results indicated that the pharmacological blockade of NAMPT upregulates PD-L1 levels. Therefore, we tested whether NAD⁺-deprivation-induced impairment of Sirt activity is

responsible for the upregulation of PD-L1 in cancer cells. We treated A2780 and LLC cells with Sirtinol, a specific SIRT1/2 inhibitor, and found that PD-L1 was upregulated at both the mRNA and protein levels (Fig. 19F, G). Notably, compared to the monotherapy groups, combined treatment with FK866 and Sirtinol resulted in an obvious increase in the protein expression of PD-L1 (Fig. 19H, **Fig. S1A**). To assess whether FK866 is involved in the regulation of Sirt activity, we also detected the expression of Ac-Tubulin, a substrate of Sirt, in LLC cells. Our results indicated that treatment of FK866 could increase Ac-Tubulin, while NMN reversed the increase of the Ac-Tubulin (**Fig. S1B**). The above results showed that the inhibition of NAMPT induced PD-L1 expression through Sirt. In order to further explore how NAMPT affects the expression of PD-L1 through Sirt, we examined the Sirt-regulated acetylation K310 of NF- κ B p65, which is considered as the transcription factor of PD-L1 (2-4). After treatment with FK866 and NMN to A2780 and LLC cells, we detected the acetylation level of p65-K310. The data revealed that FK866 increased the acetylation of p65-K310, while NMN reversed the event (Fig. 19I, **Fig S1D**). Next, we want to know whether NAMPT inhibition could affect the binding of p65-K310 on the PD-L1 promoter. According to the JASPAR database, the p65 binding site on the PD-L1 promoter was predicted (Fig. 19J). The results of ChIP-qPCR showed that after treatment with FK866 or Sirtinol, the enrichment p65-K310 on the P3 region, but not P1 and P2 region, was increased in A2780 cells, while NMN blocked this binding (Fig. 19J), which suggests NAMPT inhibition affects PD-L1 expression through regulating acetylation of NF- κ B p65 and its activity. Taken together, we found a negative relationship between NAMPT and PD-L1, and the promotion of PD-L1 expression after the treatment of NAMPT inhibition, which was mediated by the epigenetic regulation of Sirt, and subsequently resulted in transcription activation of PD-L1 by NF- κ B p65.

Figure 19. NAMPT inhibitor upregulates the mRNA levels of PD-L1 in mouse and human cells.

(A) Analysis of NAMPT and PD-L1 mRNA levels in various cancer and para-cancerous tissues from TCGA data. (B) Kaplan-Meier curve analysis of overall patient survival based on different NAMPT and PD-L1 expression levels in the TCGA database. (C) Correlation of the level of PD-L1 membrane protein with the level of NAMPT protein in different cancer cells. r = Pearson correlation coefficients; p = p values. (D) Western blot analysis of PD-L1 protein expression after adding FK866 to A2780 and LLC cells for 72 h (A2780: 0, 0.5, 1 μ M; LLC: 0, 2, 4 μ M). (E) Effects of FK866 and NMN for 72 h on PD-L1 levels in A2780 cells (FK866: 0.4 μ M; NMN: 1 mM) and LLC cells (FK866: 0.8 μ M; NMN, 1 mM), which were detected by flow cytometry. (F) Effects of Sirtinol for 72 h on the expression of PD-L1 in A2780 cells (10, 20 μ M) and LLC cells (10, 50 μ M), which were detected by flow cytometry. (G) Expression of PD-L1 mRNA after adding FK866 and Sirtinol to A2780 cells for 72 h (FK866:0.4 μ M; Sirtinol: 10 μ M) and LLC cells (FK866: 0.8 μ M; Sirtinol: 10 μ M). (H) Effect of FK866 and/or Sirtinol for 72h on PD-L1 levels in A2780 cells (FK866: 0.4 μ M; Sirtinol: 10 μ M) and LLC cells (FK866: 0.4 μ M; Sirtinol: 10 μ M), which were detected by flow cytometry. (I) Analysis of Ac-P65 (K310), PD-L1 protein expression after adding FK866 and NMN to A2780 and LLC cells for 72 h by western blot. (A2780: FK866 0.5 μ M, NMN 1 mM; LLC: FK866 2 μ M, NMN 1 mM). (J) After treatment with FK866, Sirtinol and NMN for 72 h in A2780 cells, the enrichment of Ac-p65 (K310) in PD-L1 promoter region (P1, P2, P3) was detected by Chip-qPCR (A2780: FK866

0.5 μ M, Sirtinol 10 μ M; NMN 1 mM). Data are presented as mean \pm SEM; *P < 0.05, **P < 0.01, ***P < 0.001 compared with the control group.

2.2 PD-L1 inhibition inducing expression of NAMPT by activating the glycolytic pathway in HIF-1 α dependent manner.

In order to further explore the relationship between NAMPT and PD-L1, we also detected NAMPT expression after treatment with CQ-85, a small-molecule inhibitor of PD-L1, in LLC constructed xenograft model. Interestingly, our results revealed that inhibition of PD-L1 can upregulate the protein expression of NAMPT compared with the control group (Fig. 20A). The upregulation of NAMPT induced by PD-L1 inhibition was further verified by various PD-L1 inhibitors, including CQ-85 and BMS-202, and gene manipulation *in vitro* (Fig. 20B, C). Notably, the induction of NAMPT by PD-L1 inhibition was dependent on the presence of T cells (Fig. 20B), suggesting the regulation of PD-L1 and NAMPT had immune microenvironment-dependent characteristics.

It has been reported that the presence of inflammatory factors IFN- γ and TNF- α can activate the NAD⁺ dependent glycolytic pathway by upregulating the expression of HIF-1 α (5). We hypothesized that PD-L1 inhibition will increase the expression of NAMPT by promoting inflammatory factors release of T cells. Thus, we first assess whether inflammatory factors IFN- γ and TNF- α can regulate NAMPT expression. Our results indicated that the exogenous addition of IFN- γ and TNF- α , without T cells, could induce expression of NAMPT in LLC cells (Fig. 20B). Then, we explored whether PD-L1 inhibitors could regulate the secretion of IFN- γ and TNF- α . The ELISA results showed that treatment with PD-L1 inhibitors would promote the secretion of IFN- γ and TNF- α in tumor tissue and *in vitro* co-culture system (Fig 20D). Next, the relationship between PD-L1 inhibition and HIF-1 α and glycolysis was further explored. Our results showed that PD-L1 inhibitors could upregulate the expression of HIF-1 α under the co-culture with T cells, as well as the exogenous addition of IFN- γ and TNF- α (Fig 20E). At the same time, the results indicated that treatment with PD-L1 inhibitors also could increase the mRNA levels of

glycolytic related HIF-1 α target genes (Fig 20F), raise the contents of pyruvate and consumption of NAD⁺ under the co-culture condition (Fig 20G, H). Furthermore, the crucial role of HIF-1 α in this process was confirmed by gene manipulation. Compared with wildtype A549 cells, PD-L1 inhibitors induced changes, including the expression of NAMPT and glycolytic-related genes, and the contents of pyruvate and NAD⁺ (Fig 20 I-K) disappeared in HIF-1 α knock-out A549 cells. Taken together, these results indicated that PD-L1 inhibition can upregulate the expression of HIF-1 α by increasing the level of inflammatory factors, activating the glycolytic pathway promoting the consumption of NAD⁺, and subsequently inducing NAMPT expression.

Fig 20: Effects of PD-L1 inhibition on NAMPT under co-culture conditions.

(A) Relative expression of NAMPT in tumors from Ctrl and CQ-85 (20 mg/kg) groups in which mice were inoculated with LLC cells with double over-expression of NAMPT and PD-L1 (n = 4). (B) With or without the T cells system, western blot analyzed the NAMPT protein expression after adding CQ-85 (0.1 μ M), BMS202 (0.1 μ M), IFN- γ and TNF- α (50 ng/ml each) for 72 h in LLC cells. (C) After co-culture, the western blot analyzed the NAMPT protein expression after using si Ctrl and si PD-L1 in LLC cells. (D) The levels of IFN- γ and TNF- α were detected in tumor tissues of LLC OE NAMPT/PD-L1 mouse models by ELISA (left). The contents of IFN- γ and TNF- α in LLC cells were detected by ELISA under the co-culture condition (right). (E) With or without the T cells system, the expression of HIF-1 α was detected by flow cytometry after adding CQ-85 (0.1 μ M), BMS202 (0.1 μ M), IFN- γ and TNF- α (50 ng/ml each) for 72 h to LLC cells. (F) With or without the T cells system, the mRNA levels of several glycolytic genes were tested in LLC cells by RT-PCR. (G-H) With or without the T cells system, the content of pyruvate (PA) (G) and NAD⁺ (H) was detected in LLC cells. (I) With or without PBMC cells system,

the expression of HIF-1 α was detected by flow cytometry after adding CQ-85 (0.1 μ M), BMS202(0.1 μ M), IFN- γ and TNF- α (50 ng/ml each) for 72 h to A549 cells (left). Western blot verified the A549 knocks out HIF-1 α cells (middle). Co-cultured with T cells, western blot analyzed the NAMPT protein expression after adding CQ-85 (0.1 μ M), BMS202 (0.1 μ M), IFN- γ and TNF- α (50 ng/ml each) for 72 h in A549 and A549 KO HIF-1 α cells (right). (J) Heat map represented the expression of several glycolytic genes in A549 and A549 KO HIF-1 α cells co-cultured with PBMC cells by RT-PCR. (K) With or without PBMC cells system, the content of pyruvate (PA) and NAD⁺ was detected in A549 and A549 KO HIF-1 α cells. Data are presented as mean \pm SEM; *P < 0.05, **P < 0.01, ***P < 0.001 compared with the control group.

Discussion

→

The regulation of PD-L1 expression is a complex and highly coordinated series of molecular signaling events. In tumor cells, PD-L1 expression may be related to metabolism, which in turn affects the immune status of the TME and even antitumor immunity. Studies have shown that tumors stabilized PD-L1 expression to control glucose metabolism by enhancing N-glycosylation and the EGFR/ERK/c-Jun pathways (6,7). In addition, several studies have revealed that PKM2, which converted phosphoenolpyruvate into pyruvate, was highly expressed in tumors (8,9) and enhanced PD-L1 expression in both tumor and immune cells via epigenetic mechanisms (10). Furthermore, Wang S *et al* reported that NAD⁺ depletion and autophagy induced by NAMPT inhibitors mediate the upregulation of PD-L1 transcripts and cell surface protein levels in glioblastoma cells (11). A recent study demonstrated study revealed that PD-L1 expression also could be regulated by IFN- γ driven by NAMPT in a CD8⁺ T cell-dependent manner, suggesting the possible role of TME in this process (12). Our study found the same phenomenon: NAMPT inhibition would result in the upregulation of PD-L1 *in vitro* and *in vivo*. Mechanistically, our results reveal that NAMPT inhibition decreases Sirt activity, and raises the acetylation level of the NF- κ B p65 K310 site which is critical for NF- κ B transcriptional activity (13, 14), and next contributes to the binding of NF- κ B p65 on CD274 (PD-L1) promoter region, subsequently results in the upregulation of PD-L1. Our results suggested that NAMPT regulated PD-L1 via Sirt, providing an epigenetic perspective to explain the possible regulatory mechanism of PD-L1. Moreover, our study revealed

another phenomenon, namely that in a microenvironment with limited NAD⁺ metabolism, which can decrease antitumor immunity through upregulation of PD-L1 in cancer cells. Our findings explained why targeting NAD⁺ metabolism alone has limited efficacy in preclinical and clinical studies (15) and suggested a new feasible strategy.

It has been recognized that metabolism and the immune system interact with each other during cancer development and progression (16). Here, we found that inhibition of PD-L1 also could induce the expression of NAMPT in the presence of T cells. To be specific, inhibition of the PD-1/PD-L1 axis could activate the T cells immunity, promoting the release of IFN- γ , TNF- α , and other inflammatory factors, and upregulating the expression of HIF-1 α under the normal oxygen state. The upregulation of HIF-1 α will increase the expression of downstream glycolytic-related genes and the content of pyruvate, the product of glycolysis, thus promoting the consumption of NAD⁺ and inducing the expression of NAMPT to provide the high demand for NAD⁺. The preliminary findings of our study indicate that the immune response exerts a certain influence on tumor metabolism, and HIF-1 α plays a crucial role as a regulatory factor in this process.

References

1. Zhao Y, Zhang J, Zheng Y, Zhang Y, Zhang XJ, Wang H, *et al.* NAD⁺ improves cognitive function and reduces neuroinflammation by ameliorating mitochondrial damage and decreasing ROS production in chronic cerebral hypoperfusion models through Sirt1/PGC-1 α pathway. *J Neuroinflammation* **2021**;18(1):207.
2. Qin M, Cao Q, Wu X, Liu C, Zheng S, Xie H, *et al.* Discovery of the programmed cell death-1/programmed cell death-ligand 1 interaction inhibitors bearing an indoline scaffold. *Eur J Med Chem* **2020**;186:111856.
3. Qin M, Meng Y, Yang H, Liu L, Zhang H, Wang S, *et al.* Discovery of 4-Arylindolines Containing a Thiazole Moiety as Potential Antitumor Agents Inhibiting the Programmed Cell Death-1/Programmed Cell Death-Ligand 1 Interaction. *J Med Chem* **2021**;64(9):5519-34.
4. Pan C, Yang H, Lu Y, Hu S, Wu Y, He Q, *et al.* Recent advance of peptide-based molecules and nonpeptidic small-molecules modulating PD-1/PD-L1 protein-protein interaction or targeting PD-L1 protein degradation. *Eur J Med Chem* **2021**;213:113170.
5. Lai F, Ji M, Huang L, Wang Y, Xue N, Du T, *et al.* YPD-30, a prodrug of YPD-29B, is an oral small-molecule inhibitor targeting PD-L1 for the treatment of human cancer. *Acta Pharm Sin B* **2022**;12(6):2845-58.

6. Li C-W, Lim S-O, Xia W, Lee H-H, Chan L-C, Kuo C-W, *et al.* Glycosylation and stabilization of programmed death ligand-1 suppresses T-cell activity. *Nat Commun* **2016**;7:12632.
7. Yu Y, Liang Y, Li D, Wang L, Liang Z, Chen Y, *et al.* Glucose metabolism involved in PD-L1-mediated immune escape in the malignant kidney tumour microenvironment. *Cell Death Discov* **2021**;7(1):15.
8. Christofk HR, Vander Heiden MG, Harris MH, Ramanathan A, Gerszten RE, Wei R, *et al.* The M2 splice isoform of pyruvate kinase is important for cancer metabolism and tumour growth. *Nature* **2008**;452(7184):230-3.
9. Elf SE, Chen J. Targeting glucose metabolism in patients with cancer. *Cancer* **2014**;120(6):774-80.
10. Pålsson-McDermott EM, Dyck L, Zaslona Z, Menon D, McGettrick AF, Mills KHG, *et al.* Pyruvate Kinase M2 Is Required for the Expression of the Immune Checkpoint PD-L1 in Immune Cells and Tumors. *Front Immunol* **2017**;8:1300.
11. Wang S, Yao F, Lu X, Li Q, Su Z, Lee J-H, *et al.* Temozolomide promotes immune escape of GBM cells via upregulating PD-L1. *Am J Cancer Res* **2019**;9(6):1161-71.
12. Lv H, Lv G, Chen C, Zong Q, Jiang G, Ye D, Cui X, He Y, Xiang W, Han Q, Tang L, Yang W, Wang H. NAD⁺ Metabolism Maintains Inducible PD-L1 Expression to Drive Tumor Immune Evasion. *Cell Metab.* **2021** Jan 5;33(1):110-127.
13. Yeung F, Hoberg JE, Ramsey CS, Keller MD, Jones DR, Frye RA, Mayo MW. Modulation of NF-kappaB-dependent transcription and cell survival by the SIRT1 deacetylase. *EMBO J.* **2004** Jun 16;23(12):2369-80.
14. Chen LF, Mu Y, Greene WC. Acetylation of RelA at discrete sites regulates distinct nuclear functions of NF-kappaB. *EMBO J.* 2002 Dec 2;21(23):6539-48.
15. Sampath D, Zabka TS, Misner DL, O'Brien T, Dragovich PS. Inhibition of nicotinamide phosphoribosyltransferase (NAMPT) as a therapeutic strategy in cancer. *Pharmacol Ther* **2015**;151:16-31.
16. Jiang Z, Lim S-O, Yan M, Hsu JL, Yao J, Wei Y, *et al.* TYRO3 induces anti-PD-1/PD-L1 therapy resistance by limiting innate immunity and tumoral ferroptosis. *J Clin Invest* **2021**;131(8) :e139434.

Minor :

There are formatting errors of graph annotation in many figures. For example, there should be one space between the numerical value and the unit symbol, please check and revise it.

<Response>

According to the reviewer's comment, we checked and corrected it in the articles and figures.

31st Jan 2024

Dear Dr. Wang,

Thank you for the submission of your revised manuscript to EMBO Molecular Medicine. We have now received the enclosed reports from the referees that were asked to re-assess it. As you will see the reviewers are now globally supportive and I am pleased to inform you that we will be able to accept your manuscript pending the following final amendments:

- 1) Please place individual sections of the manuscript in the following order: Title page - Abstract & Keywords - Introduction - Results - Discussion - Materials & Methods - Data Availability - Acknowledgements - Disclosure and Competing Interests Statement - The Paper Explained - For More Information - References - Figure Legends - Expanded View Figure Legends.
- 2) In the main manuscript file, please do the following:
 - Title: please remove the comma from the title
 - Data availability: The Data availability statement appears twice in the manuscript. Please ensure that only one statement is at the end of the Materials and Methods section
 - An Acknowledgments section should be provided after the Data Availability statement.
 - Please note that funding information should be given in the "Acknowledgements" section (not in its own separate section). In addition, the Natural Science Foundation of Shenyang should be re-entered in the manuscript submission system as there characters/font that could not be recognized by the system (沈阳市科学技术局)
 - Please include a "Disclosure and competing interests statement". We updated our journal's competing interests policy in January 2022 and request authors to consider both actual and perceived competing interests. Please review the policy <https://www.embopress.org/competing-interests> and update your competing interests if necessary.
 - Author contributions: Please remove it from the manuscript. We only need the author contributions in our submission system.
 - Please correct the reference citation in the reference list. Currently the references are listed numerically, but they should be alphabetical. Where there are more than 10 authors on a paper, note that only 10 will be listed, followed by "et al.". Please check "Author Guidelines" for more information on proper reference formatting.
<https://www.embopress.org/page/journal/17574684/authorguide#referencesformat>
- 3) The Paper Explained: Please add "The Paper Explained" to the main manuscript text. It is not necessary to upload this as a separate document. Please check "Author Guidelines" for more information.
<https://www.embopress.org/page/journal/17574684/authorguide#researcharticleguide>
- 4) For more information: This space should be used to list relevant web links for further consultation by our readers. Could you identify some relevant ones and provide such information as well? Some examples are patient associations, relevant databases, OMIM/proteins/genes links, author's websites, etc...
- 5) In the Materials and Methods, please take care of the following:
 - ChIP: Insufficient information is given on how the ChIP was performed. Please provide the full details on the methods used.
 - Cell lines: Please include all information requested in the author checklist for cell lines used in the manuscript (please ensure catalog number, clone number, and/or RRID are included). Please also be sure to include a sentence in the Materials and Methods as to whether or not the cell lines were recently authenticated.
 - human research participants: You have indicated that the study involves human participants in the Author Checklist and the IRB protocol number is given in the Materials & Methods. Please ensure that you include a statement confirming that informed consent was obtained from all subjects and that the experiments conformed to the principles set out in the WMA Declaration of Helsinki and the Department of Health and Human Services Belmont Report as is written in the Author Checklist.
 - Please ensure that a statement on whether or not blinding was done is included in the Materials and Methods even if no blinding was done. Please also update this in the Author Checklist.
 - Antibodies: please ensure that company name, catalog number, and dilutions/amounts of each antibody are reported. Currently the catalog number is missing for antibodies in Sections 4.9, 4.11, 4.15, and 4.20 and dilutions/amounts are missing for all antibodies in Section 4.10 Materials & Methods. All information seems to be missing in section 4.13. Insufficient information on antibodies used for ChIP in Section 4.21.
- 6) For the figures and figure legends, please take care of the following:
 - Please remove all figures from main manuscript file and leave only main figure legends placed after the references.
 - There is a docx file with 7 Supplementary figures and their legends - these should either be separately uploaded as Expanded View figures (up to 5 of them) or can remain in the single file renamed as the Appendix. Callouts in the manuscript should be updated as well to match the updated name of each figure. In either case, each figure will need to fit onto one page. If you choose to make these EV figures, they should be labelled as Fig EV1-5, and the legends should stay in the manuscript, with the heading Expanded View Figures Legends, and placed after the main figure legends. If you choose to keep these figures in a single file named the Appendix, please refer to the figures as Appendix Figures S1-S7. Please also ensure that the each legend appears under each figure. The appendix should be uploaded in PDF format and needs a table of content with page numbers.
 - There is a docx file named Supplementary Table 1 and has 3 spectra labeled Figure 1-3. The file name needs to be corrected and/or the spectra should be included in the Appendix as Appendix Figures. See above for further directions here. Callouts in the manuscript will need to be corrected to match the updated figure names.
 - There is a pdf named Table S2 - this should be corrected - if this an Expanded View table then it should be Table EV1; if it is an

Appendix Table it should be labeled as Appendix Table S1 and included in the Appendix. Callouts in the manuscript will need Please check "Author Guidelines" for more information:

<https://www.embopress.org/page/journal/17574684/authorguide#figureformat>

- Re-use of image between Figure 6G And Supplementary Figure 6C - This needs to be clarified in the figure legend.
- Re-use of tumour images between Figure 6 A and E Balb/C mice - The control lane is repeated. The control and LZ90-treatment should be labeled and it needs to be clarified in the figure legend that the controls in 6E were repeated from 6A.
- Please note that a separate 'Data Information' section is required in the legends of figures 1b, d, g, j; 2d, f-h, k; 3b-c; 4c-e; 5b-f, 6b-h; 7b-c, e-h, supplementary figures 3c-d, f; 4a, c; 5a-e; 7b-e.
- Please note that the supplementary figure 3e does not contain any quantification graph, kindly rectify the statistics related information in the figure legend appropriately.
- Please define the annotated p values *****/**/*/#** in the legend of supplementary figures 6a-d; as appropriate.
- Please indicate the statistical test used for data analysis in the legends of figures 1b, d, g, j; 2d, f-h, k; 3b; 4c-e; 5b-f, 6b-h; 7b-c, e-h, supplementary figures 3c-d, f; 4a, c; 5b-c, d-e; 6a-d; 7c-e.
- Please note that in figure 3b, supplementary figures 4a, c; 5b-c; there is a mismatch between the annotated p values in the figure legend and the annotated p values in the figure file that should be corrected.
- Please note that information related to n is missing in the legends of figures 1d, g, j; 2d, f-h, k; 3b (i-ii)-c; 4c-e; 5b-f, 6c-d, g-h; 7e-f, supplementary figures 2d; 3c-d, f; 4a, c; 5a; 6b-d.
- Although 'n' is provided, please describe the nature of entity for 'n' in the legends of figures 7g-h.
- Please note that the error bar is not defined in the legend of supplementary figure 2d.
- Please note that in figure 6f the scale bar unit should be corrected from μM to μm .
- Please note that we require exact p-values to be reported. Currently exact p-values are not provided.

7) Source Data: Please ensure that the Source Data file names are updated to match the updated names of the Appendix/Expanded view files.

8) Synopsis:

- Synopsis image: Figure 8 and the Synopsis figure are currently the same. Please provide a new synopsis figure that is not included in the main manuscript, but uploaded separately as a jpeg with no legend. Please ensure that the size of the synopsis image is 550 pixels wide x (250-400) pixels high. The synopsis image should be a simple graphic that summarises the main findings of the manuscript on a glance.
- Synopsis text: Please use the passive voice - i.e. for bullet point 1, please remove 'our results revealed that'. In addition the last bullet point should be edited for English as it does not currently make sense.
- Please check your synopsis text and image before submission with your revised manuscript. Please be aware that in the proof stage minor corrections only are allowed (e.g., typos).

9) As part of the EMBO Publications transparent editorial process initiative (see our policy here:

https://www.embopress.org/transparent-process#Review_Process), EMBO Molecular Medicine will publish online a Peer Review File (PRF) to accompany accepted manuscripts. This file will be published in conjunction with your paper and will include the anonymous referee reports, your point-by-point response and all pertinent correspondence relating to the manuscript. Let us know whether you agree with the publication of the PRF and as here, if you want to remove or not any figures from it prior to publication. Please note that the Authors checklist will be published at the end of the PRF.

10) Please provide a point-by-point letter INCLUDING my comments as well as the reviewer's reports and your detailed responses (as Word file).

I look forward to reading a new revised version of your manuscript as soon as possible.

Yours sincerely,

Poonam Bheda

Poonam Bheda, PhD
Scientific Editor
EMBO Molecular Medicine

*** Instructions to submit your revised manuscript ***

To submit your manuscript, please follow this link:

<https://embomolmed.msubmit.net/cgi-bin/main.plex>

***** Reviewer's comments *****

Referee #1 (Remarks for Author):

In this revision, the biological mechanisms and therapeutic advantages of dual NAMPT and PD-L1 inhibition were strengthened. Before the acceptance of this work a minor point remains to be addressed. Comparison of dual NAMPT-PD-L1 inhibitor with other NAMPT-based dual inhibitors (e.g. NAMPT-HDAC inhibitors, NAMPT-DNA inhibitors, NAMPT-IDO1 inhibitors) and NAMPT degraders should be added into the DISCUSSION section.

Referee #2 (Remarks for Author):

I have no further comments.

Referee #3 (Comments on Novelty/Model System for Author):

The author investigated the reciprocal feedback regulation, between NAMPT and PD L1 and designed a dual targeting inhibitor LZFPN 90 to disrupt PD 1/PD L1 interactions and inhibit NAMPT activity, thereby enhancing T cell mediated antitumor immunity. The observation of a negative correlation between the expression of NAMPT and PD L1 in various cancer cell lines is novel, and the dual inhibitor exhibited strong inhibition to both targets. In the revised manuscript, the authors provided some mechanistic evidence of the reciprocal feedback regulation between NAMPT and PD-L1.

Referee #3 (Remarks for Author):

The authors tried hard to address all the questions raised by the reviewers. They basically addressed all the technical issues, and tried to investigate the correlation between NAMPT and PD- L1. These mechanistic studies, are preliminary in my opinion, but may inspire future investigations. Nevertheless, this compound is the first dual inhibitor of PD-L1 and NAMPT, I thereby recommend publish this work.

Revision Decision

Response to Reviewers

Point-by-point reply to reviewers' comments

Referee #1 (Remarks for Author):

In this revision, the biological mechanisms and therapeutic advantages of dual NAMPT and PD-L1 inhibition were strengthened. Before the acceptance of this work a minor point remains to be addressed. Comparison of dual NAMPT-PD-L1 inhibitor with other NAMPT-based dual inhibitors (e.g. NAMPT-HDAC inhibitors, NAMPT-DNA inhibitors, NAMPT-IDO1 inhibitors) and NAMPT degraders should be added into the DISCUSSION section.

<Response>

:Thanks to the reviewer for his/her valuable suggestions. According to the reviewer's comment, we added comparison of dual NAMPT-PD-L1 inhibitor with other NAMPT-based dual inhibitors (e.g. NAMPT-HDAC inhibitors, NAMPT-DNA inhibitors, NAMPT-IDO1 inhibitors) and NAMPT degraders into the DISCUSSION section.

→

With the in-depth researchs on the NAMPT, it has been found to many strategy target NAMPT achieve potent efficacy against tumor. Bi K *et al* reported targeting NAMPT by PROTAC technology gained potent tumor growth inhibition and demonstrated good biosafety (Bi, Cheng et al., 2023). Furthermore, several other studies showed that co-targeting strategy, including NAMPT-HDAC (Dong, Chen et al., 2017, Yue, Sun et al., 2024) and NAMPT-IDO1 (Wang, Ye et al., 2023), also displayed the encouraging effects in inhibiting the growth of tumors or overcoming drug resistance. Here, based on the interacted relationship between NAMPT and PD-L1, we developed a co-targeting strategy from NAMPT and PD-L1. Compared with other NAMPT-based target inhibitors, dual targeting NAMPT and PD-L1 inhibitor has the priority effect of activating the immune system. In addition, the reciprocal regulatory relationship between NAMPT and PD-L1 makes the inhibitor have an enhanced effect on both targets, which is not reported by other dual-targeted strategy. Taken together, NAMPT-based anti-tumor strategy might own a promising future, especially in tumor immune environment regulation.

References

- Bi K, Cheng J, He S, Fang Y, Huang M, Sheng C, Dong G (2023) Discovery of Highly Potent Nicotinamide Phosphoribosyltransferase Degradors for Efficient Treatment of Ovarian Cancer. *Journal of medicinal chemistry* 66: 1048-1062
- Dong G, Chen W, Wang X, Yang X, Xu T, Wang P, Zhang W, Rao Y, Miao C, Sheng C (2017) Small Molecule Inhibitors Simultaneously Targeting Cancer Metabolism and Epigenetics: Discovery of Novel Nicotinamide Phosphoribosyltransferase (NAMPT) and Histone Deacetylase (HDAC) Dual Inhibitors. *Journal of medicinal chemistry* 60: 7965-7983
- Wang K, Ye K, Zhang X, Wang T, Qi Z, Wang Y, Jiang S, Zhang K (2023) Dual Nicotinamide

Phosphoribosyltransferase (NAMPT) and Indoleamine 2,3-Dioxygenase 1 (IDO1) Inhibitors for the Treatment of Drug-Resistant Nonsmall-Cell Lung Cancer. *Journal of medicinal chemistry* 66: 1027-1047

Yue K, Sun S, Liu E, Liu J, Hou B, Qi K, Chou CJ, Jiang Y, Li X (2024) HDAC/NAMPT dual inhibitors overcome initial drug-resistance in p53-null leukemia cells. *European journal of medicinal chemistry* 266: 116127

Referee #2 (Remarks for Author):

I have no further comments.

Referee #3 (Comments on Novelty/Model System for Author):

The author investigated the reciprocal feedback regulation, between NAMPT and PD L1 and designed a dual targeting inhibitor LZFPN 90 to disrupt PD-1/PD-L1 interactions and inhibit NAMPT activity, thereby enhancing T cell mediated antitumor immunity. The observation of a negative correlation between the expression of NAMPT and PD-L1 in various cancer cell lines is novel, and the dual inhibitor exhibited strong inhibition to both targets. In the revised manuscript, the authors provided some mechanistic evidence of the reciprocal feedback regulation between NAMPT and PD-L1.

<Response>

We express our gratitude to the reviewer for validating our research. Your invaluable suggestions and insights have significantly contributed to the advancement of our scientific investigation.

Referee #3 (Remarks for Author):

The authors tried hard to address all the questions raised by the reviewers. They basically addressed all the technical issues, and tried to investigate the correlation between NAMPT and PD- L1. These mechanistic studies, are preliminary in my opinion, but may inspire future investigations. Nevertheless, this compound is the first dual inhibitor of PD-L1 and NAMPT, I thereby recommend publish this work.

<Response>

We express our gratitude to the reviewer for validating our research. Your invaluable suggestions and insights have significantly contributed to the advancement of our scientific investigation.

Revision Decision

Response to Scientific Editor

Point-by-point reply to editor's comments

1. We require both an institutional email address and ORCID for co-corresponding author Mingze Qin.

<Response>

: According to the editor's request, we have provided an institutional email address and ORCID for co-corresponding author Mingze Qin.

institutional email address: qinmingze@syphu.edu.cn

ORCID: 0000-0002-4634-535X

2. Please correct the reference citation in the reference list. Currently there are still references that have more than 10 authors where all are listed. When there are more than 10 authors on a paper, only the first 10 should be listed, followed by "et al.". Please check "Author Guidelines" for more information on proper reference formatting.

<https://www.embopress.org/page/journal/17574684/authorguide#referencesformat>

<Response>

: According to the "Author Guidelines" , we have corrected the references according to relevant requirements.

3. Please ensure that you also include (in the Materials and Methods section after the statement on the Declaration of Helsinki) that the experiments involving human subjects conformed to the principles set out by the Department of Health and Human Services Belmont Report as previously requested and is written in the Author Checklist.

<Response>

: According to the editor's suggestion, we include (in the Materials and Methods section after the statement on the Declaration of Helsinki) that the experiments involving human subjects conformed to the principles set out by the Department of Health and Human Services Belmont Report as previously requested and is written in the Author Checklist.

→

4.22 Human specimens statement

According to National Comprehensive Cancer Network (NCCN) guidelines, after consulting the patients, those who consented to the use of their remaining biological samples for scientific research provided the signed informed consent form for biological sample analysis. We used such patients' remaining biological samples in our study. Before being conducted, this study was approved by the ethics committee of the Affiliated Cancer Hospital of Shengjing Hospital of China Medical University (IRB No. 2023PS1105K). All protocols adhered to the principles outlined in the Declaration of Helsinki and the Department of Health and Human Services Belmont Report.

4. Exact p-values: for all panels were $P > 0.05$, the exact values should be given. For all panels with $p < 0.0001$ exact p-values should also be given, but you may consider using scientific notation to save space

<Response>

:According to the editor's suggestion, we have provided all panels with $P > 0.05$ and $p < 0.0001$ exact p-values.

5. Please remove the Synopsis figure from the Synopsis text and upload it as a separate file as a jpeg with no legend. Please ensure that the size of the synopsis image is 550 pixels wide x (250-400) pixels high.

<Response>

:We removed the Synopsis figure from the Synopsis text and upload it as a separate file as a jpeg with no legend., and ensure that the size of the synopsis image is 550 pixels wide 389 pixels high.

6. In the Synopsis text, please change "was negatively" to "is negatively" in bullet point 1, and in bullet point 2, please correct "would induce" to "induces". In the 4th bullet point, please change "were effectively" to "are effectively".

<Response>

: Thanks to the editor for his valuable suggestions, we made changes to the Synopsis text.

→

The proposed interaction of NAMPT and PD-L1 and their dual targeting by LZFPN-90 were explored, raising the possibility that the pharmacological blockade of NAMPT and simultaneous immune checkpoint blockade represent a promising strategy for cancer therapy.

- The expression of PD-L1 and NAMPT is negatively correlated by epigenetic and glycolysis regulatory mechanisms.
- Pharmacological inhibition of NAMPT results in the transcription upregulation of PD-L1 by SIRT-mediated acetylation change of NF-κB p65.
- Blocking PD-L1 induces NAMPT expression through a HIF-1-dependent glycolysis pathway.
- The PD-1/PD-L1 interaction and NAMPT activity are effectively inhibited by the dual NAMPT/PD-L1 targeting compound LZFPN-90.

22nd Feb 2024

Dear Dr. Wang,

We are pleased to inform you that your manuscript is now accepted for publication and is being sent to our publisher to be included in the next available issue of EMBO Molecular Medicine.

Yours sincerely,

Poonam Bheda, PhD
Scientific Editor
EMBO Molecular Medicine
